# Efficient Diversified Attack: Multiple Diversification Strategies Lead to the Efficient Adversarial Attacks

## Abstract

Deep learning models are vulnerable to adversarial examples (AEs). Recently, adversarial attacks that generate AEs by optimizing a multimodal function with many local optimums have attracted considerable research attention. Quick convergence to a nearby local optimum (intensification) and fast enumeration of multiple different local optima (diversification) are important to construct strong attacks. Most existing white-box attacks that use the model's gradient enumerate multiple local optima based on multi-restart; however, our experiments suggest that the ability to diversify based on multi-restart is limited. Therefore, we propose the multi-directions/objectives (MDO) strategy, which uses multiple search directions and objective functions for diversification. The MDO strategy showed higher diversification performance and promising attack performance. Efficient Diversified Attack (EDA), a combination of MDO and multi-target strategies, showed further diversification performance, resulting in state-of-the-art attack performance against more than 90% of 41 robust models compared to Adaptive Auto Attack ($A^3$). EDA particularly outperformed $A^3$ in attack performance and runtime for models trained on ImageNet, where the MDO strategy showed higher diversification performance. These results suggest a relationship between attack and diversification performances, which is beneficial to constructing more potent attacks.

## 1 Introduction

Deep neural networks (DNNs) with have demonstrated excellent performance in several applications are increasingly being used in safety-critical domains such as automated driving (Gupta et al., 2021), facial recognition (Adjabi et al., 2020), and cybersecurity (Liu et al., 2022b). However, DNNs are known to misclassify adversarial examples (AEs) generated by tiny perturbing inputs that are imperceptible to humans (Szegedy et al., 2014). Vulnerabilities caused by AEs can have fatal consequences, especially in safety-critical applications. Therefore, the robustness of DNNs against AEs is extremely important. To this end, several defense mechanisms, including adversarial training (AT) (Madry et al., 2018), which uses AEs during training and is one of the most effective defenses, have been proposed (Zhang et al., 2019b; Carmon et al., 2019; Ding et al., 2020; Addepalli et al., 2022). The model is expected to be more robust if more AEs are used in AT. Moreover, if AEs are generated faster, more robust models can be trained in less time. Thus, to improve the security of the DNNs, developing stronger and faster adversarial attacks in generating AEs is beneficial.

Adversarial attacks optimize a challenging nonconvex nonlinear function to find AEs. We focus on white-box attacks that use gradient-based optimization algorithms, assuming access to the outputs and gradients of the DNN. Higher objective function values increase misclassification chances, creating AE candidates out of local optima. The objective function of this problem is multimodal because its maximization involves a complex DNN. Because a multimodal function has a myriad of local optima, quick convergence to a nearby local optimum and fast enumeration of multiple different local optima are important. These are referred to as intensification and diversification, respectively (Glover & Samorani, 2019). Many existing gradient-based attacks are considered to achieve some degree of intensification because the objective value can be improved by moving in the gradient direction within a neighborhood. While many existing attacks diversify the search based on the

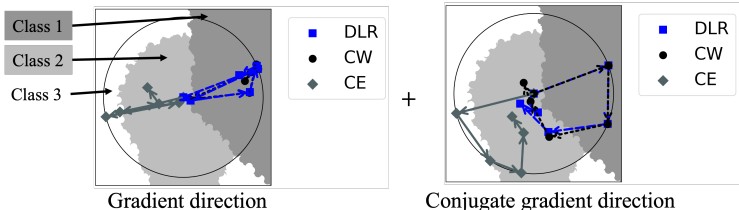

Figure 1: Illustration of the MDO strategy with two search directions and three objective functions on a toy model. The different search directions/objective functions search for different regions.

multi-restart (Dong et al., 2013; Madry et al., 2018; Croce & Hein, 2020b; Liu et al., 2022c), few studies have examined other diversification strategies. Therefore, further research on diversification is needed.

Our experiments using diversification indices in section 3.1 suggests that the multi-restart strategy does not yield high diversification performance, and different local solutions may be enumerated using different search directions and objective functions. Inspired by this observation, we propose the multi-directions/objectives (MDO) strategy that uses multiple search directions and objective functions. To implement the MDO strategy, we propose the Automated Diversified Selection (ADS) algorithm in section 3.2, which selects search directions and objective functions based on Diversity Index (DI) (Yamamura et al., 2022), and GS+LS in section 3.3, which is a search framework explicitly considering the diversification/intensification phase. Figure 1 is a toy example of an attack using MDO strategy.

We measured the effectiveness of diversification in terms of robust accuracy and the number of queries in successful attacks. The experimental results in section 4.1 suggest that ADS contributes to search diversification and GS+LS improves the attack performance using the MDO strategy. In addition, the MDO strategy found AEs for some inputs in less time than the multi-target (MT) strategy (Gowal et al., 2019), which is considered promising. From these results, it is expected that the combination of MDO and MT strategies can realize stronger and faster attacks. Therefore, we experimentally investigated the attack performance of Efficient Diversified Attack (EDA) using MDO and MT strategies. Experimental results in section 4.2 show that EDA exhibits higher diversification performance than the attack using MDO strategy alone, resulting in better attack performance in less computation time than the state-of-the-art (SOTA) attacks, including Adaptive Auto Attack ($A^3$) (Liu et al., 2022c) and Auto Attack (AA) (Croce & Hein, 2020b). Given the difference in robust accuracy between AA and $A^3$, the improvement in robust accuracy of EDA compared to $A^3$ is large. The above experimental results suggest a relationship between diversification and search performance and that improving diversification leads to improved attack.

The major contributions of this study are summarized below.
1. **Multi-directions/objectives (MDO)** strategy and its implementation: A novel search strategy using multiple search directions and objective functions, realized by Automated Diversified Selection (ADS) algorithm and GS+LS.
2. **Efficient Diversified Attack (EDA)**: A faster and stronger attack using MDO and MT strategies.

## 2 PRELIMINARIES

### 2.1 PROBLEM SETTINGS

Let $g : D \to \mathbb{R}^C$ be a locally differentiable $C$-classifier, $\boldsymbol{x}_{\text{org}} \in D$ be a point with $c$ as the correct label, and $d : D \times D \to \mathbb{R}$ be a distance function. Given $\varepsilon > 0$, the feasible region $\mathcal{S}$ is defined as the set of points $\boldsymbol{x} \in D$ that are within a distance of $\varepsilon$ from $\boldsymbol{x}_{\text{org}}$, i.e., $\mathcal{S} := \{\boldsymbol{x} \in D \mid d(\boldsymbol{x}, \boldsymbol{x}_{\text{org}}) \le \varepsilon\}$. Then, we define an AE as $\boldsymbol{x}_{\text{adv}} \in \mathcal{S}$ satisfying $\arg\max_{i=1,\dots,C} g_i(\boldsymbol{x}_{\text{adv}}) \ne c$. Let $L$ be the objective function to search for $\boldsymbol{x}_{\text{adv}}$. The following expression is a formulation of one type of adversarial attack, the untargeted attack where the attacker does not specify the misclassification target.

$$\max_{\boldsymbol{x} \in \mathcal{S}} L(g(\boldsymbol{x}), c) \tag{1}$$

The formulation 1 aims to reduce the probability that $\boldsymbol{x}$ is classified in class $c$ by $g$. Therefore, $\boldsymbol{x}$ with a high objective value $L(g(\boldsymbol{x}), c)$ is more likely to be misclassified by $g$. When $d(\boldsymbol{x}_{\mathrm{adv}}, \boldsymbol{x}_{\mathrm{org}})$ is small, the norm of the adversarial perturbation is also small. The targeted attack aims at maximizing the probability that $\boldsymbol{x}_{\mathrm{adv}}$ is classified in a particular class $t \neq c$ by solving $\max_{\boldsymbol{x} \in \mathcal{S}} L(g(\boldsymbol{x}), c, t)$. For adversarial attacks on image classifiers, $D = [0, 1]^n$ and $d(\boldsymbol{v}, \boldsymbol{w}) := \|\boldsymbol{v} - \boldsymbol{w}\|_p, (p = 2, \infty)$ is typically used. In this study, we focus on the untargeted attack on image classifier using $d(\boldsymbol{v}, \boldsymbol{w}) := \|\boldsymbol{v} - \boldsymbol{w}\|_\infty$, referred to as $\ell_\infty$ attacks.

## 2.2 RELATED WORK

In the white-box attack, the initial point sampling $\phi$ determines $\boldsymbol{x}^{(0)}$ first. Then, the step size update rule $\psi$ and the update formula $\boldsymbol{\delta} = \boldsymbol{\delta}(L)$ updates the step size $\eta^{(k)} = \eta_\psi^{(k)}$ and the search direction $\boldsymbol{\delta}^{(k)}$, respectively. Subsequently, the search point $\boldsymbol{x}^{(k+1)}$ is calculated by the following formula.

$$\boldsymbol{x}^{(0)} \leftarrow \text{sampled by } \phi, \ \boldsymbol{x}^{(k+1)} \leftarrow P_\mathcal{S}\left(\boldsymbol{x}^{(k)} + \eta^{(k)}\boldsymbol{\delta}^{(k)}\right), \tag{2}$$

where $k$ is the iteration, and $P_\mathcal{S}$ is a projection onto $\mathcal{S}$. The search direction $\boldsymbol{\delta}^{(k)}$ is usually computed based on the gradient $\nabla L(g(\boldsymbol{x}^{(k)}), c)$. According to equation 2, attack methods are characterized by the initial point sampling $\phi$, step size update rule $\psi$, search direction $\boldsymbol{\delta}$, and objective function $L$. The tuple $a = (\phi, \psi, \boldsymbol{\delta}, L)$ is referred to as an attack $a$. Because the search direction $\boldsymbol{\delta}^{(k)}$ depends on $\boldsymbol{\delta}$ and $L$, $\boldsymbol{\delta}$ and $L$ are the most important components of the gradient-based attack.

Projected Gradient Descent (PGD) (Madry et al., 2018) is a fundamental white-box attack. PGD uses a fixed step size ($\psi_{\mathrm{fix}}$) and moves to the normalized gradient direction ($\boldsymbol{\delta}_{\mathrm{PGD}}$). Auto-PGD (APGD) (Croce & Hein, 2020b) is a variant of PGD using heuristic ($\psi_{\mathrm{APGD}}$) for updating step size and moves to the momentum direction ($\boldsymbol{\delta}_{\mathrm{APGD}}$). In addition, some studies use cosine annealing ($\psi_{\cos}$) (Loshchilov & Hutter, 2017) for updating step size. Auto-Conjugate Gradient attack (ACG) (Yamamura et al., 2022) uses $\psi_{\mathrm{APGD}}$ and moves to the normalized conjugate gradient direction ($\boldsymbol{\delta}_{\mathrm{ACG}}$). While the sort of steepest directions, such as $\boldsymbol{\delta}_{\mathrm{PGD}}$ and $\boldsymbol{\delta}_{\mathrm{APGD}}$, are suitable for intensification, the conjugate gradient-based direction is suitable for diversification. For the initial point, uniform sampling from $\mathcal{S}$ or input points ($\phi_{\mathrm{org}}$) are usually used. Output Diversified Sampling (ODS, $\phi_{\mathrm{ODS}}$) and its variant, which consider output diversity of the threat model (Tashiro et al., 2020; Liu et al., 2022c), are also used. For the objective functions, cross-entropy (CE) loss (Goodfellow et al., 2015) ($L_{\mathrm{CE}}$) and margin-based losses such as CW loss (Carlini & Wagner, 2017) ($L_{\mathrm{CW}}$), a variation of CW loss scaled by the softmax function ($L_{\mathrm{SCW}}$), and Difference of Logits Ratio (DLR) loss (Croce & Hein, 2020b) ($L_{\mathrm{DLR}}$) are often used. We denote the targeted version of these functions as $L^T$, e.g., $L_{\mathrm{CE}}^T$ for targeted CE loss.

The robustness of DNNs is usually evaluated using AA, combining four different attacks, including the MT strategy. The high computational cost of AA has motivated the research community to pursue faster attacks for AT and robustness evaluation (Gao et al., 2022; Xu et al., 2022). Composite Adversarial Attack (CAA) (Mao et al., 2021) combines multiple attacks by solving the additional multi-objective optimization problem, requiring the pre-execution of candidate attack methods on relatively large samples. Consequently, CAA is still computationally expensive. $A^3$ demonstrated SOTA attack performance in less computation time by improving the initial point sampling and discarding the hard-to-attack images. Some studies have investigated how to switch either the search direction or the objective function based on case studies to improve attack performance (Yamamura et al., 2022; Antoniou et al., 2022). However, to the best of our knowledge, this is the first study that investigates the combination of multiple search directions and objective functions based on search diversification. Please refer to appendix A for details, including mathematical formulas about existing attacks.

# 3 MULTI-DIRECTIONS/OBJECTIVES STRATEGY

## 3.1 MOTIVATION

We hypothesize that diversification contributes to attack performance. This section empirically demonstrates that attacks with multiple search directions ($\boldsymbol{\delta}$) and objective functions ($L$) can achieve more efficient diversification than attacks with a single $\boldsymbol{\delta}$ and $L$. Let $\mathcal{D} = \{\boldsymbol{\delta}_{\mathrm{PGD}}, \boldsymbol{\delta}_{\mathrm{APGD}}, \boldsymbol{\delta}_{\mathrm{ACG}}, \boldsymbol{\delta}_{\mathrm{Nes}}\}$

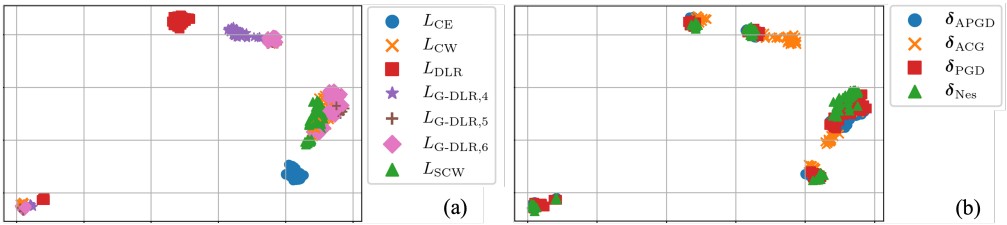

Figure 2: Violin plot of $DI(X^*(\boldsymbol{x}, a, 10), M)$. The left/right figure represent the attack with the same objective function/search direction, respectively.

Figure 3: 2D visualization of $X^*(\boldsymbol{x}, a, 10)$ using UMAP. Points of the same color in (a)/(b) represent points obtained using the same objective function/search direction, respectively.

be a set of search directions, and $\mathcal{L} = \{L_{\text{CE}}, L_{\text{CW}}, L_{\text{SCW}}, L_{\text{DLR}}\} \cup \{L_{\text{G-DLR},q} \mid q = 4, 5, 6\}$ a set of objective functions. We have proposed $\boldsymbol{\delta}_{\text{Nes}}$ and $L_{\text{G-DLR},q}$ in this study. $\boldsymbol{\delta}_{\text{Nes}}$ is the search direction of Nesterov's accelerated gradient (NAG) (Nesterov, 2004) normalized by the sign function to accommodate $\ell_\infty$ attacks. We refer to $L_{\text{G-DLR},q}$ as generalized-DLR (G-DLR) loss with the denominator of DLR loss extended from $g_{\pi_1}(\boldsymbol{x}) - g_{\pi_3}(\boldsymbol{x})$ to $g_{\pi_1}(\boldsymbol{x}) - g_{\pi_q}(\boldsymbol{x})$; $\pi_q \in Y$ denotes the class label that has the $q$-th largest value of $g(\boldsymbol{x})$. Mathematical expressions of $\boldsymbol{\delta}_{\text{Nes}}$ and $L_{\text{G-DLR},q}$ can be found in appendix B. Given an initial point selection method $\phi$ and a step size update rule $\psi$, we define a set of attacks as $\mathcal{A}(\phi, \psi) = \{a = (\phi, \psi, \boldsymbol{\delta}, L) \mid \boldsymbol{\delta} \in \mathcal{D}, L \in \mathcal{L}\}$. Let us consider an attack $a$ that iterates $N_{\max}$ starting with $R$ initial points to find AEs for the image $\boldsymbol{x}_i \in D$. Let $(\boldsymbol{x}_i)_{a,r}^{(k)}$ be a search point in the $k$-th iteration of an attack $a$ starting at the $r$-th initial point. Then, let $(\boldsymbol{x}_i)_{a,r}^*$ be the search point with the highest objective value obtained by an attack $a$ from $r$-th initial point, and $X^*(\boldsymbol{x}_i, a, R) \subset D$ be the set of search points with the highest objective values found by an attack $a$ from $R$ initial points. In the following paragraphs, we analyze the characteristics of the attack $a \in \mathcal{A}(\phi_{\text{ODS}}, \psi_{\cos})$, which uses a single $\boldsymbol{\delta}$ and $L$, based on the experimental results with $N_{\max} = 30$ and $R = 10$. For this experiment, we used 10,000 images from CIFAR-10 (Krizhevsky et al., 2009) as test samples and attacked the robust model proposed by Sehwag et al. (2022). Please refer to appendix C for more information and results of other models.

**Limited diversification ability of attacks using single $\boldsymbol{\delta}$ and $L$:** We quantify the diversity of $X^*(\boldsymbol{x}_i, a, R)$ using DI to reveal the diversification ability of the attack $a$, which uses a single $\boldsymbol{\delta}$ and $L$. Figure 2 shows the violin plot of DI. DI quantifies the degree of density of any point set as a value between 0 and 1 based on the global clustering coefficient of the graph. DI tends to be small when the point set forms clusters. We computed $DI\left(X^*(\boldsymbol{x}_i, a, R), M\right)$ for 10,000 images and all attacks $a \in \mathcal{A}(\phi_{\text{ODS}}, \psi_{\cos})$ as well as the mean and standard deviation for the first, second, and third quartiles of DI over the attacks. $M$ denotes the size of the feasible region. In this study, we used the same value of $M$ as in Yamamura et al. (2022). The mean and standard deviation of the first, second, and third quartiles were $0.190 \pm 0.019$, $0.223 \pm 0.023$, and $0.269 \pm 0.033$, respectively. These DI values suggest that the diversity of the best point set $X^*(\boldsymbol{x}_i, a, R)$ is relatively low. Thus, the diversification ability of $a$ seems to be limited.

**Attacks using multiple $\boldsymbol{\delta}$ and $L$ can lead to efficient diversification:** Figure 3 shows the best point set $X^*(\boldsymbol{x}_i, a, R)$ embedded in a two-dimensional space using Uniform Manifold Approximation and Projection (UMAP) (McInnes et al., 2018). Dimensionality reduction methods, such as UMAP, preserve the maximum possible distance information in high-dimensional spaces as possible. In fig. 3 (a), the points obtained by the attack using the same $L$ are plotted in the same color. Figure 3 (a) shows that sets of best search points obtained from searches with different $L$ tend to form different clusters. Figure 3 (b) is the same as fig. 3 (a), except that the points obtained by the attack using the

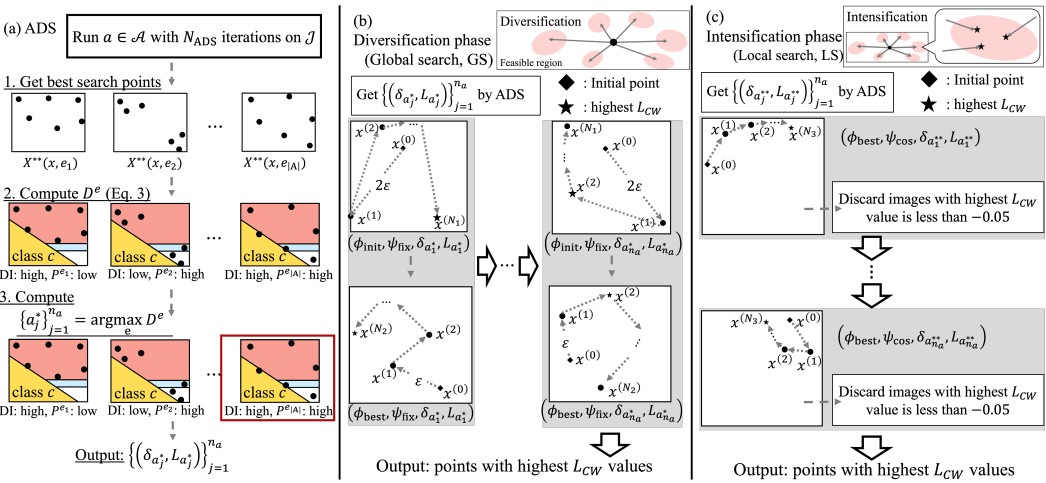

Figure 4: (a) The procedure of ADS. The black circle represents a single search point, and DI tends to be smaller when the search points form clusters. Regions of different colors represent different classification classes, and the more search points are distributed around multiple regions, the higher the degree of diversification in the output space, i.e., the value of $P_i^e$. (b) The procedure of GS. (c) The procedure of LS. Please refer to section 3.3 for more details of (b)/(c).

same $\boldsymbol{\delta}$ are depicted in the same color. Similarly, sets of best search points obtained by the attack using different $\boldsymbol{\delta}$ also tend to form different clusters. Based on these observations, it is possible to efficiently search for different local optima using different $\boldsymbol{\delta}$ and $L$, or an appropriate combination of both. We name this strategy the *multi-directions/objectives (MDO) strategy*.

## 3.2 AUTOMATED DIVERSIFIED SELECTION

The analysis in section 3.1 suggests that the MDO strategy can efficiently search for different local solutions. However, this strategy does not work well unless the combinations of $\boldsymbol{\delta}$ and $L$ are properly determined because different attacks may search similar regions. To address this issue, we propose the Automated Diversified Selection (ADS) algorithm, which selects the combinations of search directions and objective functions considering the degree of diversification in input/output space. Figure 4 (a) illustrates the procedure of ADS.

Let $X^{**}(\boldsymbol{x}_i, e)$ be the set of search points with the highest $L_{\text{CW}}$ values obtained by the attacks in $e = \{a_j\}_{j=1}^{n_a} \subset \mathcal{A}(\phi, \psi)$ to image $\boldsymbol{x}_i \in D$, and $\pi(\boldsymbol{x}_i)^{(k)} = \arg\max_{q \neq c_i} g_q((\boldsymbol{x}_i)_{a,1}^{(k)})$ be a class label with the highest prediction probability excluding the correct classification label. First, $N_{\text{ADS}}$ iterations of the attack candidate $a \in \mathcal{A}(\phi, \psi)$ are executed on the image set $\mathcal{J} \subset D$, with an initial step size of $\eta$. In this study, $\mathcal{J}$ is 1% of the images uniformly sampled from the entire test samples. Subsequently, the set of best search points $X^*(\boldsymbol{x}_i, a, 1)$ and class labels $\Pi_i^a = \{\pi(\boldsymbol{x}_i)^{(k)} \mid k = 1, \ldots, N_{\text{ADS}}\} \subset Y$ are obtained for each attack candidate $a \in \mathcal{A}(\phi, \psi)$. Let $A = \{\{a_j\}_{j=1}^{n_a} \subset \mathcal{A}(\phi, \psi) \mid |\{L_{a_j}\}_{j=1}^{n_a}| = n_a\}$ be a set of the candidate combination of $n_a$ attacks. From the observations in section 3.1, the degree of diversification may be greatly reduced when the selected attacks employ the same objective function. Therefore, constraints are imposed on $e \in A$ so that each attack uses a different $L$. The weighted average of the DI is calculated for all $e \in A$ to quantify the diversity of the best point set $X^{**}(\boldsymbol{x}_i, e)$ as follows:

$$D^e = \frac{1}{|\mathcal{J}|} \sum_{i=1}^{|\mathcal{J}|} P_i^e \cdot DI(X^{**}(\boldsymbol{x}_i, e), M), \tag{3}$$

where $P_i^e = |\cup_{a \in e} \Pi_i^a|$ is the number of types of classification labels with the highest prediction probability except the correct classification labels collected during the attack $a \in e$. $M$ is the size of the feasible region. A high DI indicates a high diversity of $X^{**}(\boldsymbol{x}_i, e)$, and a high $P_i^e$ indicates

a high degree of diversification in the output space. Finally, ADS outputs $\{(\boldsymbol{\delta}_{a_j^*}, L_{a_j^*})\}_{j=1}^{n_a}$ as the appropriate combinations of $\boldsymbol{\delta}$ and $L$, where $\{a_j^*\}_{j=1}^{n_a} = \arg\max_{e \in A(\phi, \psi)} D^e$. The pseudocode of ADS is provided in algorithm 3 in the appendix.

### 3.3 SEARCH FRAMEWORK FOR MDO STRATEGY

Considering the difference in diversification/intensification performance between PGD and CG-based search directions reported by Yamamura et al. (2022), we propose a search framework consisting of a diversification phase (global search, GS) and an intensification phase (local search, LS). GS searches a wide area, and LS searches the nearby area around the best point obtained by GS in more detail to improve the objective value. Figure 4 (b)/(c) illustrates the procedure of GS/LS, respectively. This framework is called *GS+LS*. The pseudocode of GS+LS is described in algorithm 4 in the appendix.

**Diversification phase:** GS uses large step sizes to search a broader area. First, ADS is executed with a step size of $\eta = 2\varepsilon$ and initial point sampling $\phi_{\text{init}}$ to determine the pairs $\{(\boldsymbol{\delta}_{a_j^*}, L_{a_j^*})\}_{j=1}^{n_a}$, where $(a_1^*, a_2^*, \ldots, a_{n_a}^*) \subset \mathcal{A}(\phi_{\text{init}}, \psi_{\text{fix}})$. Then, the following process is performed for each pair $(\boldsymbol{\delta}_{a_j^*}, L_{a_j^*})$. (1) Select an initial point by the initial point sampling $\phi_{\text{init}}$. (2) Perform $N_1$ iterative searches with initial step size of $2\varepsilon$, using step size update rule $\psi_{\text{fix}}$. (3) Perform $N_2$ iterative searches starting at the point with the highest $L_{\text{CW}}$ value found in (2), using initial step size of $\varepsilon$ and step size update rule $\psi_{\text{fix}}$. GS ends when (1)$\sim$(3) are executed for all selected pairs of $\boldsymbol{\delta}$ and $L$.

**Intensification phase:** LS takes the solution with the highest $L_{\text{CW}}$ value found by the GS as the initial point and searches for different local optimums within a range not far from the initial point. ADS is executed with a step size of $\eta = \varepsilon/2$ to determine the pairs $\{(\boldsymbol{\delta}_{a_j^{**}}, L_{a_j^{**}})\}_{j=1}^{n_a}$, where $(a_1^{**}, a_2^{**}, \ldots, a_{n_a}^{**}) \subset \mathcal{A}(\phi_{\text{best}}, \psi_{\text{fix}})$, and $\phi_{\text{best}}$ denotes the initial point sampling that uses the solution with the highest $L_{\text{CW}}$ value as the initial point. Subsequently, the following process is performed for each pair $(\boldsymbol{\delta}_{a_j^{**}}, L_{a_j^{**}})$. (1) Perform $N_3$ iterative searches with the initial point determined by $\phi_{\text{best}}$, and the initial step size of $\varepsilon/2$, using step size update rule $\psi_{\text{cos}}$. Note that $\psi_{\text{APGD}}$ is used to update the step size when the search direction $\boldsymbol{\delta}_{a_j^{**}}$ is equal to $\boldsymbol{\delta}_{\text{ACG}}$, aiming for a better intensification performance. (2) Perform the same search as in (1) on the images with the highest $L_{\text{CW}}$ values greater than or equal to $-0.05$ to accelerate the intensification. LS ends when (1) and (2) are executed for all selected pairs of $\boldsymbol{\delta}$ and $L$.

## 4 EXPERIMENTS

The efficacy of the proposed methods was examined through a series of experiments involving an $\ell_\infty$ attack against $\ell_\infty$ defense models listed in RobustBench (Croce et al., 2021).

**Dataset and models:** We used 41 models and 21 different defenses[1], including 25 models trained on CIFAR-10, 11 models trained on CIFAR-100 (Krizhevsky et al., 2009), and five models trained on ImageNet (Russakovsky et al., 2015). We performed $\ell_\infty$ attacks on 10,000 images with $\varepsilon = 8/255$ for CIFAR-10 and CIFAR-100 models and on 5,000 images with $\varepsilon = 4/255$ for ImageNet models. The test images were sampled in the same way as in RobustBench. Owing to the space limitation, the experimental results for the following nine models are described in the text: 1. ResNet-18 (RN-18) (Sehwag et al., 2022), 2. WideResNet-28-10 (WRN-28-10) (Carmon et al., 2019), 3. WRN-70-16 (Rebuffi et al., 2021), 4. PreActResNet-18 (PARN-18) (Rice et al., 2020), 5. WRN-34-10 (Sitawarin et al., 2021), 6. WRN-70-16 (Gowal et al., 2020), 7. RN-18 (Salman et al., 2020), 8. RN-50 (Engstrom et al., 2019), and 9. RN-50 (Wong et al., 2020). The models 1$\sim$3, 4$\sim$6, and 7$\sim$9 are trained on CIFAR-10, CIFAR-100, and ImageNet, respectively. These numbers correspond to the "No." column in tables. Complete results are described in the appendix D. In addition, computation specifications are provided in appendix D.1.

**Hyperparameters:** The parameters of the ADS are $n_a = 5$ and $N_{\text{ADS}} = 4$, which are the number of pairs of $\boldsymbol{\delta}$ and $L$ and the number of iterations, respectively. These parameters were determined based on small-scale experiments in appendix D.2. The parameters of GS are the number of iterations $N_1 = 22, N_2 = 19$, and the initial step size $\eta = 2\varepsilon, \varepsilon$. The parameters of LS are the number of

---

[1]The used models are publicly available as of robustbench v1.1.

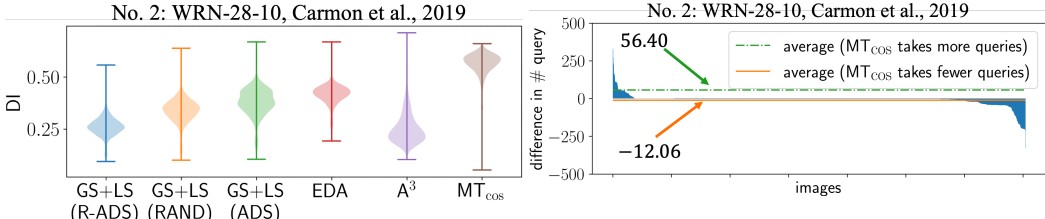

Figure 5: Violin plot of DI.

Figure 6: The difference between $MT_{cos}$ and GS+LS in #queries to find AEs.

iterations $N_3 = 59$ and the initial step size $\eta = \varepsilon/2$. The parameters in GS and LS were determined inspired by the APGD's step-size updating rule. Specifically, assuming $N = N_1 + N_2 + N_3 = 100$, $N_1 = \lceil 0.22N \rceil$ and $N_2 = \lceil 0.19N \rceil$ correspond to the first and second checkpoint in APGD's heuristic, respectively. Unless otherwise noted, the initial point sampling ($\phi_{init}$) in ADS and GS was Prediction Aware Sampling ($\phi_{PAS}$, PAS), a variant of ODS described in appendix E.

## 4.1 EVALUATION OF ADS AND GS+LS

This section describes the experimental results from a single run with a fixed random seed for reproducibility. We also checked that the standard deviations of robust accuracy over five runs of GS+LS were small as a part of the ablation study in appendix G. The key results are provided in the following sections.

**The combination of $\delta$ and $L$ selected by ADS brings a higher degree of diversification.** We compared the attack performance of GS+LS with three selection algorithms to realize the MDO strategy, including ADS, R-ADS, which finds the pairs minimizing equation 3, and RAND, which uniformly samples the pair. GS+LS(ADS) in fig. 5 and table 1 represents GS+LS with the selection algorithm ADS, and the same is applied to GS+LS(R-ADS) and GS+LS(RAND). Figure 5 shows that the DI of the best search points obtained by GS+LS (ADS) tends to be higher than that obtained by GS+LS (R-ADS/RAND), indicating ADS may select the pairs that enhance the diversification. According to table 1, GS+LS(ADS) showed lower robust accuracy than GS+LS(R-ADS/RAND), while GS+LS(R-ADS) showed significantly higher robust accuracy on some models. These results suggest that ADS selected the pairs that diversify the search more, leading to stronger attacks.

**GS+LS is one of the good implementations of the MDO strategy.** To confirm that GS+LS contribute to improvement in attack performance, we compared the robust accuracy obtained by GS+LS(ADS) with that obtained by Naive, which independently executes the attacks $a_j = (\phi_{PAS}, \psi_{cos}, \delta_{a_j^*}, L_{a_j^*})$ for $j = 1, \ldots, n_a$. The initial step size and number of iterations of each $a_j$ were set to $2\varepsilon$ and $N = 100$, respectively. Table 1 shows that the attack performance of GS+LS(ADS) is higher than that of Naive in most cases. In addition, GS+LS(ADS) showed higher attack performance than the standard version of AA[2] for several models in fewer queries. These experimental results suggest that GS+LS is one of the effective implementations of the MDO strategy. Subsequently, we compared the attack performance of GS(ADS), LS(ADS), and GS+LS(ADS) to confirm whether the combination of GS and LS works well. Table 1 shows that GS alone or LS alone cannot provide sufficient attack performance, but combining GS and LS can improve attack performance. Therefore, GS+LS, the combination of GS and LS, is considered to work well.

**MDO strategy tends to succeed in fewer queries attacks compared to the MT strategy.** To analyze the MDO strategy's characteristics, we compared GS+LS and $MT_{cos}$, a step size variant of MT-PGD (Gowal et al., 2019). In our notation, $MT_{cos}$ is expressed as $(\phi_{org}, \psi_{cos}, \delta_{GD}, L_{CW}^T)$. For the parameters of $MT_{cos}$, the number of explored target classes was $K = 9$, and iterations per target class was $N_T = 56$. Hence a slight difference was present between the number of queries of GS+LS and that of $MT_{cos}$. Figure 6 shows the difference between the number of queries spent by $MT_{cos}$ and GS+LS for images that both succeeded in attack. The positive value means $MT_{cos}$ spent more queries than GS+LS. Figure 6 indicates that GS+LS found AEs in fewer queries than $MT_{cos}$ on average and much fewer queries for some images. Appendix D.5 provides further explanation.

---

[2]https://github.com/fra31/auto-attack

Table 1: Comparison in robust accuracy. The lower the robust accuracy, the higher the attack performance. The lowest robust accuracy is in **bold**, and the second lowest is underlined. The "#query" row shows the number of queries in the worst-case per image. The robust accuracy of AA is the value reported by RobustBench.

| No. | clean acc | AA | ACG 5 restarts | Naive | GS (ADS) | LS (ADS) | GS+LS (ADS) | GS+LS (R-ADS) | GS+LS (RAND) | $MT_{cos}$ |
|---|---|---|---|---|---|---|---|---|---|---|
| #query | | 6.1k | 500 | 501.12 | 206.12 | 296.12 | 502.24 | 502.24 | 502.24 | 504 |
| 1 | 84.59 | **55.54** | 56.19 | 55.68 | 55.97 | 56.40 | 55.58 | 55.58 | 55.61 | **55.54** |
| 2 | 89.69 | 59.53 | 60.10 | 59.70 | 60.13 | 59.92 | **59.46** | 59.53 | 59.56 | 59.54 |
| 3 | 88.54 | 64.25 | 64.80 | 64.53 | 65.08 | 64.81 | 64.32 | 64.54 | **64.23** | 64.28 |
| 4 | 53.83 | **18.95** | 19.48 | 19.08 | 19.30 | 19.67 | 18.97 | 18.99 | 18.98 | 18.99 |
| 5 | 62.82 | 24.57 | 25.69 | 24.74 | 24.94 | 25.27 | 24.65 | 24.68 | 24.71 | **24.55** |
| 6 | 69.15 | **36.88** | 37.84 | 37.19 | 37.46 | 38.04 | 36.96 | 37.53 | 37.05 | 36.95 |
| 7 | 52.92 | 25.32 | 26.40 | 25.46 | 25.66 | 25.56 | **25.22** | 25.44 | **25.22** | 25.24 |
| 8 | 62.56 | 29.22 | 31.54 | 29.64 | 30.42 | 30.14 | **29.20** | 29.26 | 29.56 | 29.34 |
| 9 | 55.62 | 26.24 | 28.46 | 26.84 | 27.58 | 28.10 | **26.22** | 26.36 | 26.26 | 26.40 |

**Targeted loss is good at diversification, while untargeted loss succeeds in attacks by biasing the search.** As described in fig. 5, $MT_{cos}$ showed higher diversification performance than GS+LS(ADS), while the attack performance was inconsistent. Although untargeted losses move the search points toward decreasing the correct prediction probability, this is not necessarily the case with targeted losses. Therefore, it is expected that multi-targeted attacks show higher diversification performance, but sometimes untargeted attacks show higher attack performance.

## 4.2 COMPARISON WITH THE SOTA ATTACKS

According to the comparison of GS+LS and $MT_{cos}$ in section 4.1, we expected that combining MDO and MT strategies would lead to a faster and more potent attack. The combination of MDO and MT strategies is called Efficient Diversified Attack (EDA). We used GS+LS(ADS) for the MDO strategy and the targeted attack $a^t = (\phi_{PAS}, \psi_{APGD}, \delta_{GD}, L_{CW}^T)$ with $N_4 = 100$ iterations and a target class $T$ selected based on a small-scale search for the multi-target strategy. The detail of the target selection scheme is described in appendix F. The pseudocode of EDA is described in algorithm 5 in the appendix. The comparative experiments of EDA with the standard version of AA and $A^3$ were conducted to investigate the performance of EDA. The parameters of $A^3$ were the default values in the official code[3]. The key results have been discussed below.

**EDA showed SOTA performance in less runtime.** The summary in table 2 shows that the attack performance of EDA exceeds that of $A^3$ for 38 out of 41 models, and the runtime of EDA is 86.9 % of that of $A^3$ on average. Although the improvement in robust accuracy by EDA is approximately $0.01 \sim 0.3\%$, it is sufficiently large compared to that by recent SOTA methods. We compared the runtime of EDA with $A^3$ because comparing the computation cost by the number of queries is difficult owing to the $A^3$'s adaptive determination of the number of queries depending on the threat model. Because the bottleneck of adversarial attacks is the forward/backward propagation (query), there is a close relationship between runtime and the number of queries. We discuss the complexity of the compared methods in the number of queries in appendix D.3 The EDA's performance may be further improved by considering the execution order of the attacks with selected $\delta$ and $L$. However, determining the execution order requires that $(n_a!)^2$ permutations be considered, which is computationally expensive. The experimental results suggest that EDA achieves a satisfactory trade-off between the computation cost and the attack performance.

**EDA specifically showed higher performance for ImageNet models.** EDA showed relatively large improvements in robust accuracy within almost half the runtime of $A^3$ against models 7 to 9, which were trained on ImageNet. Given that GS+LS(ADS) showed promising results against the same models in table 1, the MDO strategy may have advantages in the attack against models trained on ImageNet, which is more practical regarding image size and the number of classification classes.

---

[3] https://github.com/liuye6666/adaptive_auto_attack

Table 2: Average robust accuracy, DI, and computation time over five runs. The "acc" columns denote the accuracy; the "sec" columns denote the runtime in seconds; and the "DI" columns denote the mean and standard deviation of DI obtained by each method over all test images. The "ratio" column represents the ratio of the runtime of EDA to that of $A^3$. The "$\Delta$" shows the difference in robust accuracy after attacks by $A^3$ and EDA. The "Summary" row summarizes the experimental results for all 41 models. AA's robust accuracy is the value reported by RobustBench. The lowest robust accuracy is in bold. The "#query" is the number of queries per image in the worst-case.

| No. | clean | AA, #query:6.1k | $A^3$, #query:1k | | | EDA, #query:802.24 | | | | $\Delta$ |
|---|---|---|---|---|---|---|---|---|---|---|
| | acc | acc | acc | DI | sec | acc | DI | sec | ratio | acc |
| 1 | 84.59 | 55.54 | 55.53 | 0.27±0.09 | 1,103 | **55.49** | 0.38±0.05 | 589 | 0.53 | 0.04 |
| 2 | 89.69 | 59.53 | 59.44 | 0.26±0.08 | 4,220 | **59.40** | 0.41±0.05 | 3,316 | 0.79 | 0.03 |
| 3 | 88.54 | 64.25 | 64.24 | 0.23±0.09 | 28,722 | **64.20** | 0.37±0.05 | 24,652 | 0.86 | 0.04 |
| 4 | 53.83 | 18.95 | 18.89 | 0.34±0.13 | 1,734 | **18.88** | 0.39±0.06 | 1,237 | 0.29 | 0.01 |
| 5 | 62.82 | 24.57 | 24.56 | 0.32±0.12 | 4,925 | **24.50** | 0.38±0.06 | 1,985 | 0.40 | 0.07 |
| 6 | 69.15 | 36.88 | 36.87 | 0.31±0.14 | 22,741 | **36.81** | 0.42±0.06 | 15,641 | 0.69 | 0.06 |
| 7 | 52.92 | 25.32 | 25.22 | 0.26±0.07 | 2,943 | **25.11** | 0.43±0.05 | 1,667 | 0.57 | 0.10 |
| 8 | 62.56 | 29.22 | 29.32 | 0.29±0.06 | 9,352 | **29.01** | 0.44±0.04 | 3,159 | 0.34 | 0.30 |
| 9 | 55.62 | 26.24 | 26.42 | 0.27±0.07 | 8,737 | **26.12** | 0.43±0.05 | 4,278 | 0.51 | 0.31 |
| Summary (Total of 41 models) | | | AA | $A^3$ | | EDA | | | Average ratio | |
| **#Bolded (acc)** | | | 0/41 | 3/41 | | 38/41 | | | 0.869 | |

**The higher diversification performance is one of the reasons for EDA's performance.** According to fig. 5, EDA showed higher DI values than GS+LS, which indicates that the MT strategy enhanced the diversification performance of EDA. The point set with a higher value of DI is less likely to form clusters. Thus, the DI of the best point set obtained by EDA is higher than that obtained by $A^3$ in table 2 suggesting that EDA may search a larger area than $A^3$. Many gradient-based methods can enhance the performance of intensification by appropriately adjusting the step size such that intensification and diversification can be achieved simultaneously for high values of DI.

**MDO strategy mainly contributed to the attack performance of EDA.** We examined the ratio of AEs generated only by GS+LS, $a^t$, and both methods to entire images that EDA succeeded but $A^3$ failed. In the case of CIFAR-10, the percentages of AEs generated only by GS+LS, $a^t$, and both methods are 58.98%, 6.71%, and 34.31%, respectively. These values are averages over the 25 models trained on CIFAR-10. Similarly, the percentages are 35.02%, 8.28%, and 56.70% for CIFAR-100 and 25.90%, 15.51%, and 58.59% for ImageNet. Here, the percentages in CIFAR-100 are averages over 11 models, and those in ImageNet over five. This analysis suggests that the MDO strategy contributes to the attack performance more than the MT strategy and it can find AEs that are difficult to find for existing attacks.

**Additional results and ablations:** Appendix G shows the result of the ablation studies, including the hyperparameter sensitivity of EDA. Appendix D.7 describes the analysis of EDA for the model of Ding et al. (2020), which showed different trends. The analysis using the Euclid distance-based measure in appendix D.6 showed similar trends to DI-based analysis. EDA worked well for randomized defenses and transformer-based models as described in appendix H and appendix I, respectively. EDA showed high transferability as described in appendix L.

## 5 CONCLUSION

In this study, we experimentally confirmed that different local optimums can be efficiently enumerated using various search directions and objective functions. Based on this observation, we have proposed the MDO strategy and its implementation, including ADS and GS+LS. The experiments on 41 robust models have demonstrated that the MDO strategy realized by GS+LS has a higher diversification ability. In addition, EDA, a combination of MDO and MT strategy, showed higher attack and diversification performance than SOTA attacks. Though more appropriate indices may exist, these results suggest that the attack designed based on the DI shows higher diversification performance, resulting in a stronger attack.

REPRODUCIBILITY

**Problem setting and proposed methods.** The mathematical setting of the adversarial attack is provided in section 2.1. The details of $\delta_{\text{Nes}}$, $L_{\text{G-DLR}}$, ADS, GS+LS, PAS, target selection, and EDA are described in appendix B.1, appendix B.2, section 3.2, section 3.3, appendix E, appendix F, and section 4.2, respectively. In addition, the pseudocode of a basic white-box attack, ADS, GS+LS, targeted attack, and EDA are described in algorithm 1, algorithm 3, algorithm 4, algorithm 2, and algorithm 5, respectively. The limitations and assumptions are provided in appendix J. The time complexity of EDA is discussed in appendix D.3 In addition, the source code is provided as supplementary material.

**Experiments** The computer specification is described in appendix D.1. We chose the hyperparameter of the proposed methods based on preliminary experiments and APGD's heuristic step size update. The hyperparameters and related experiments are described in section 4, appendix D.2, and appendix G. To investigate the stability of proposed methods, we report the mean and standard deviation from five runs with different random seeds in section 4.2 and appendix G.2. The results in section 4.2 and appendix G.2 suggest a stable performance of the proposed methods. Owing to the computation cost, the remaining experiments were conducted with a single fixed random seed.

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

APPENDIX

This supplementary material provides additional information as follows.

1. The summary of abbreviations and mathematical notations defined in the main text (Tables 3 and 4).

2. The pseudocode of proposed methods (Algorithms 1 to 5).

3. More information about related work (Appendix A).

4. The proposed search direction and objective function (Appendix B).

5. Additional results of the analysis in section 3.1 (Appendix C).

6. Complete results of the experiments in section 4 (Appendix D).

7. The details of Prediction Aware Sampling (Appendix E).

8. The details of the targeted attack used in EDA (Appendix F).

9. Ablation study for EDA (Appendix G).

10. Experiments on randomized defenses (Appendix H).

11. Experiments on transformer-based models (Appendix I).

12. Limitations and assumptions (Appendix J).

13. Broader impacts (Appendix K).

14. Evaluation of EDA's transferability (Appendix L).

Table 3: Summary of abbreviations

| Existing attack techniques | |
|---|---|
| PGD | Projected Gradient Descent |
| MT-PGD | MultiTargeted-PGD |
| APGD | Auto-PGD |
| ACG | Auto Conjugate Gradient attack |
| AA | AutoAttack |
| CAA | Composite Adversarial Attack |
| $A^3$ | Adaptive Auto Attack |
| CE | Cross-entropy |
| DLR | Difference of Logit Ratio |
| ODS | Output Diversified Sampling |
| DI | Diversity Index |
| Proposed methods | |
| G-DLR | Generalized-DLR |
| NAG | Nesterov's accelerated gradient |
| PAS | Prediction Aware Sampling |
| MDO | Multi-directions/objectives |
| ADS | Automated Diversified Sampling |
| GS+LS | Global search + Local search |
| EDA | Efficient Diversified Attack |
| Others | |
| DNN | Deep neural network |
| AE | Adversarial example |
| AT | Adversarial training |
| SOTA | State-of-the-art |
| MT | Multi-target |
| UMAP | Uniform Manifold Approximation and Projection |

Table 4: Summary of mathematical notations

| Defined in Section 2.1 | |
|---|---|
| $g : D \to \mathbb{R}^C$ | Locally differentiable $C$-classifier. |
| $\boldsymbol{x}_{\text{org}} \in D$ | Original input. |
| $c \in Y = \{1, 2, \dots, C\}$ | The correct classification label of $\boldsymbol{x}_{\text{org}}$. |
| $d : D \times D \to \mathbb{R}$ | A distance function. |
| $\mathcal{S}$ | The feasible region. |
| $L : \mathbb{R}^C \times Y \to \mathbb{R}$ | Objective function (untargeted attack). |
| $L : \mathbb{R}^C \times Y \times Y \to \mathbb{R}$ | Objective function (targeted attack). |

| Defined in Section 2.2 | |
|---|---|
| $P_{\mathcal{S}} : D \to \mathcal{S}$ | Projection onto the feasible region $\mathcal{S}$. |
| $\phi$ | Initial point sampling. |
| $\psi$ | Step size update rule. |
| $\boldsymbol{\delta}$ | Update formula. |
| $\boldsymbol{x}^{(k)} \in D$ | Search point at iteration $k$. |
| $\eta^{(k)} \in \mathbb{R}$ | Step size at iteration $k$. |
| $\boldsymbol{\delta}^{(k)} \in \mathbb{R}^n$ | Search direction at iteration $k$. |
| $a = (\phi, \psi, \boldsymbol{\delta}, L)$ | An attack. |

| Defined in Section 3.1 | |
|---|---|
| $\pi_q \in Y$ | The class label that has the $q$-th largest value of $g(\boldsymbol{x})$. |
| $\mathcal{D}$ | A set of update formulas. |
| $\mathcal{L}$ | A set of objective functions. |
| $\mathcal{A}(\phi, \psi)$ | A set of attacks with initial point sampling $\phi$ and step size update rule $\psi$. |
| $(\boldsymbol{x})_{a,r}^{(k)} \in D$ | A search point in the $k$-th iteration of an attack $a$ starting at the $r$-th initial point. |
| $(\boldsymbol{x})_{a,r}^* = \underset{k=1,\dots,N_{\max}}{\arg\max}\ L_{\text{CW}}(g((\boldsymbol{x})_{a,r}^{(k)}), c)$ | The best search point found by an attack $a$ starting at $r$-th initial point. |
| $X^*(\boldsymbol{x}_i, a, R) = \{(\boldsymbol{x}_i)_{a,r}^* \mid r = 1, \dots, R\}$ | The set of best search points found by an attack $a$ with $R$ restarts. |
| $DI(X, M)$ | Diversity Index of a point set $X$. $M$ is the size of feasible region. |

| Defined in Section 3.2 | |
|---|---|
| $e = \{a_j\}_{j=1}^{n_a}$ | A combination of attacks. |
| $X^{**}(\boldsymbol{x}_i, e) = \cup_{a \in e} X^*(\boldsymbol{x}_i, a, 1)$ | The set of best search points found by a combination of attacks. |

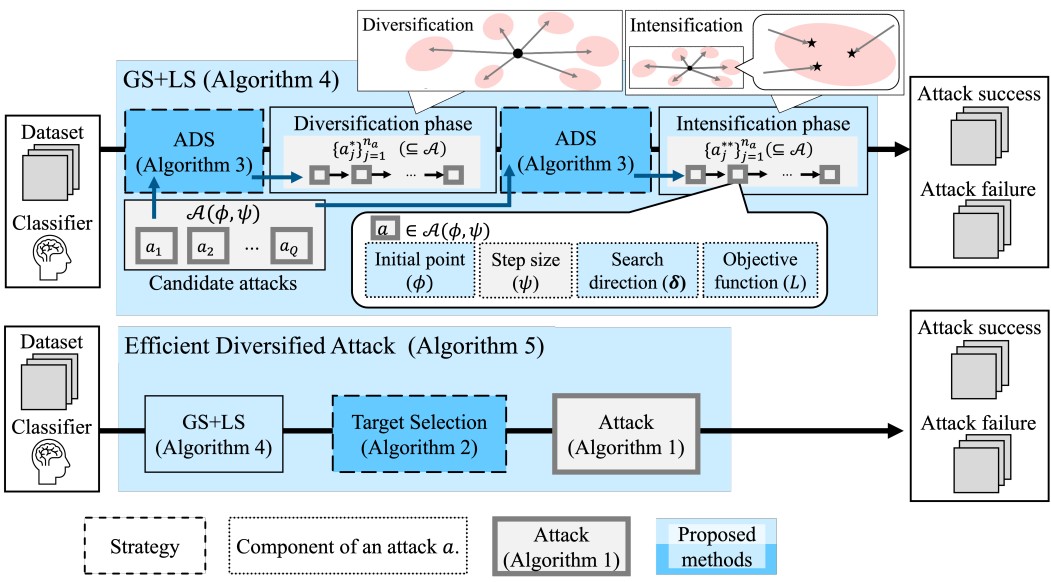

Figure 7: Relationship between Algorithms 1 to 5.

---

**Algorithm 1** Basic white-box Attacks without muti-restart

---

**Input:** $a = (\phi, \psi, \boldsymbol{\delta}, L)$: an attack, $N$: maximum iteration, $P_{\mathcal{S}}$: a projection function, $g$: a classifier, $\mathcal{I}$: a set of images

**Output:** $X^a$: a set of best search points, $F^a$: a set of best CW loss values, $\Pi^a$: a set of classification labels with the highest probabilities except for the correct classification class

1: **for** $i = 1, \ldots, |\mathcal{I}|$ **do**
2: $\quad \Pi_i^a \leftarrow \emptyset, \boldsymbol{x}_{\text{org}} \leftarrow \boldsymbol{x}_i \in \mathcal{I}, c_i \in Y \leftarrow$ correct classification label corresponding to $\boldsymbol{x}_i$
3: $\quad \boldsymbol{x}^{(0)} \leftarrow$ initialize by $\phi$, $\boldsymbol{x}_i^{adv} \leftarrow \boldsymbol{x}^{(0)}, f_i^{best} \leftarrow L_{\text{CW}}(\boldsymbol{x}^{(0)}, c_i)$
4: $\quad$ **for** $k = 0, \ldots, N-1$ **do**
5: $\qquad$ Update $\eta^{(k)}$ and $\boldsymbol{\delta}^{(k)}$ by update rule $\psi$ and search direction $\boldsymbol{\delta}$.
6: $\qquad \boldsymbol{x}^{(k+1)} \leftarrow P_{\mathcal{S}}\left(\boldsymbol{x}^{(k)} + \eta^{(k)} \cdot \boldsymbol{\delta}^{(k)}\right)$
7: $\qquad$ Update $\boldsymbol{x}_i^{adv}$ and $f_i^{best}$ $\qquad\qquad \triangleright \boldsymbol{x}_i^{adv}$ is equivalent to $(\boldsymbol{x}_i)_{a,1}^*$ in the main paper.
8: $\qquad \Pi_i^a \leftarrow \Pi_i^a \cup \{\arg\max_{j \neq c_i} g_j(\boldsymbol{x}^{(k+1)})\}$
9: $\quad$ **end for**
10: $\quad X^*(\boldsymbol{x}_i, a, 1) \leftarrow \{\boldsymbol{x}_i^{adv}\}$
11: $\quad X_i^a \leftarrow X^*(\boldsymbol{x}_i, a, 1), F_i^a \leftarrow \{f_i^{best}\}$
12: **end for**

---

**Algorithm 2** TargetSelection

---

**Input:** $\mathcal{I}$: a set of images, $g$: image classifier, $P_{\mathcal{S}}$: projection function, $K$: number of classification labels explored , $\phi^{initial}$: initial point sampling method, $N_s$: number of iterations in small-scale search

**Output:** $T$: Approximately easiest-to-attack classification label

1: $T \leftarrow \{0\}^{|\mathcal{I}|}$ $\qquad\qquad\qquad\qquad\qquad \triangleright$ initialize target label of misclassification
2: $a^t \leftarrow (\phi^{initial}, \psi_{\cos}, \boldsymbol{\delta}_{\text{GD}}, L_{\text{CW}}^T)$
3: **for** $t = 0, \ldots K$ **do**
4: $\quad X^{a^t}, F^{a^t}, \Pi^{a^t} \leftarrow \text{Attack}(a^t, N_s, P_{\mathcal{S}}, g, \mathcal{I})$ $\quad \triangleright$ Run targeted attack with the target class $\pi_t$
5: $\quad$ Update $T_i$ for images $\boldsymbol{x}_i$ that replaces the past highest loss value
6: **end for**

---

---

**Algorithm 3** Automated Diversified Selection (ADS)

---

**Input:** $\mathcal{J}$: Images, $\mathcal{A}$: a set of attacks, $N_{\text{ADS}}$: maximum iterations, $P_{\mathcal{S}}$: projection function, $g$: image classifier, $M$: the size of feasible region, $n_a$: number of attacks to be selected

**Output:** $\{(\boldsymbol{\delta}_{a_j^*}, L_{a_j^*})\}_{j=1}^{n_a}$: The pairs of search direction and objective function

1: **for** $a = (\phi, \psi, \boldsymbol{\delta}, L) \in \mathcal{A}$ **do**
2:     $X^a, F^a, \Pi^a \leftarrow \text{Attack}(a, N_{\text{ADS}}, P_{\mathcal{S}}, g, \mathcal{J})$      $\triangleright$ Run the an attack $a$ following Algorithm 1
3: **end for**
4: $A \leftarrow \{e \subset \mathcal{A} \mid |e| = n_a \wedge |\{L_{a_j}\}_{j=1}^{n_a}| = n_a\}$
5: **for** $e \in A$ **do**
6:     **for** $i = 1, \ldots, |\mathcal{J}|$ **do**
7:         $X^{**}(\boldsymbol{x}_i, e) \leftarrow \cup_{a \in e} X_i^a, P_i^e \leftarrow |\cup_{a \in e} \Pi_i^a|$
8:     **end for**
9:     Compute $D^e$ by equation 3
10: **end for**
11: $e^* = \{a_j^*\}_{j=1}^{n_a} \leftarrow \arg\max_{e \in A} D^e$
12: Get $\{(\boldsymbol{\delta}_{a_j^*}, L_{a_j^*})\}_{j=1}^{n_a}$ from $e^*$.

---

**Algorithm 4** GS+LS

---

**Input:** $\mathcal{I}$: a set of images, $g$: image classifier, $\varepsilon$: maximum norm of the adversarial perturbation, $\mathcal{D}$: a set of search directions, $\mathcal{L}$: a set of objective functions, $n_a$: number of pairs to be selected, $N_{\text{ADS}}$: search iterations in ADS, $M$: the size of feasible region, $N_1, N_2$: search iterations in GS, $N_3$: search iterations in LS

**Output:** $X^{\text{advs}}$: adversarial examples (AEs)

1: $X^{\text{advs}} \leftarrow \emptyset$
2: /* Diversification phase (GS) */
3: $\mathcal{J} \leftarrow$ uniformly sampled 1% images of $\mathcal{I}$.
4: $\{(\boldsymbol{\delta}_{a_j^*}, L_{a_j^*})\}_{j=1}^{n_a} \leftarrow \text{ADS}\,(\mathcal{J}, \mathcal{A}(\phi_{\text{PAS}}, \psi_{\text{fix}}), N_{\text{ADS}}, P_{\mathcal{S}}, g, M, n_a)$
5: **for** $j = 1, \ldots, n_a$ **do**
6:     $a' \leftarrow (\phi_{\text{PAS}}, \psi_{\text{fix}}, \boldsymbol{\delta}_{a_j^*}, L_{a_j^*})$
7:     $X^{a'}, F^{a'}, \Pi^{a'} \leftarrow \text{Attack}(a', N_1, P_{\mathcal{S}}, g, \mathcal{I})$      $\triangleright$ Run the attack $a'$ following Algorithm 1
8:     $a'' \leftarrow (\phi_{\text{best}}, \psi_{\text{fix}}, \boldsymbol{\delta}_{a_j^*}, L_{a_j^*})$
9:     $X^{a''}, F^{a''}, \Pi^{a''} \leftarrow \text{Attack}(a'', N_2, P_{\mathcal{S}}, g, \mathcal{I})$      $\triangleright$ Run the attack $a''$ following Algorithm 1
10:     Update $X^{\text{advs}}$ by $X^{a''}$
11:     Update $\mathcal{I}$ by excluding images succeeded in attack.
12: **end for**
13: /* Intensification phase (LS) */
14: $\mathcal{I}_{sub} \leftarrow \mathcal{I}$      $\triangleright$ $\mathcal{I}_{sub}$ is used to filter the images to be attacked.
15: $\mathcal{J}' \leftarrow$ uniformly sampled 1% images of $\mathcal{I}$.
16: $\{(\boldsymbol{\delta}_{a_j^{**}}, L_{a_j^{**}})\}_{j=1}^{n_a} \leftarrow \text{ADS}\,(\mathcal{J}', \mathcal{A}(\phi_{\text{best}}, \psi_{\text{fix}}), N_{\text{ADS}}, P_{\mathcal{S}}, g, M, n_a)$
17: **for** $j = 1, \ldots, n_a$ **do**
18:     $a^l \leftarrow (\phi_{\text{best}}, \psi_{\text{cos}}, \boldsymbol{\delta}_{a_j^{**}}, L_{a_j^{**}})$      $\triangleright$ Use $\psi_{\text{APGD}}$ instead of $\psi_{\text{cos}}$ when $\boldsymbol{\delta}_{a_j^{**}} = \boldsymbol{\delta}_{\text{ACG}}$
19:     $X^{a^l}, F^{a^l}, \Pi^{a^l} \leftarrow \text{Attack}(a^l, N_3, P_{\mathcal{S}}, g, \mathcal{I}_{sub})$      $\triangleright$ Run the attack $a^l$ following Algorithm 1
20:     Update $X^{\text{advs}}$ by $X^{a^l}$.
21:     $\mathcal{I}' \leftarrow \left\{\boldsymbol{x}_i \in \mathcal{I} \mid -0.05 \leq F_i^{a^l} \leq 0\right\}$      $\triangleright$ Extract images for further intensification.
22:     $\mathcal{I}_{sub} \leftarrow \mathcal{I}_{sub} \cap \mathcal{I}'$
23:     Repeat lines 18-19
24: **end for**

---

---

**Algorithm 5** Efficient Diversified Attack (EDA)

---

**Input:** $\mathcal{I}$:a set of images, $g$: image classifier, $\varepsilon$: maximum norm of the adversarial perturbation, $\mathcal{D}$: a set of search directions, $\mathcal{L}$: a set of objective functions, $n_a$: number of pairs to be selected, $K$: number of classification labels to be explored, $N_{\text{ADS}}$: search iterations in ADS, $M$: the size of feasible region, $N_1, N_2$: search iterations in GS, $N_3$: search iterations in LS

**Output:** $X^{\text{advs}}$: AEs
1: $X^{\text{advs}} \leftarrow \emptyset$
2: $X^{\text{advs}} \leftarrow \text{GS+LS}(\mathcal{I}, g, \varepsilon, \mathcal{D}, \mathcal{L}, n_a, N_{\text{ADS}}, M, N_1, N_2, N_3)$          $\triangleright$ Algorithm 4
3: Update $\mathcal{I}$ by excluding images succeeded in attack.
4: /* Multi-target attack */
5: $T \leftarrow \text{TargetSelection}(\mathcal{I}, g, P_{\mathcal{S}}, K, \phi_{\text{PAS}}, 10)$
6: $a^t \leftarrow (\phi_{\text{PAS}}, \psi_{\text{APGD}}, \boldsymbol{\delta}_{\text{GD}}, L_{\text{CW}}^T)$                 $\triangleright$ Algorithm 2
7: $X^{a^t}, F^{a^t}, \Pi^{a^t} \leftarrow \text{Attack}(a^t, 100, P_{\mathcal{S}}, g, \mathcal{I})$      $\triangleright$ Run targeted attack with the target $T_i$
8: Update $X^{\text{advs}}$ by $X^{a^t}$.

---

## A  MORE INFORMATION ABOUT RELATED WORK

### A.1  PROCEDURE OF BASIC WHITE-BOX ATTACKS

This section provides a detailed explanation of basic white-box attacks introduced in section 2 in the main paper. We characterize the basic white-box attacks by the tuple $a = (\phi, \psi, \boldsymbol{\delta}, L)$, where $\phi$ is the initial point sampling, $\psi$ is the step size update rule, $\boldsymbol{\delta}$ is the search direction, and $L$ is the objective function for the attack. We refer to the tuple $a$ as an attack $a$. Algorithm 1 shows the procedure of the basic white-box attacks $a = (\phi, \psi, \boldsymbol{\delta}, L)$ without multi-restart. We assume that the basic white-box attack $a$ returns a set of best search points $X^a$, a set of best CW loss values $F^a$, and a set of classification labels with the highest probability except for the correct classification labels during the search $\Pi^a$. The basic white-box attacks iteratively update the search point $\boldsymbol{x}^{(k)}$ as lines 5& 6 in algorithm 1 to search for adversarial examples. At each iterations $k$, the best search point $\boldsymbol{x}_i^{adv}$ is updated if $L(g(\boldsymbol{x}_i^{adv}), c) \leq L(g(\boldsymbol{x}^{(k)}), c)$. After the $N$ iterations of updates, the attack procedure is finished.

For the attacks with a multi-restart strategy, algorithm 1 is repeated from different initial points.

### A.2  SEARCH DIRECTIONS

**Projected Gradient Descent**    Projected Gradient Descent (PGD) (Madry et al., 2018) is the most fundamental adversarial attack based on the steepest gradient descent. The search direction of PGD is computed as follows:

$$\boldsymbol{\delta}_{\text{PGD}}^{(k)} = \text{sign}\left(\nabla L(g(\boldsymbol{x}^{(k)}), c)\right) \tag{4}$$

Also, Fast Gradient Sign Method (FGSM) (Goodfellow et al., 2015) updates towards the same direction.

**Auto-PGD**    Auto-PGD (APGD) (Croce & Hein, 2020b) is a PGD variant that adjusts the step size and updates towards momentum direction in addition to the PGD's search direction. The search direction of APGD is defined as follows.

$$\boldsymbol{z}^{(k)} = P_{\mathcal{S}}\left(\boldsymbol{x}^{(k)} + \eta^{(k)}\boldsymbol{\delta}_{\text{PGD}}^{(k)}\right), \tag{5}$$

$$\boldsymbol{\delta}_{\text{APGD}}^{(k)} = \alpha(\boldsymbol{z}^{(k)} - \boldsymbol{x}^{(k)}) + (1-\alpha)(\boldsymbol{x}^{(k)} - \boldsymbol{x}^{(k-1)}), \tag{6}$$

where $\alpha$ is a coefficient of momentum term. APGD uses $\alpha = 1$ for the first iteration and $\alpha = 0.75$ for the remaining iterations. The step size $\eta^{(k)}$ is halved if the following conditions are satisfied at the $w_j \in W$ iteration, with the initial value $\eta^{(0)} = 2\varepsilon$.

1. $\displaystyle\sum_{i=w_{j-1}}^{w_j-1} \mathbf{1}_{L\left(g(\boldsymbol{x}^{(i+1)}), c\right) > L\left(g(\boldsymbol{x}^{(i)}), c\right)} < \rho \cdot (w_j - w_{j-1})$

2. $L_{\max}\left(g(\boldsymbol{x}^{(w_{j-1})}), c\right) = L_{\max}\left(g(\boldsymbol{x}^{(w_j)}), c\right)$ and $\eta^{(w_{j-1})} = \eta^{(w_j)}$

The sequence of checkpoints $W$ is computed based on the following gradual equation depending on the total number of iterations $N_{iter}$. $p_0 = 0, p_1 = 0.22, p_{j+1} = \min(p_j - p_{j-1} - 0.03, 0.06), w_j = \lceil p_j N_{iter} \rceil$. In our notation, $\psi_{\text{APGD}}$ denotes this step size updating rule.

**Auto-Conjugate Gradient attack**  Auto-Conjugate Gradient (ACG) attack (Yamamura et al., 2022) is inspired by the Conjugate Gradient method for nonlinear optimization problems. ACG performs a more diversified search than the attacks based on the steepest descent. The search direction of ACG is as follows.

$$\boldsymbol{y}^{(k)} = \nabla L(g(\boldsymbol{x}^{(k-1)}), c) - \nabla L(g(\boldsymbol{x}^{(k)}), c) \tag{7}$$

$$\beta^{(k)} = -\frac{\nabla L(\boldsymbol{x}^{(k)}, c)^T \boldsymbol{y}^{(k)}}{(\boldsymbol{y}^{(k)})^T \boldsymbol{\delta}_{\text{ACG}}^{(k-1)}} \tag{8}$$

$$\boldsymbol{\delta}_{\text{ACG}}^{(k)} = \nabla L(\boldsymbol{x}^{(k)}, c) + \beta^{(k)} \boldsymbol{\delta}_{\text{ACG}}^{(k-1)} \tag{9}$$

Moreover, Yamamura et al. (2022) proposed the *Diversity Index(DI)* to quantify the degree of diversification during the attacks. DI is defined as

$$\text{DI}(X, M) := \frac{1}{M} \int_0^M h(\theta; X)\, d\theta, \tag{10}$$

where $M = \sup\{\|\boldsymbol{x} - \boldsymbol{y}\|_2 \mid \boldsymbol{x}, \boldsymbol{y} \in \mathcal{S}\}$ is the size of the feasible region, $X$ is the set of search points, and $h(\theta; X)$ is the function of $\theta$ based on the global clustering coefficient of the graph $G(X, \theta) = (X, E(\theta)) = (X, \{(u, v) \mid \boldsymbol{u}, \boldsymbol{v} \in X, \|u - v\|_2 \leq \theta\})$.

### A.3 OBJECTIVE FUNCTIONS

**Cross-entropy loss** The untargeted version of cross-entropy (CE) loss is defined as follows.

$$L_{\text{CE}}(g(\boldsymbol{x}), c) = -g_c(\boldsymbol{x}) + \log\left(\sum_{j \neq c} \exp\left(g_j(\boldsymbol{x})\right)\right) \tag{11}$$

Also, the targeted version of CE loss is defined as follows.

$$L_{\text{CE}}^T(g(\boldsymbol{x}), c, t) = g_t(\boldsymbol{x}) - \log\left(\sum_{j \neq t} \exp\left(g_j(\boldsymbol{x})\right)\right), \tag{12}$$

where $t$ denotes the target label of misclassification. CE loss is known to be sensitive to the scaling of the logit, i.e., the attack performance significantly varies depending on the scaling of the logit (Carlini & Wagner, 2017; Croce & Hein, 2020b).

**CW loss** The untargeted version of CW loss is defined as follows.

$$L_{\text{CW}}(g(\boldsymbol{x}), c) = \max_{j \neq c} g_j(\boldsymbol{x}) - g_c(\boldsymbol{x}) \tag{13}$$

Also, the targeted version of CW loss is defined as follows.

$$L_{\text{CW}}^T(g(\boldsymbol{x}), c, t) = g_t(\boldsymbol{x}) - g_c(\boldsymbol{x}), \tag{14}$$

where $t$ denotes the target label of misclassification.

**Difference of Logit Ratio loss** The untargeted version of the Difference of Logit Ratio (DLR) loss is defined as follows.

$$L_{\text{DLR}}(g(\boldsymbol{x}), c) = \frac{\max_{j \neq c} g_j(\boldsymbol{x}) - g_c(\boldsymbol{x})}{g_{\pi_1} - g_{\pi_3}}, \tag{15}$$

where $\pi_q$ denotes the classification label with $q$-th highest value in $g(\boldsymbol{x})$. Also, the targeted version of DLR loss is defined as follows.

$$L_{\text{DLR}}^T(g(\boldsymbol{x}), c, t) = \frac{g_t(\boldsymbol{x}) - g_c(\boldsymbol{x})}{g_{\pi_1} - (g_{\pi_3}(\boldsymbol{x}) + g_{\pi_4}(\boldsymbol{x}))/2}, \tag{16}$$

where $t$ denotes the target label of misclassification.

## A.4 Comparison of existing attacks and EDA

Table 5 summarizes the characteristics of PGD-like attacks, Auto Attack (AA) (Croce & Hein, 2020b), Adaptive Auto Attack ($A^3$) (Liu et al., 2022c), and EDA in terms of diversification, intensification, and computational cost. The attacks in table 5 perform well in intensification because they include gradient-based attacks with appropriate step size management. Although PGD-like attacks have several variations, this section describes the representative one in table 5. PGD-like attacks and $A^3$ use multi-restart for diversification, and both attacks spend a relatively short computational time. However, $A^3$ outperforms PGD-like attacks because $A^3$ uses better initial point sampling. AA considers multiple objective functions and multi-target attacks for diversification in addition to multi-restart. While AA achieves a high attack success rate, AA is computationally expensive because AA consists of four attacks, including APGD with untargeted CE loss, APGD with targeted DLR loss, FAB attack (Croce & Hein, 2020a), and square attack (Andriushchenko et al., 2020). In the sense that AA uses different types of attacks, we can consider that AA employs the diversification strategy based on multi-$\delta$. However, AA uses only a single search direction to solve (2) in the white-box setting. Therefore, we do not consider AA an attack with a diversification strategy based on multi-$\delta$. In contrast, EDA, the proposed attack, uses all diversification strategies listed in table 5 and achieves a higher attack success rate in a short computation time.

Table 5: Characteristics of PGD-like attacks, AA, $A^3$, and EDA. multi-$L$ denotes the diversification using multiple objective functions, and multi-$\delta$ denotes the diversification using multiple search directions.

| Attacks | Diversification | | | | Intensification | Runtime |
| --- | --- | --- | --- | --- | --- | --- |
| | multi-$L$ | multi-$\delta$ | multi-restart | multi-target | | |
| PGD-like | - | - | ✓ | - | ✓ | short |
| AA | ✓ | - | ✓ | ✓ | ✓ | long |
| $A^3$ | - | - | ✓ | - | ✓ | short |
| EDA | ✓ | ✓ | ✓ | ✓ | ✓ | short |

## B  The proposed search direction and objective function

### B.1  Search direction inspired by Nesterov's Accelerated Gradient

Although some attacks were inspired by Nesterov's accelerated gradient (Nesterov, 2004), most of them apply constant value to the coefficient of momentum (Lin et al., 2020; Liu et al., 2022a). However, the original Nesterov's accelerated gradient method determines the coefficient of momentum term by solving the quadratic equations. So then we try to adopt Nesterov's accelerated gradient to $\ell_\infty$ attacks. Mathematically, $\delta_{\text{Nes}}$ is computed by the following equations.

$$\rho^{(k)} \text{ is a positive solution of } (\rho^{(k)})^2 = (1 - \rho^{(k)})(\rho^{(k-1)})^2 \tag{17}$$

$$\gamma^{(k)} \leftarrow \frac{\rho^{(k-1)}\left(\rho^{(k-1)} - 1\right)}{\rho^{(k)} + (\rho^{(k-1)})^2} \tag{18}$$

$$\tilde{x}^{(k)} \leftarrow x^{(k)} + \gamma^{(k)}\left(x^{(k)} - x^{(k-1)}\right) \tag{19}$$

$$\delta_{\text{Nes}}^{(k+1)} \leftarrow \text{sign}\left(\nabla L(g(\tilde{x}^{(k)}), c)\right) \tag{20}$$

### B.2  Generalized-DLR loss

We generalize DLR loss by extending the denominator of DLR loss from $g_{\pi_1}(x) - g_{\pi_3}(x)$ to $g_{\pi_1}(x) - g_{\pi_q}(x)$. $\pi_q \in Y$ denotes the class label that has the $q$-th largest value of $g(x)$. More precisely, $L_{\text{G-DLR},q}$ is defined as

$$L_{\text{G-DLR},q}(g(x), c) = -\frac{g_c(x) - g_{\pi_2}(x)}{g_{\pi_1}(x) - g_{\pi_q}(x)}. \tag{21}$$

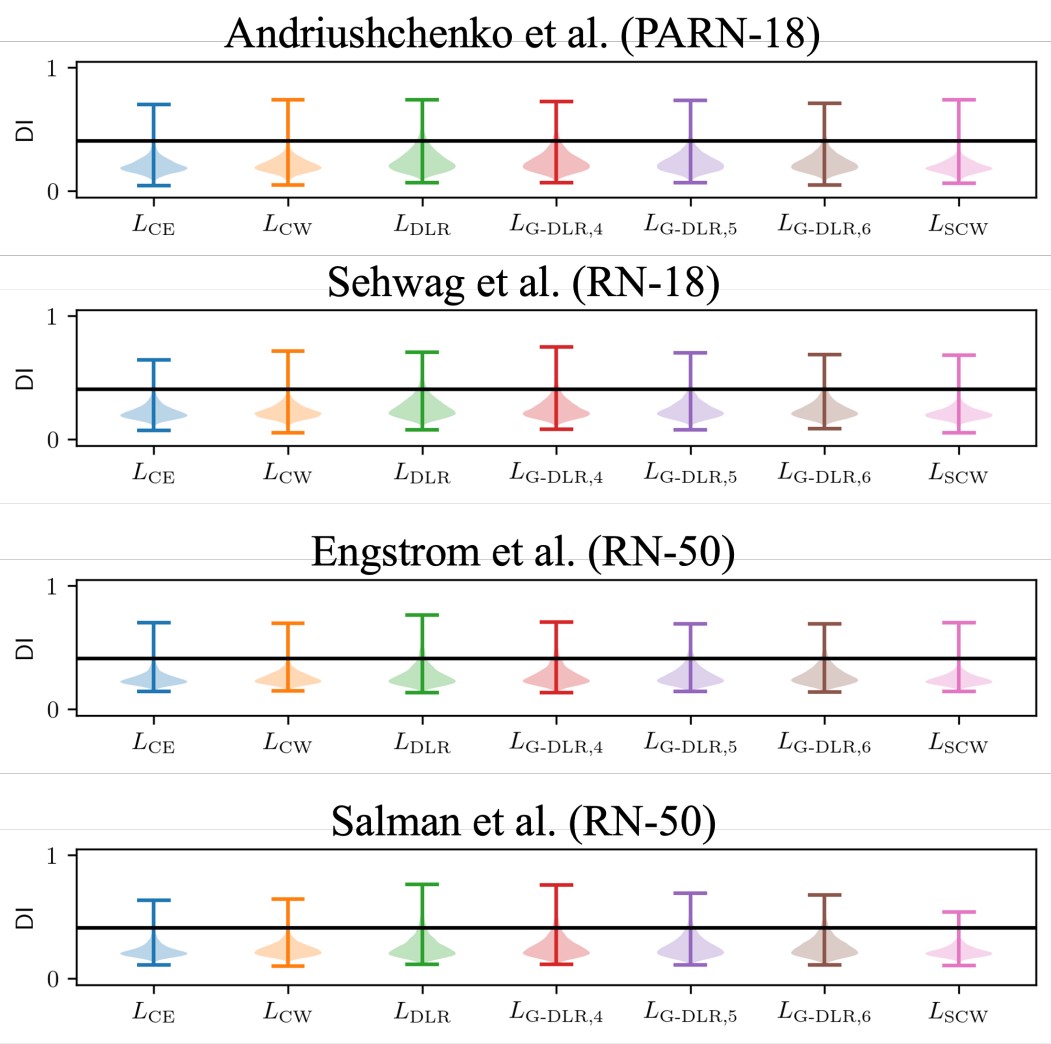

Figure 8: Violin plot of DI obtained by attacking the models proposed by (Sehwag et al., 2022; Andriushchenko & Flammarion, 2020) for CIFAR-10 and (Engstrom et al., 2019; Salman et al., 2020) for ImageNet.

## C  ADDITIONAL RESULTS OF THE ANALYSIS IN SECTION 3.1

### C.1  DIVERSITY INDEX FOR THE SET OF BEST SEARCH POINTS

The main paper only includes the results for the model proposed by Sehwag et al. (2022). This section describes the violin plot of DI for several models (Andriushchenko & Flammarion, 2020; Sehwag et al., 2022; Engstrom et al., 2019; Salman et al., 2020). Figures 8 and 9 show the violin plot of DI obtained by attacking the robust models. According to figs. 8 and 9, the best point sets obtained by attacks with a single search direction and objective function have similar DI value trends.

### C.2  VISUALIZATION OF THE BEST SEARCH POINTS VIA UMAP

This section describes the 2D visualization of the best search points using UMAP for several models (Andriushchenko & Flammarion, 2020; Sehwag et al., 2022; Engstrom et al., 2019; Salman et al., 2020). Figure 10 shows the 2D visualization of the best search points obtained by attacking the

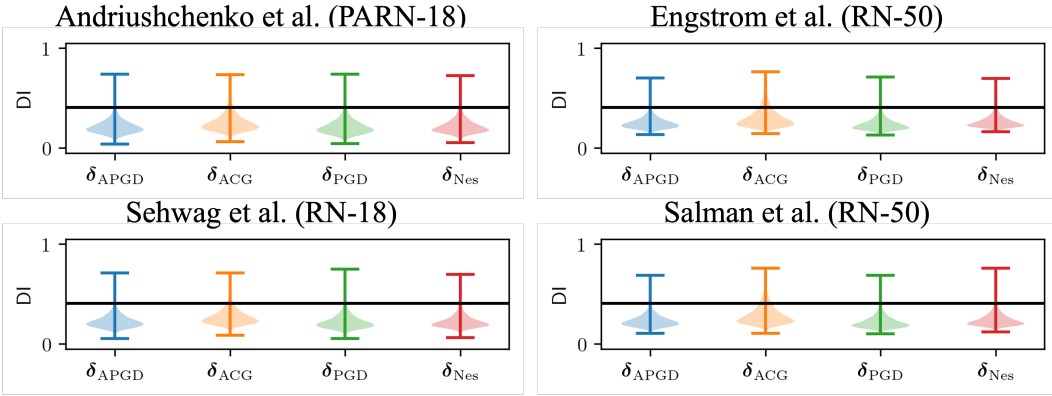

Figure 9: Violin plot of DI obtained by attacking the models proposed by (Sehwag et al., 2022; Andriushchenko & Flammarion, 2020) for CIFAR-10 and (Engstrom et al., 2019; Salman et al., 2020) for ImageNet.

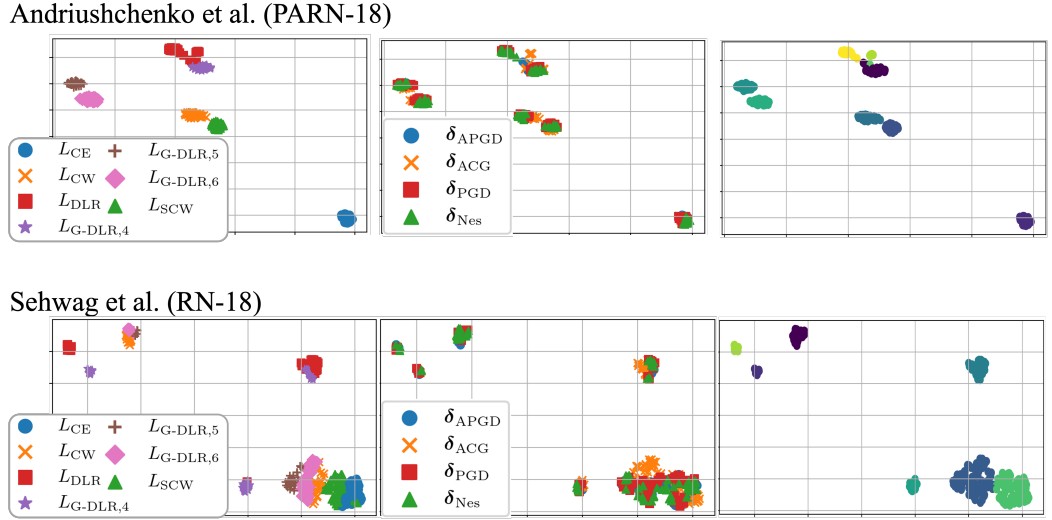

Figure 10: 2D visualization of the best search points obtained by attacking the models proposed by (Sehwag et al., 2022; Andriushchenko & Flammarion, 2020). The dataset is CIFAR-10. The same color in the left/center figure represents points obtained using the same objective function/search direction, respectively. The same color in the right figure shows the points determined by X-means to belong to the same cluster.

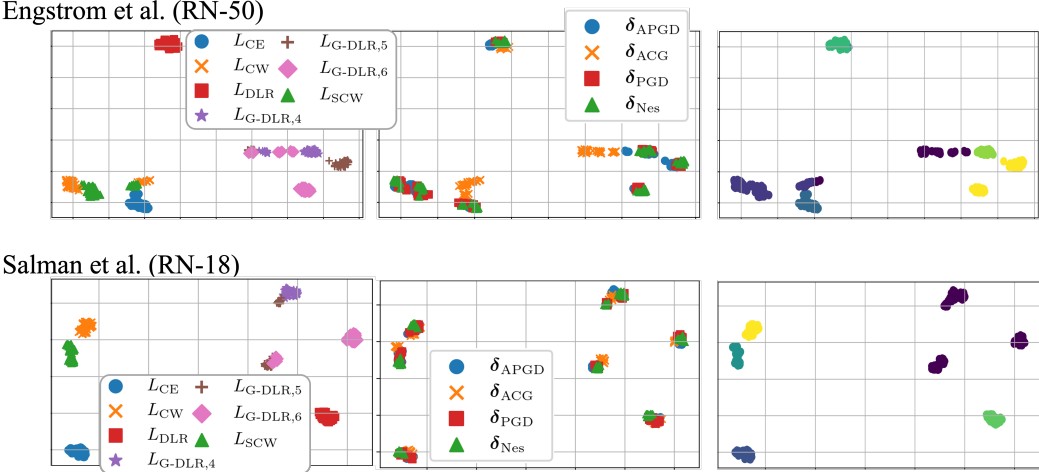

Figure 11: 2D visualization of the best search points obtained by attacking the models proposed by (Engstrom et al., 2019; Salman et al., 2020). The dataset is ImageNet. The same color in the left/center figure represents points obtained using the same objective function/search direction, respectively. The same color in the right figure shows the points determined by X-means to belong to the same cluster.

robust models trained on CIFAR-10. Figure 11 shows the 2D visualization of the best search points obtained by attacking the robust models trained on ImageNet. According to figs. 10 and 11, the best point sets obtained by attacks with different search directions and objective functions tend to form different clusters. The points determined to belong to the same cluster due to clustering using the X-means (Pelleg & Moore, 2000) are also plotted close together in the visualization using UMAP. These results suggest that 2D visualizations using UMAP are expected to reflect the actual distribution of search points.

### C.3 THE REASON WHY WE USED UMAP

The objective of the qualitative evaluation using UMAP is to know how the best points obtained by attacks using different objective functions/search directions are distributed and form different clusters. In order to achieve this goal, it is necessary to consider the distance between any two points and the distance between clusters. We have tried quantitative evaluation. However, we finally chose qualitative evaluation using UMAP because quantitative evaluations based on indicators such as objective values or DI are difficult to achieve our objective for the reasons described below. First, in adversarial attacks, distant points may show the same objective value or close points may show very different objective values because the adversarial attack is a maximization problem with many local optimums. Therefore, quantitative evaluation using objective values is considered difficult. In addition, DI cannot consider the distance between clusters because DI shows low values when a point set forms one or more clusters. Another possible evaluation method is clustering, such as k-means, but this is a qualitative evaluation as with UMAP. UMAP is a dimensionality reduction method that preserves the distance information in the original space as much as possible so it can reflect important information, such as the distance between any two points or clusters. Therefore, we think that the qualitative evaluation by UMAP provides convincing results.

### D COMPLETE RESULTS OF THE EXPERIMENTS

Tables 6 and 7 are the complete results of the experiments in section 4.1 described in table 1. Table 8 shows the complete results of the experiments in section 4.2 described in table 2. Also, table 9 shows the quantified degree of diversification of $A^3$, GS+LS(ADS), GS+LS(R-ADS), GS+LS(RAND), and EDA. Figure 12 shows the violin plot of DI for several models. Figure 13 shows the difference between $MT_{cos}$ and GS+LS in #queries to find AEs for some models.

Table 6: Comparisons with GS, LS, GS+LS, Naive, MT$_{cos}$, and AA in robust accuracy. The abbreviations are the same as those used in the main text.

| Defense | model | clean acc | AA | GS | LS | GS + LS | Naive | MT$_{cos}$ |
|---|---|---|---|---|---|---|---|---|
| CIFAR-10 | | | | | | | | |
| (Andriushchenko & Flammarion, 2020) | PARN-18 | 79.84 | 43.93 | 44.29 | 44.65 | 43.95 | 44.12 | 43.96 |
| (Addepalli et al., 2022) | RN-18 | 85.71 | 52.48 | 52.98 | 52.73 | 52.50 | 52.63 | 52.48 |
| (Sehwag et al., 2022) | RN-18 | 84.59 | 55.54 | 55.97 | 56.40 | 55.58 | 55.68 | 55.54 |
| (Engstrom et al., 2019) | RN-50 | 87.03 | 49.25 | 50.18 | 50.26 | 49.52 | 49.79 | 49.21 |
| (Carmon et al., 2019) | WRN-28-10 | 89.69 | 59.53 | 60.13 | 59.92 | 59.46 | 59.70 | 59.54 |
| (Gowal et al., 2020) | WRN-28-10 | 89.48 | 62.80 | 63.30 | 63.40 | 62.83 | 62.92 | 62.86 |
| (Hendrycks et al., 2019) | WRN-28-10 | 87.11 | 54.92 | 55.20 | 55.45 | 54.87 | 55.09 | 60.80 |
| (Rebuffi et al., 2021) | WRN-28-10 | 87.33 | 60.75 | 61.44 | 61.43 | 60.77 | 60.82 | 57.18 |
| (Sehwag et al., 2020) | WRN-28-10 | 88.98 | 57.14 | 57.78 | 57.53 | 57.14 | 57.43 | 59.67 |
| (Sridhar et al., 2022) | WRN-28-10 | 89.46 | 59.66 | 60.14 | 60.06 | 59.59 | 59.89 | 56.38 |
| (Wang et al., 2020) | WRN-28-10 | 87.50 | 56.29 | 56.89 | 57.24 | 56.33 | 56.66 | 60.06 |
| (Wu et al., 2020) | WRN-28-10 | 88.25 | 60.04 | 60.50 | 60.35 | 59.99 | 60.14 | 41.99 |
| (Ding et al., 2020) | WRN-28-4 | 84.36 | 41.44 | 43.81 | 48.44 | 43.24 | 43.49 | 57.77 |
| (Addepalli et al., 2022) | WRN-34-10 | 88.71 | 57.81 | 58.12 | 58.00 | 57.72 | 57.92 | 53.32 |
| (Sehwag et al., 2022) | WRN-34-10 | 86.68 | 60.27 | 60.74 | 60.97 | 60.21 | 60.52 | 60.31 |
| (Sitawarin et al., 2021) | WRN-34-10 | 86.84 | 50.72 | 51.36 | 51.42 | 50.70 | 50.98 | 50.75 |
| (Zhang et al., 2019a) | WRN-34-10 | 87.20 | 44.83 | 45.14 | 45.05 | 44.62 | 44.87 | 44.69 |
| (Zhang et al., 2020) | WRN-34-10 | 84.52 | 53.51 | 54.06 | 53.65 | 53.55 | 53.60 | 53.52 |
| (Sridhar et al., 2022) | WRN-34-15 | 86.53 | 60.41 | 60.81 | 60.50 | 60.39 | 60.46 | 60.43 |
| (Gowal et al., 2020) | WRN-34-20 | 85.64 | 56.86 | 57.11 | 57.05 | 56.88 | 56.90 | 56.83 |
| (Pang et al., 2020) | WRN-34-20 | 85.14 | 53.74 | 54.14 | 54.06 | 53.81 | 54.00 | 53.71 |
| (Rice et al., 2020) | WRN-34-20 | 85.34 | 53.42 | 53.73 | 54.30 | 53.42 | 53.52 | 53.39 |
| (Gowal et al., 2020) | WRN-70-16 | 85.29 | 57.20 | 57.41 | 57.55 | 57.18 | 57.27 | 57.15 |
| (Gowal et al., 2020) | WRN-70-16 | 91.10 | 65.88 | 66.42 | 66.61 | 65.85 | 66.04 | 65.96 |
| (Rebuffi et al., 2021) | WRN-70-16 | 88.54 | 64.25 | 65.08 | 64.81 | 64.32 | 64.53 | 64.28 |
| CIFAR-100 | | | | | | | | |
| (Rice et al., 2020) | PARN-18 | 53.83 | 18.95 | 19.30 | 19.67 | 18.97 | 19.08 | 18.99 |
| (Hendrycks et al., 2019) | WRN-28-10 | 59.23 | 28.42 | 28.67 | 29.40 | 28.44 | 28.61 | 28.43 |
| (Rebuffi et al., 2021) | WRN-28-10 | 62.41 | 32.06 | 32.49 | 32.98 | 32.08 | 32.22 | 32.07 |
| (Addepalli et al., 2022) | WRN-34-10 | 68.75 | 31.85 | 32.12 | 32.62 | 31.86 | 31.91 | 31.80 |
| (Cui et al., 2021) | WRN-34-10 | 60.64 | 29.33 | 29.45 | 29.12 | 28.99 | 29.20 | 28.99 |
| (Sitawarin et al., 2021) | WRN-34-10 | 62.82 | 24.57 | 24.94 | 25.27 | 24.65 | 24.74 | 24.55 |
| (Wu et al., 2020) | WRN-34-10 | 60.38 | 28.86 | 29.18 | 29.31 | 28.88 | 28.97 | 28.86 |
| (Cui et al., 2021) | WRN-34-20 | 62.55 | 30.20 | 30.39 | 30.22 | 30.01 | 30.15 | 30.03 |
| (Gowal et al., 2020) | WRN-70-16 | 60.86 | 30.03 | 30.21 | 30.93 | 30.05 | 30.11 | 30.00 |
| (Gowal et al., 2020) | WRN-70-16 | 69.15 | 36.88 | 37.46 | 38.04 | 36.96 | 37.19 | 36.95 |
| (Rebuffi et al., 2021) | WRN-70-16 | 63.56 | 34.64 | 35.04 | 35.38 | 34.65 | 34.88 | 34.68 |
| ImageNet | | | | | | | | |
| (Salman et al., 2020) | RN-18 | 52.92 | 25.32 | 25.66 | 25.56 | 25.22 | 25.46 | 25.24 |
| (Engstrom et al., 2019) | RN-50 | 62.56 | 29.22 | 30.42 | 30.14 | 29.20 | 29.64 | 29.34 |
| (Salman et al., 2020) | RN-50 | 64.02 | 34.96 | 35.36 | 35.26 | 34.84 | 35.00 | 34.68 |
| (Wong et al., 2020) | RN-50 | 55.62 | 26.24 | 27.58 | 28.10 | 26.22 | 26.84 | 26.40 |
| (Salman et al., 2020) | WRN-50-2 | 68.46 | 38.14 | 38.62 | 39.20 | 38.28 | 38.52 | 38.22 |

Table 7: Comparisons with GS+LS(ADS), GS+LS(R-ADS), and GS+LS(RAND) in robust accuracy to validate ADS. The abbreviations are the same as those used in the main text. The lowest robust accuracies are in bold.

| CIFAR-10 ($\varepsilon = 8/255$) | model | GS+LS (RAND) | GS+LS (R-ADS) | GS+LS (ADS) |
|---|---|---|---|---|
| (Andriushchenko & Flammarion, 2020) | PARN-18 | 44.09 | 44.02 | **43.95** |
| (Addepalli et al., 2022) | RN-18 | **52.45** | 52.83 | 52.50 |
| (Sehwag et al., 2022) | RN-18 | 55.61 | **55.58** | **55.58** |
| (Engstrom et al., 2019) | RN-50 | 49.60 | **49.40** | 49.52 |
| (Carmon et al., 2019) | WRN-28-10 | 59.56 | 59.53 | **59.46** |
| (Gowal et al., 2020) | WRN-28-10 | **62.82** | 62.99 | 62.83 |
| (Hendrycks et al., 2019) | WRN-28-10 | 54.94 | 55.02 | **54.87** |
| (Rebuffi et al., 2021) | WRN-28-10 | **60.67** | 60.84 | 60.77 |
| (Sehwag et al., 2020) | WRN-28-10 | 57.16 | 57.38 | **57.14** |
| (Sridhar et al., 2022) | WRN-28-10 | 59.63 | 59.76 | **59.59** |
| (Wang et al., 2020) | WRN-28-10 | **56.32** | 56.50 | 56.33 |
| (Wu et al., 2020) | WRN-28-10 | 60.02 | 60.13 | **59.99** |
| (Ding et al., 2020) | WRN-28-4 | 43.54 | 44.88 | **43.24** |
| (Addepalli et al., 2022) | WRN-34-10 | 57.75 | 57.87 | **57.72** |
| (Sehwag et al., 2022) | WRN-34-10 | 60.26 | 60.38 | **60.21** |
| (Sitawarin et al., 2021) | WRN-34-10 | 50.83 | 50.78 | **50.70** |
| (Zhang et al., 2019a) | WRN-34-10 | 44.78 | 44.70 | **44.62** |
| (Zhang et al., 2020) | WRN-34-10 | 53.51 | **53.50** | 53.55 |
| (Sridhar et al., 2022) | WRN-34-15 | 60.40 | 60.46 | **60.39** |
| (Gowal et al., 2020) | WRN-34-20 | 56.86 | **56.85** | 56.88 |
| (Pang et al., 2020) | WRN-34-20 | **53.77** | 53.85 | 53.81 |
| (Rice et al., 2020) | WRN-34-20 | 53.47 | 53.48 | **53.42** |
| (Gowal et al., 2020) | WRN-70-16 | 57.23 | 57.21 | **57.18** |
| (Gowal et al., 2020) | WRN-70-16 | 65.86 | 66.02 | **65.85** |
| (Rebuffi et al., 2021) | WRN-70-16 | **64.23** | 64.54 | 64.32 |

| CIFAR-100 ($\varepsilon = 8/255$) | model | RAND acc | R-ADS acc | ADS acc |
|---|---|---|---|---|
| (Rice et al., 2020) | PARN-18 | 18.98 | 18.99 | **18.97** |
| (Hendrycks et al., 2019) | WRN-28-10 | 28.56 | 28.83 | **28.44** |
| (Rebuffi et al., 2021) | WRN-28-10 | **32.08** | 32.13 | **32.08** |
| (Addepalli et al., 2022) | WRN-34-10 | 31.91 | 32.23 | **31.86** |
| (Cui et al., 2021) | WRN-34-10 | **28.97** | 29.20 | 28.99 |
| (Sitawarin et al., 2021) | WRN-34-10 | 24.71 | 24.68 | **24.65** |
| (Wu et al., 2020) | WRN-34-10 | 28.93 | 29.46 | **28.88** |
| (Cui et al., 2021) | WRN-34-20 | 30.07 | 30.35 | **30.01** |
| (Gowal et al., 2020) | WRN-70-16 | 30.06 | 30.42 | **30.05** |
| (Gowal et al., 2020) | WRN-70-16 | 37.05 | 37.53 | **36.96** |
| (Rebuffi et al., 2021) | WRN-70-16 | **34.61** | 34.97 | 34.65 |

| ImageNet ($\varepsilon = 4/255$) | model | RAND acc | R-ADS acc | ADS acc |
|---|---|---|---|---|
| (Salman et al., 2020) | RN-18 | **25.22** | 25.44 | **25.22** |
| (Engstrom et al., 2019) | RN-50 | 29.56 | 29.26 | **29.20** |
| (Salman et al., 2020) | RN-50 | **34.68** | **34.68** | 34.84 |
| (Wong et al., 2020) | RN-50 | 26.26 | 26.36 | **26.22** |
| (Salman et al., 2020) | WRN-50-2 | 38.38 | 38.54 | **38.28** |

Table 8: Average robust accuracy and computation time over five runs. The lowest accuracies are in bold. RN: ResNet, WRN: WideResNet, PARN: PreActResNet, $\Delta$: $A^3$-EDA. "EDA/$A^3$" column is the same as the "ratio" column in table 2. We report the computation time in seconds.

| CIFAR-10 | model | clean | AA | $A^3$ | | EDA | | $\Delta$ | | EDA/$A^3$ |
|---|---|---|---|---|---|---|---|---|---|---|
| ($\varepsilon = 8/255$) | | acc | acc | acc | time (sec) | acc | time (sec) | acc | time | time |
| Andriushchenko & Flammarion (2020) | PARN-18 | 79.84 | 43.93 | 43.96±0.00 | 382±2 | **43.85±0.02** | 513±32 | 0.11 | -130 | 1.34 |
| Addepalli et al. (2022) | RN-18 | 85.71 | 52.48 | 52.46±0.02 | 434±2 | **52.43±0.03** | 625±25 | 0.03 | -193 | 1.45 |
| Sehwag et al. (2022) | RN-18 | 84.59 | 55.54 | 55.53±0.01 | 1,121±68 | **55.49±0.01** | 589±36 | 0.04 | 514 | 0.53 |
| Engstrom et al. (2019) | RN-50 | 87.03 | 49.25 | 49.25±0.02 | 1,572±11 | **49.10±0.03** | 1,485±78 | 0.14 | 89 | 0.94 |
| Carmon et al. (2019) | WRN-28-10 | 89.69 | 59.53 | 59.44±0.01 | 4,223±4 | **59.40±0.01** | 3,316±65 | 0.03 | 903 | 0.79 |
| Gowal et al. (2020) | WRN-28-10 | 89.48 | 62.80 | 62.77±0.01 | 3,841±13 | **62.75±0.02** | 4,557±112 | 0.02 | -718 | 1.19 |
| Hendrycks et al. (2019) | WRN-28-10 | 87.11 | 54.92 | 54.85±0.01 | 2,719±50 | **54.77±0.02** | 3,121±45 | 0.08 | -354 | 1.13 |
| Rebuffi et al. (2021) | WRN-28-10 | 87.33 | 60.75 | 60.72±0.01 | 3,928±30 | **60.64±0.01** | 4,459±74 | 0.08 | -521 | 1.13 |
| Sehwag et al. (2020) | WRN-28-10 | 88.98 | 57.14 | 57.14±0.02 | 2,662±50 | **57.03±0.01** | 3,255±64 | 0.11 | -583 | 1.22 |
| Sridhar et al. (2022) | WRN-28-10 | 89.46 | 59.66 | 59.56±0.01 | 3,245±119 | **59.46±0.02** | 3,355±55 | 0.10 | -140 | 1.04 |
| Wang et al. (2020) | WRN-28-10 | 87.50 | 56.29 | 56.28±0.01 | 2,732±4 | **56.15±0.02** | 3,285±78 | 0.12 | -555 | 1.20 |
| Wu et al. (2020) | WRN-28-10 | 88.25 | 60.04 | 60.02±0.01 | 3,273±8 | **59.94±0.01** | 3,502±39 | 0.08 | -222 | 1.07 |
| Ding et al. (2020) | WRN-28-4 | 84.36 | 41.44 | **41.24±0.06** | 2,017±90 | 41.74±0.06 | 695±29 | -0.50 | 1,624 | 0.30 |
| Addepalli et al. (2022) | WRN-34-10 | 88.71 | 57.81 | 57.73±0.01 | 3,926±7 | **57.69±0.02** | 4,225±109 | 0.04 | -295 | 1.08 |
| Sehwag et al. (2022) | WRN-34-10 | 86.68 | 60.27 | 60.22±0.01 | 3,858±6 | **60.18±0.02** | 4,172±18 | 0.04 | -311 | 1.08 |
| Sitawarin et al. (2021) | WRN-34-10 | 86.84 | 50.72 | 50.69±0.02 | 3,845±22 | **50.59±0.01** | 3,591±103 | 0.10 | 252 | 0.93 |
| Zhang et al. (2019a) | WRN-34-10 | 87.20 | 44.83 | 44.63±0.03 | 3,500±12 | **44.51±0.02** | 3,219±32 | 0.12 | 280 | 0.92 |
| Zhang et al. (2020) | WRN-34-10 | 84.52 | 53.51 | 53.46±0.01 | 3,912±16 | **53.42±0.02** | 3,936±134 | 0.03 | -22 | 1.01 |
| Sridhar et al. (2022) | WRN-34-15 | 86.53 | 60.41 | 60.38±0.01 | 6,805±14 | **60.32±0.01** | 7,785±116 | 0.06 | -976 | 1.14 |
| Gowal et al. (2020) | WRN-34-20 | 85.64 | 56.86 | 56.81±0.01 | 13,463±38 | **56.79±0.03** | 14,693±284 | 0.02 | -1,251 | 1.09 |
| Pang et al. (2020) | WRN-34-20 | 85.14 | 53.74 | 53.69±0.01 | 12,436±21 | **53.66±0.01** | 18,775±241 | 0.03 | -6,392 | 1.52 |
| Rice et al. (2020) | WRN-34-20 | 85.34 | 53.42 | 53.38±0.01 | 12,290±5 | **53.34±0.01** | 11,255±377 | 0.04 | 991 | 0.92 |
| Gowal et al. (2020) | WRN-70-16 | 85.29 | 57.20 | **57.11±0.01** | 26,587±1,032 | 57.12±0.01 | 21,790±430 | -0.01 | 4,747 | 0.82 |
| Gowal et al. (2020) | WRN-70-16 | 91.10 | 65.88 | 65.85±0.01 | 29,544±252 | **65.83±0.01** | 24,885±371 | 0.02 | 4,468 | 0.85 |
| Rebuffi et al. (2021) | WRN-70-16 | 88.54 | 64.25 | 64.24±0.01 | 29,075±887 | **64.20±0.03** | 24,652±479 | 0.04 | 4,070 | 0.86 |
| CIFAR-100 | model | clean | AA | $A^3$ | | EDA | | $\Delta$ | | EDA/$A^3$ |
| ($\varepsilon = 8/255$) | | acc | acc | acc | time (sec) | acc | time (sec) | acc | time | time |
| Rice et al. (2020) | PARN-18 | 53.83 | 18.95 | 18.89±0.00 | 1,531±924 | **18.88±0.01** | 497±74 | 0.01 | 1,237 | 0.29 |
| Hendrycks et al. (2019) | WRN-28-10 | 59.23 | 28.42 | 28.32±0.02 | 2,684±10 | **28.27±0.02** | 1,981±38 | 0.04 | 670 | 0.75 |
| Rebuffi et al. (2021) | WRN-28-10 | 62.41 | 32.06 | 32.00±0.02 | 3,044±8 | **31.94±0.03** | 2,701±86 | 0.06 | 331 | 0.89 |
| Addepalli et al. (2022) | WRN-34-10 | 68.75 | 31.85 | 31.81±0.02 | 3,046±17 | **31.78±0.01** | 2,792±95 | 0.03 | 266 | 0.91 |
| Cui et al. (2021) | WRN-34-10 | 60.64 | 29.33 | 28.84±0.02 | 3,002±7 | **28.83±0.02** | 3,075±27 | 0.01 | -74 | 1.02 |
| Sitawarin et al. (2021) | WRN-34-10 | 62.82 | 24.57 | 24.56±0.03 | 4,935±137 | **24.50±0.01** | 1,985±40 | 0.07 | 2,940 | 0.40 |
| Wu et al. (2020) | WRN-34-10 | 60.38 | 28.86 | 28.79±0.02 | 3,258±58 | **28.76±0.01** | 2,360±31 | 0.03 | 902 | 0.72 |
| Cui et al. (2021) | WRN-34-20 | 62.55 | 30.20 | **29.84±0.01** | 9,798±8 | 29.85±0.01 | 8,027±197 | -0.01 | 1,751 | 0.82 |
| Gowal et al. (2020) | WRN-70-16 | 60.86 | 30.03 | 29.97±0.01 | 21,452±701 | **29.96±0.01** | 13,060±348 | 0.01 | 8,535 | 0.60 |
| Gowal et al. (2020) | WRN-70-16 | 69.15 | 36.88 | 36.87±0.00 | 22,423±1,969 | **36.81±0.01** | 15,641±467 | 0.06 | 7,100 | 0.69 |
| Rebuffi et al. (2021) | WRN-70-16 | 63.56 | 34.64 | 34.62±0.01 | 21,546±190 | **34.55±0.01** | 15,474±404 | 0.08 | 6,403 | 0.71 |
| ImageNet | model | clean | AA | $A^3$ | | EDA | | $\Delta$ | | EDA/$A^3$ |
| ($\varepsilon = 4/255$) | | acc | acc | acc | time (sec) | acc | time (sec) | acc | time | time |
| Salman et al. (2020) | RN-18 | 52.92 | 25.32 | 25.22±0.03 | 2,937±10 | **25.11±0.02** | 1,667±119 | 0.10 | 1,276 | 0.57 |
| Engstrom et al. (2019) | RN-50 | 62.56 | 29.22 | 29.32±0.01 | 9,380±188 | **29.01±0.01** | 3,159±136 | 0.30 | 6,194 | 0.34 |
| Salman et al. (2020) | RN-50 | 64.02 | 34.96 | 34.75±0.04 | 9,989±234 | **34.52±0.02** | 3,525±302 | 0.22 | 6,502 | 0.35 |
| Wong et al. (2020) | RN-50 | 55.62 | 26.24 | 26.42±0.04 | 8,472±194 | **26.12±0.10** | 4,459±110 | 0.31 | 4,278 | 0.51 |
| Salman et al. (2020) | WRN-50-2 | 68.46 | 38.14 | 38.26±0.02 | 9,886±120 | **38.03±0.02** | 5,102±119 | 0.23 | 4,811 | 0.51 |

Table 9: The quantified degree of diversification. DI denotes the Diversity Index, and E denotes the metric defined by equation 22. RAND, R-ADS, and ADS represent GS+LS(RAND), GS+LS(R-ADS), and GS+LS(ADS), respectively.

| CIFAR-10 ($\varepsilon = 8/255$) | Models | $A^3$ DI | $A^3$ E | RAND DI | RAND E | R-ADS DI | R-ADS E | ADS DI | ADS E | EDA DI | EDA E |
|---|---|---|---|---|---|---|---|---|---|---|---|
| Andriushchenko & Flammarion (2020) | PARN-18 | 0.26±0.09 | 0.66±0.27 | 0.30±0.06 | 0.95±0.20 | 0.25±0.05 | 0.73±0.14 | 0.33±0.07 | 0.98±0.18 | 0.36±0.05 | 1.11±0.15 |
| Addepalli et al. (2022) | RN-18 | 0.27±0.10 | 0.69±0.28 | 0.31±0.07 | 0.94±0.20 | 0.23±0.05 | 0.78±0.11 | 0.36±0.08 | 1.02±0.19 | 0.40±0.06 | 1.18±0.17 |
| Sehwag et al. (2022) | RN-18 | 0.27±0.09 | 0.72±0.28 | 0.31±0.06 | 0.93±0.20 | 0.26±0.05 | 0.79±0.15 | 0.34±0.06 | 0.99±0.19 | 0.38±0.05 | 1.15±0.15 |
| Engstrom et al. (2019) | RN-50 | 0.28±0.09 | 0.72±0.27 | 0.33±0.06 | 0.99±0.19 | 0.25±0.05 | 0.76±0.15 | 0.37±0.07 | 1.04±0.19 | 0.39±0.05 | 1.17±0.16 |
| Carmon et al. (2019) | WRN-28-10 | 0.26±0.08 | 0.65±0.25 | 0.33±0.06 | 1.00±0.18 | 0.26±0.05 | 0.84±0.14 | 0.38±0.07 | 1.06±0.19 | 0.41±0.05 | 1.22±0.17 |
| Gowal et al. (2020) | WRN-28-10 | 0.22±0.10 | 0.58±0.30 | 0.30±0.07 | 0.94±0.21 | 0.20±0.05 | 0.71±0.11 | 0.37±0.08 | 1.04±0.22 | 0.41±0.06 | 1.20±0.18 |
| Hendrycks et al. (2019) | WRN-28-10 | 0.25±0.09 | 0.64±0.27 | 0.30±0.06 | 0.94±0.20 | 0.22±0.04 | 0.71±0.12 | 0.34±0.07 | 1.01±0.19 | 0.37±0.05 | 1.14±0.15 |
| Rebuffi et al. (2021) | WRN-28-10 | 0.24±0.10 | 0.63±0.30 | 0.30±0.06 | 0.89±0.21 | 0.24±0.05 | 0.75±0.12 | 0.34±0.07 | 0.97±0.18 | 0.37±0.05 | 1.09±0.16 |
| Sehwag et al. (2020) | WRN-28-10 | 0.25±0.08 | 0.64±0.26 | 0.33±0.06 | 0.99±0.19 | 0.23±0.04 | 0.75±0.11 | 0.38±0.07 | 1.09±0.20 | 0.43±0.05 | 1.24±0.19 |
| Sridhar et al. (2022) | WRN-28-10 | 0.25±0.09 | 0.64±0.26 | 0.33±0.06 | 1.00±0.18 | 0.24±0.04 | 0.76±0.10 | 0.37±0.07 | 1.05±0.18 | 0.41±0.05 | 1.24±0.16 |
| Wang et al. (2020) | WRN-28-10 | 0.28±0.08 | 0.68±0.24 | 0.31±0.06 | 0.93±0.18 | 0.25±0.06 | 0.79±0.12 | 0.35±0.07 | 0.99±0.18 | 0.35±0.06 | 1.04±0.15 |
| Wu et al. (2020) | WRN-28-10 | 0.25±0.09 | 0.64±0.27 | 0.32±0.06 | 0.98±0.19 | 0.23±0.04 | 0.76±0.10 | 0.38±0.07 | 1.08±0.19 | 0.43±0.06 | 1.25±0.16 |
| Ding et al. (2020) | WRN-28-4 | 0.24±0.10 | 0.91±0.34 | 0.33±0.07 | 0.98±0.22 | 0.29±0.07 | 0.91±0.23 | 0.36±0.08 | 1.02±0.22 | 0.38±0.07 | 1.19±0.16 |
| Addepalli et al. (2022) | WRN-34-10 | 0.27±0.09 | 0.67±0.28 | 0.31±0.06 | 0.94±0.20 | 0.24±0.05 | 0.83±0.13 | 0.37±0.08 | 1.04±0.20 | 0.41±0.06 | 1.21±0.17 |
| Sehwag et al. (2022) | WRN-34-10 | 0.25±0.09 | 0.65±0.26 | 0.32±0.06 | 0.96±0.18 | 0.24±0.04 | 0.74±0.12 | 0.37±0.07 | 1.02±0.18 | 0.38±0.05 | 1.15±0.14 |
| Sitawarin et al. (2021) | WRN-34-10 | 0.27±0.10 | 0.72±0.30 | 0.32±0.06 | 0.99±0.20 | 0.24±0.05 | 0.74±0.17 | 0.33±0.06 | 1.03±0.20 | 0.38±0.05 | 1.18±0.16 |
| Zhang et al. (2019a) | WRN-34-10 | 0.27±0.11 | 0.73±0.31 | 0.30±0.06 | 0.96±0.21 | 0.23±0.05 | 0.70±0.16 | 0.32±0.07 | 1.00±0.21 | 0.37±0.05 | 1.18±0.17 |
| Zhang et al. (2020) | WRN-34-10 | 0.25±0.09 | 0.62±0.26 | 0.31±0.06 | 0.95±0.20 | 0.25±0.05 | 0.82±0.17 | 0.33±0.07 | 0.99±0.20 | 0.39±0.06 | 1.21±0.16 |
| Sridhar et al. (2022) | WRN-34-15 | 0.23±0.08 | 0.57±0.24 | 0.31±0.07 | 0.96±0.19 | 0.24±0.05 | 0.78±0.11 | 0.37±0.08 | 1.06±0.19 | 0.41±0.06 | 1.21±0.16 |
| Gowal et al. (2020) | WRN-34-20 | 0.21±0.11 | 0.55±0.32 | 0.29±0.07 | 0.95±0.22 | 0.21±0.05 | 0.73±0.15 | 0.32±0.08 | 1.01±0.21 | 0.38±0.05 | 1.19±0.18 |
| Pang et al. (2020) | WRN-34-20 | 0.24±0.09 | 0.66±0.30 | 0.22±0.08 | 0.80±0.23 | 0.16±0.07 | 0.64±0.25 | 0.25±0.10 | 0.87±0.21 | 0.31±0.08 | 1.06±0.19 |
| Rice et al. (2020) | WRN-34-20 | 0.24±0.11 | 0.64±0.32 | 0.28±0.06 | 0.91±0.22 | 0.22±0.05 | 0.71±0.17 | 0.33±0.07 | 0.98±0.21 | 0.37±0.05 | 1.14±0.17 |
| Gowal et al. (2020) | WRN-70-16 | 0.22±0.10 | 0.56±0.30 | 0.30±0.06 | 0.94±0.20 | 0.21±0.05 | 0.71±0.12 | 0.38±0.08 | 1.04±0.20 | 0.39±0.05 | 1.18±0.17 |
| Gowal et al. (2020) | WRN-70-16 | 0.19±0.10 | 0.52±0.32 | 0.30±0.06 | 0.94±0.22 | 0.22±0.05 | 0.76±0.15 | 0.31±0.07 | 1.01±0.21 | 0.38±0.05 | 1.19±0.16 |
| Rebuffi et al. (2021) | WRN-70-16 | 0.23±0.09 | 0.59±0.29 | 0.30±0.06 | 0.88±0.20 | 0.23±0.05 | 0.74±0.12 | 0.34±0.07 | 0.96±0.17 | 0.37±0.05 | 1.09±0.16 |

| CIFAR-100 ($\varepsilon = 8/255$) | Models | $A^3$ DI | $A^3$ E | RAND DI | RAND E | R-ADS DI | R-ADS E | ADS DI | ADS E | EDA DI | EDA E |
|---|---|---|---|---|---|---|---|---|---|---|---|
| Rice et al. (2020) | PARN-18 | 0.34±0.13 | 0.83±0.31 | 0.32±0.06 | 0.97±0.20 | 0.25±0.06 | 0.78±0.18 | 0.35±0.07 | 1.02±0.21 | 0.39±0.06 | 1.15±0.20 |
| Hendrycks et al. (2019) | WRN-28-10 | 0.27±0.11 | 0.71±0.30 | 0.29±0.07 | 0.94±0.22 | 0.22±0.06 | 0.75±0.16 | 0.31±0.08 | 0.97±0.21 | 0.36±0.06 | 1.13±0.18 |
| Rebuffi et al. (2021) | WRN-28-10 | 0.32±0.15 | 0.81±0.36 | 0.29±0.07 | 0.94±0.23 | 0.25±0.06 | 0.82±0.17 | 0.35±0.07 | 1.04±0.21 | 0.41±0.06 | 1.20±0.20 |
| Addepalli et al. (2022) | WRN-34-10 | 0.34±0.13 | 0.84±0.30 | 0.31±0.07 | 0.97±0.21 | 0.24±0.06 | 0.86±0.13 | 0.36±0.09 | 1.04±0.21 | 0.42±0.06 | 1.21±0.18 |
| Cui et al. (2021) | WRN-34-10 | 0.27±0.10 | 0.67±0.27 | 0.27±0.08 | 0.89±0.22 | 0.23±0.08 | 0.82±0.14 | 0.30±0.09 | 0.97±0.19 | 0.34±0.07 | 1.04±0.17 |
| Sitawarin et al. (2021) | WRN-34-10 | 0.31±0.12 | 0.79±0.30 | 0.32±0.06 | 0.98±0.20 | 0.24±0.05 | 0.75±0.17 | 0.35±0.07 | 1.02±0.21 | 0.38±0.06 | 1.14±0.20 |
| Wu et al. (2020) | WRN-34-10 | 0.28±0.12 | 0.73±0.32 | 0.29±0.07 | 0.95±0.22 | 0.24±0.05 | 0.79±0.15 | 0.32±0.08 | 0.99±0.21 | 0.37±0.06 | 1.16±0.18 |
| Cui et al. (2021) | WRN-34-20 | 0.27±0.09 | 0.66±0.27 | 0.29±0.05 | 0.92±0.21 | 0.22±0.06 | 0.75±0.13 | 0.32±0.09 | 0.99±0.21 | 0.35±0.07 | 1.07±0.18 |
| Gowal et al. (2020) | WRN-70-16 | 0.27±0.13 | 0.69±0.34 | 0.28±0.07 | 0.93±0.22 | 0.19±0.05 | 0.66±0.13 | 0.33±0.08 | 0.98±0.23 | 0.37±0.06 | 1.19±0.17 |
| Gowal et al. (2020) | WRN-70-16 | 0.31±0.14 | 0.79±0.33 | 0.31±0.07 | 0.94±0.22 | 0.23±0.06 | 0.73±0.14 | 0.33±0.08 | 1.01±0.21 | 0.40±0.06 | 1.21±0.17 |
| Rebuffi et al. (2021) | WRN-70-16 | 0.29±0.14 | 0.76±0.36 | 0.30±0.07 | 0.93±0.23 | 0.21±0.06 | 0.74±0.14 | 0.34±0.08 | 0.99±0.20 | 0.40±0.05 | 1.19±0.19 |

| ImageNet ($\varepsilon = 4/255$) | Models | $A^3$ DI | $A^3$ E | RAND DI | RAND E | R-ADS DI | R-ADS E | ADS DI | ADS E | EDA DI | EDA E |
|---|---|---|---|---|---|---|---|---|---|---|---|
| Salman et al. (2020) | RN-18 | 0.26±0.07 | 2.60±0.91 | 0.34±0.07 | 3.52±0.69 | 0.28±0.05 | 2.83±0.42 | 0.38±0.08 | 3.66±0.69 | 0.43±0.04 | 4.37±0.66 |
| Engstrom et al. (2019) | RN-50 | 0.29±0.06 | 2.79±0.83 | 0.38±0.05 | 3.69±0.58 | 0.31±0.06 | 3.18±0.51 | 0.40±0.06 | 3.88±0.61 | 0.44±0.04 | 4.30±0.64 |
| Salman et al. (2020) | RN-50 | 0.26±0.06 | 2.61±0.90 | 0.35±0.06 | 3.61±0.65 | 0.30±0.06 | 3.03±0.47 | 0.37±0.07 | 3.70±0.63 | 0.42±0.04 | 4.43±0.53 |
| Wong et al. (2020) | RN-50 | 0.27±0.07 | 3.45±1.12 | 0.36±0.06 | 4.59±0.82 | 0.30±0.06 | 3.76±0.58 | 0.38±0.07 | 4.81±0.80 | 0.43±0.05 | 5.43±0.89 |
| Salman et al. (2020) | WRN-50-2 | 0.27±0.06 | 2.63±0.82 | 0.37±0.06 | 3.69±0.60 | 0.34±0.06 | 3.53±0.58 | 0.38±0.05 | 3.80±0.61 | 0.43±0.04 | 4.22±0.65 |

Table 10: Results of the preliminary experiments to determine the hyperparameters of ADS. The robust accuracy obtained by GS+LS is described. The default parameters are in bold.

| Dataset | No. | $N_{\text{ADS}}$ | | | | $n_a$ | | | |
|---|---|---|---|---|---|---|---|---|---|
| | | 3 | **4** | 5 | 10 | 3 | 4 | **5** | 6 |
| CIFAR-10 | 1 | 55.56 | 55.58 | 55.65 | 55.67 | 55.63 | 55.66 | 55.58 | 55.57 |
| CIFAR-10 | 2 | 56.83 | 56.80 | 56.84 | 56.81 | 56.90 | 56.83 | 56.80 | 56.78 |
| CIFAR-100 | 3 | 19.01 | 18.87 | 19.03 | 19.01 | 19.16 | 19.10 | 18.87 | 18.90 |
| CIFAR-100 | 4 | 24.66 | 24.56 | 24.61 | 24.59 | 24.61 | 24.78 | 24.56 | 24.56 |
| ImageNet | 5 | 29.40 | 29.24 | 29.26 | 29.30 | 29.28 | 29.34 | 29.24 | 29.14 |

### D.1 COMPUTER SPECIFICATION

The experiments are conducted with two types of CPUs and a single type of GPU. The CPUs used in the experiments are Intel(R) Xeon(R) Gold 6240R CPU @ 2.40GHz and Intel(R) Xeon(R) Silver 4216 CPU @ 2.10GHz. The GPU used in the experiments is NVIDIA GeForce RTX 3090. When we compare the performance of each attack on model A and dataset B, all compared attacks are run on the same device. Therefore, although we use different computers, the runtime comparison is fair.

### D.2 HYPERPARAMETER DETERMINATION

The values of $n_a$, $N_{\text{ADS}}$, and $N_s$ are determined based on the following preliminary experiments, as they should be as small as possible regarding computational cost. The initial step size is determined based on the step size rules of APGD, a powerful heuristic. The step size of $\eta = 2\varepsilon$ allows the initial search to move from one end of the feasible region to the other, thus allowing a broader search. The parameters $0.22$ and $0.19$, which determine the allocation of the number of iterations for GS and LS, are inspired by the checkpoints in the APGD's step size update. $N_1$ and $N_2$ are the number of iterations to be searched with a step size of $2\varepsilon$ and $\varepsilon$, respectively, in APGD with the total number of iterations set to 100. The experiments in Yamamura et al. (2022) suggest that the CG diversification performance is well achieved by moving in the CG direction according to this iteration allocation and step size assignment. Therefore, we chose these values for step sizes and $N_1 \sim N_3$.

**Preliminary experiments to determine hyperparameters of ADS** We conducted preliminary experiments on the following five models to determine the hyperparameters of ADS. 1. ResNet-18 (Sehwag et al., 2022), 2. WideResNet-28-10 (Gowal et al., 2020), 3. PreActResNet-18 (Rice et al., 2020), 4. WideResNet-34-10 (Sitawarin et al., 2021), 5. ResNet-50 (Engstrom et al., 2019). These numbers correspond to the "No." column in table 10.

### D.3 COMPARISON IN COMPUTATION COST BASED ON THE NUMBER OF QUERIES

Since the bottleneck in an adversarial attack is forward/backward (queries), we compare the number of queries. In attack selection, CAA requires $KNt\times$# samples $= 60t\times$ #samples $\geq 60\times$ #samples, where $K = 20$ is the population size, $N = 3$ is the policy length, and $t \geq 1$ is the number of iterations for the candidate attacks. Also, #samples = 4000 for CIFAR-10 and 1000 for ImageNet. Therefore, attack selection in CAA requires more than 240000 queries for CIFAR-10 and 60000 for ImageNet. For attack selection by ADS, $2|A|N_{ADS}\times$ #samples $= 112\times$ #samples queries are required, where $|A| = 28$ is the number of candidates and $N_{ADS} = 4$ is the number of iterations for candidates. Also, #samples=100 for CIFAR-10/100 and 50 for ImageNet. Therefore, ADS requires 11200 queries for CIFAR-10 and 5600 queries for ImageNet. Also, the standard AA requires queries of $6100\times$ #images. CAA requires at least $60t\times$#samples queries. EDA requires $n_a \times (N_1 + N_2 + N_3)\times$# images$= 5 \times 100\times$#images queries for GS+LS, and $K \times N_s + N_4\times$# images$= 190 \sim 300\times$#images queries for the targeted attack $a^t$. Therefore, EDA requires $692.24 \sim 802.24\times$# images queries in total. We compared the runtime of EDA with that of $A^3$ because $A^3$ automatically terminates its search before the query limit.

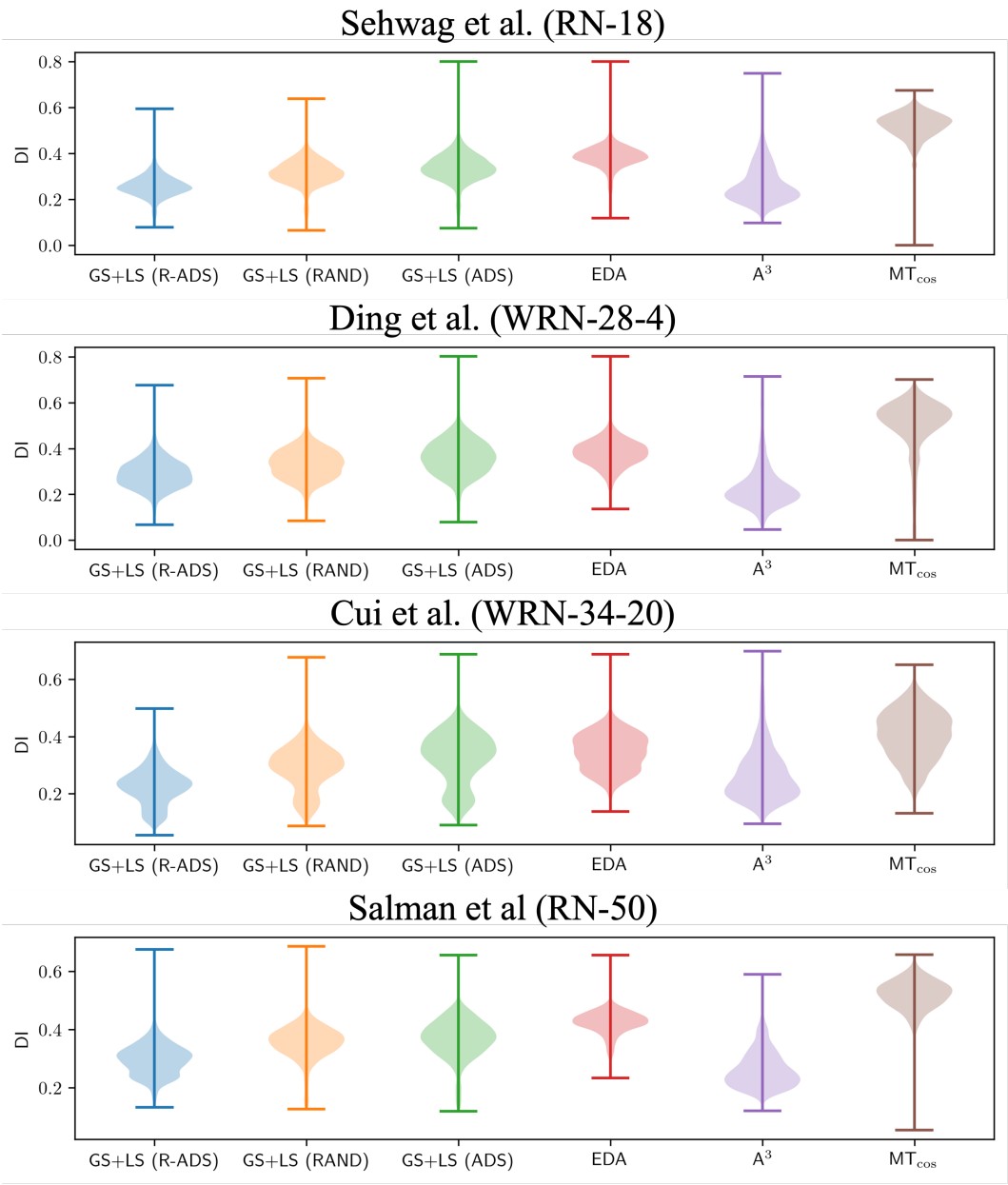

Figure 12: Violin plot of DI. The attacked models are Sehwag et al. (2022) and Ding et al. (2020) for CIFAR-10, Cui et al. (2021) for CIFAR-100, and Salman et al. (2020) for ImageNet.

## D.4 MATHEMATICAL DEFINITION OF THE BEST POINT SETS OF $A^3$ AND EDA

Mathematically, the best point sets of $A^3$ and EDA are defined as follows. First, the best point set of $A^3$ is defined as $X^*(\boldsymbol{x}_i, a_{A^3}, R_i)$, where $a_{A^3} = (\phi_{\text{ADI}}, \psi_{\cos}, \boldsymbol{\delta}_{\text{GD}}, L_{\text{CW}})$, where $\phi_{\text{ADI}}$ is Adaptive Direction Initialization (ADI) proposed by Liu et al. (2022c). Subsequently, the best point set of EDA is defined as $X^*(\boldsymbol{x}_i, e^*) \cup X^*(\boldsymbol{x}_i, a^t, 1)$, where $e^* = \{(\phi, \psi, \boldsymbol{\delta}_a, L_a) \mid a \in \{a_1^* \ldots, a_{n_a}^*, a_1^{**}, \ldots, a_{n_a}^{**}\}\}$ and $a^t = (\phi_{\text{PAS}}, \psi_{\text{APGD}}, \boldsymbol{\delta}_{\text{GD}}, L_{\text{CW}}^T)$.

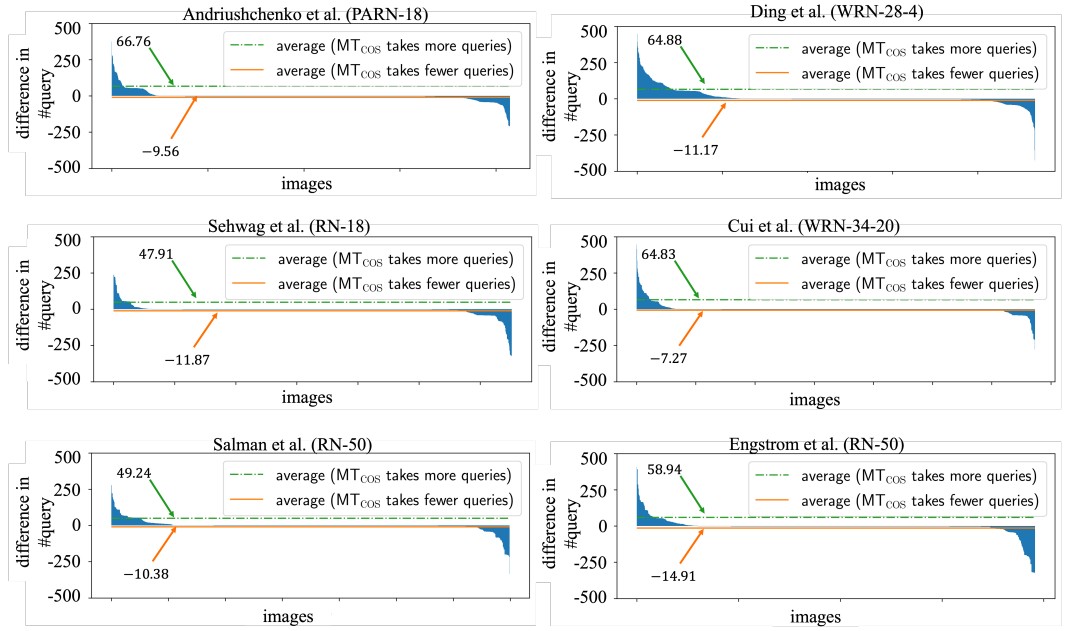

Figure 13: The difference between $MT_{cos}$ and GS+LS in #queries to find AEs for some models. The attacked models are Andriushchenko & Flammarion (2020), Sehwag et al. (2022), and Ding et al. (2020) for CIFAR-10, Cui et al. (2021) for CIFAR-100, and Salman et al. (2020) and Engstrom et al. (2019) for ImageNet.

### D.5 DETAILED EXPLANATION FOR FIGURE 6. IN THE MAIN TEXT

This analysis examines how many queries method A takes, on average, to find an adversarial example compared to method B when method A takes more queries than method B. Figures 6 and 13 illustrate the difference in the number of queries and their averages for images where attack methods A and B successfully found adversarial examples, but took different numbers of queries. For example, "average ($MT_{cos}$ takes more queries)" in figs. 6 and 13 represents the average difference in the number of queries required by $MTcos$ and those by GS+LS for images where $MT_{cos}$ spent more queries. Based on the comparison between "average ($MT_{cos}$ takes more queries)" and "average ($MT_{cos}$ takes fewer queries)", we argue in the main text that "Figure 6 shows that GS+LS found adversarial examples in fewer queries on average than $MT_{cos}$".

### D.6 ANALYSIS OF EDA USING AN INDEX BASED ON EUCLID DISTANCE

DI takes small values when the point set forms a cluster, even if the Euclidean distance between any two points is large. Therefore, quantification by DI and quantification based on the Euclidean distance between points in the point set may have different characteristics. Therefore, in this section, to compare the diversification performance from a different perspective than DI, we consider quantifying the degree of diversification of the best point set based on the average value of the Euclidean distance between the centroid of the point set X and all points in the point set $X$. Mathematically, the average Euclidean distance between all points in a point set $X$ and the centroid of the point set $X$ is defined as

$$E(X) = \frac{1}{|X|} \sum_{x \in X} \|x - \bar{x}\|_2, \tag{22}$$

where $\bar{x}$ is the centroid of the point set X, defined as $\bar{x} = \frac{1}{|X|} \sum_{x \in X} x$. As shown in fig. 14, the value of equation 22 tends to be larger for EDA than for $A^3$ in most models where EDA has higher attack performance than $A^3$. This difference is more pronounced than the difference in DI. While

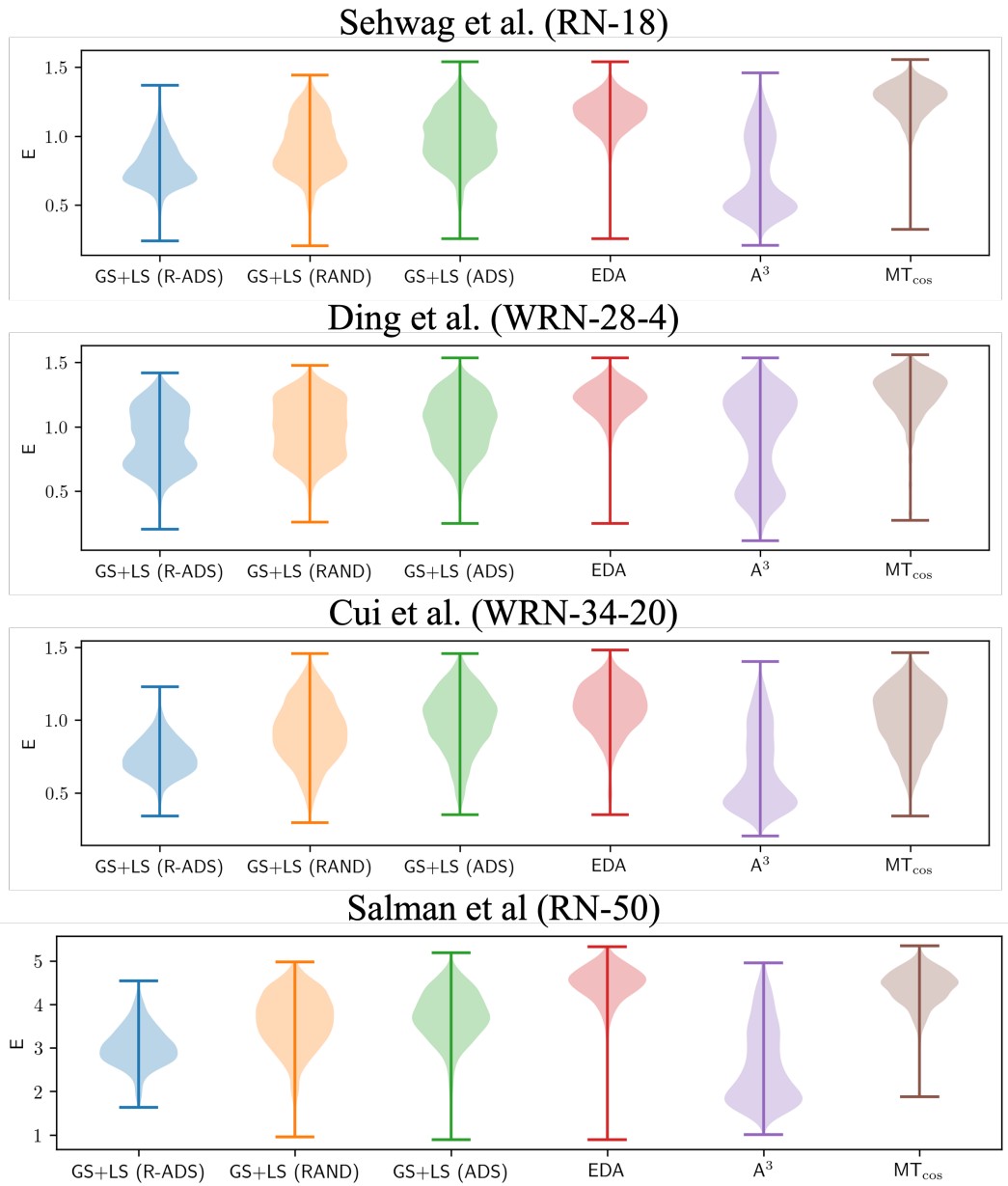

Figure 14: Violin plot of equation 22. The attacks models are Sehwag et al. (2022) and Ding et al. (2020) for CIFAR-10, Cui et al. (2021) for CIFAR-100, and Salman et al. (2020) for ImageNet.

EDA shows a similar trend for all models, $A^3$ shows a different trend in the value of equation 22 for some models. For example, as shown in fig. 14, the value of equation 22 for $A^3$ tends to be larger for the model proposed by Ding et al. (2020) than for the other models. Given the high attack performance of $A^3$ against these models, this suggests that the $A^3$ diversification strategy may be more effective for these models.

### D.7 ANALYSIS OF EDA FOR THE MODEL PROPOSED BY DING ET AL.

The attack performance of EDA is significantly lower for the model proposed by Ding et al. (2020) compared to $A^3$. This section discusses the reasons for this regarding diversification performance and computation time. As described in the main text, the value of DI for the best point set tends to be higher for EDA and lower for $A^3$, similar to the results for other models. On the other hand, the analysis in the previous section shows that for the model proposed by Ding et al. (2020), the value of equation 22 for the best point set obtained by $A^3$ tends to take larger values than the results for the other models. In addition, a comparison of the computation time for EDA and $A^3$ shows that $A^3$ takes more than three times longer than EDA. The above comparison suggests that the $A^3$ can perform better diversification for the model than for other models. In summary, setting a longer computation time and increasing the number of multi-restart are considered particularly effective in improving the attack performance for the model proposed by Ding et al. (2020).

### D.8 TRENDS OF SEARCH DIRECTIONS AND OBJECTIVE FUNCTIONS SELECTED BY ADS

Figure 15 is a bar chart displaying the number of times each search direction and objective function pair was used by EDA. Figure 15 shows that the combination of $\delta_{\text{ACG}}$ and $L_{\text{G-DLR},q}$ is frequently used in GS, and $\delta_{\text{Nes}}$ is rarely used. In LS, all combinations tend to be used at least once with $\delta_{\text{Nes}}$, $L_{\text{CW}}$, and $L_{\text{SCW}}$ being used more often. This trend is independent of $\mathcal{J}$ and may reflect ACG's high diversification performance and NAG's high intensification performance. The potential reasons for these trends are: 1. $P_i^e$ and DI play different roles from each other, 2. the ACG's search direction may be similar to the steepest for small step sizes, and 3. the difference between Nesterov's acceleration gradient direction and gradient direction.

**The role of $P_i^e$ and DI.** The $P_i^e$ measures the degree of diversification in the output space during the search. Therefore, a pair with the largest $P_i^e$ is expected to search for a high diversity in the output space. In addition, from Yamamura et al. (2022), it can be assumed that the ACG direction increases $P_i^e$, while the steepest-like direction does not. From the above, it is considered that the pair with the maximum $P_i^e$ is likely to include the ACG direction. DI measures the diversity of the best point set obtained by the search. In our use case, DI represents the dissimilarity between the best points. That is, we expect that pairs with the largest DI are more likely to enumerate dissimilar solutions. Intuitively, updates in diverse directions contribute to the enumeration of dissimilar solutions. Given that the search direction is gradient-dependent, the pair with the largest DI is likely to include a variety of objective functions and update formulas.

**ACG's search direction may be similar to the steepest for small step sizes.** The reason for this is as follows. According to the equation 9, $s^{(k)}$ is close to the gradient $\nabla L(g(\boldsymbol{x}^{(k)}), c)$ when $\beta^{(k)}$ is close to 0. From equation 8, $\langle \nabla L(g(\boldsymbol{x}^{(k)}), c), \boldsymbol{y}^{(k-1)} \rangle$ is the numerator of $\beta^{(k)}$. Therefore, as $\boldsymbol{y}^{(k-1)}$ approaches $\boldsymbol{0}$, $\beta^{(k)}$ also approaches 0. When the step size is small, $\|x^{(k)} - x^{(k-1)}\|$ is also small, so $\boldsymbol{y}^{(k-1)}$ is likely to be close to $\boldsymbol{0}$. As a result, the ACG's direction and the steepest direction may be similar. The experiments conducted by Yamamura et al. (2022) also support this claim.

**The difference between Nesterov's acceleration gradient direction and gradient direction.** Nesterov's accelerated gradient (NAG) method updates the search point using the gradient of the point moved from the current search point to the momentum direction. Assuming that the objective function is multimodal, the gradient at the current point is unlikely to be similar to NAG's search direction. Thus, if the objective function is multimodal, a search in the NAG's direction may find different local solutions from that in the gradient direction.

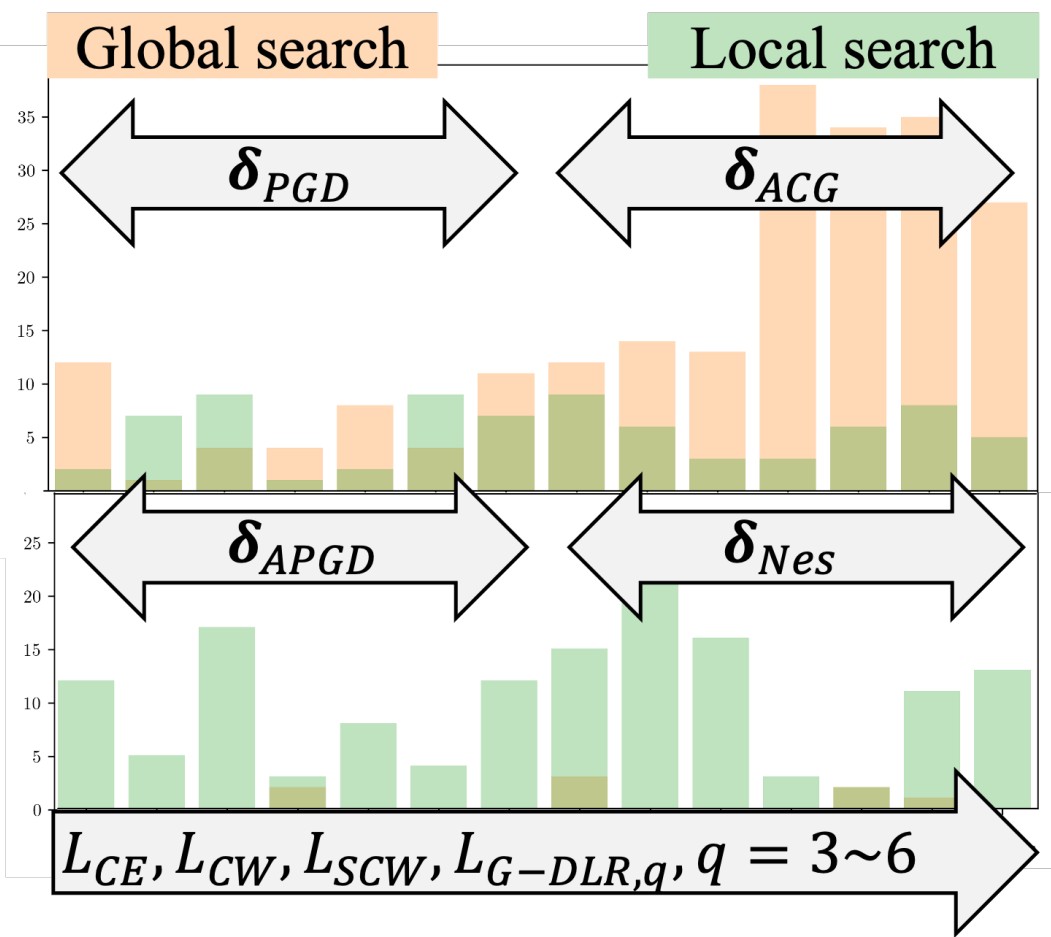

Figure 15: The number of times each pair of search direction and objective function is used in GS+LS(ADS).

# E  PREDICTION AWARE SAMPLING (PAS)

**Motivation**  The hypothesis behind PAS is that starting the search with an initial point near multiple decision boundaries increases the likelihood of finding an adversarial example. When maximizing the inner product of a random vector and logit, as ODS does, the distance to the decision boundary may be farther away than the initial point. However, when moving in the direction where the predicted probability for the correct class is as small as possible, the initial point is more likely to be closer to the decision boundary than the original point. We hypothesized that the attack's success rate could be improved by starting the search at a point closer to the decision boundary.

**Prediction Aware Sampling**  One promising initial point sampling is ODS, which considers diversification in the output space. However, there is room for improvement because its sampling does not consider image-specific information. Based on the idea that the randomly sampled initial point close to decision boundaries makes the attacks easier to succeed, we propose Prediction-Aware Sampling (PAS), a variant of ODS. PAS maximizes the following function in the same way as ODS to sample the initial point.

$$v(\boldsymbol{w}, g, \boldsymbol{x}) = \boldsymbol{w}^T g(\boldsymbol{x}) \times \exp\left(-g_c(\boldsymbol{x})\right), \quad \left(\boldsymbol{w} \sim U\left(-1, 1\right)^C\right) \tag{23}$$

PAS samples the initial point by repeating the following updates for $N_{\text{PAS}}$ iterations.

$$\boldsymbol{x} \leftarrow P_{\mathcal{S}}\left(\boldsymbol{x} + \eta_{\text{PAS}} \operatorname{sign}\left(\frac{\nabla_{\boldsymbol{x}} v(\boldsymbol{w}, g, \boldsymbol{x})}{\|\nabla_{\boldsymbol{x}} v(\boldsymbol{w}, g, \boldsymbol{x})\|_2}\right)\right) \tag{24}$$

Same as ODS, PAS used $N_{\text{PAS}} = 2$ and $\eta_{\text{PAS}} = \varepsilon$. Intuitively, maximizing equation 23 means moving the initial point closer to the decision boundary by reducing the prediction probability of the correct class $c$ and, at the same time, moving the logit $g(\boldsymbol{x})$ closer to the random vector $\boldsymbol{w}$.

**Experiments**  To test our hypothesis, we compared the success rate for each class in targeted attacks with nine target classes, using the input point, the point sampled by ODS, and the point sampled by PAS as initial points. In our notation, we compared the attack performance of $(\phi, \psi_{\cos}, \boldsymbol{\delta}_{\text{GD}}, L_{\text{CW}}^T), \phi \in \{\phi_{\text{org}}, \phi_{\text{ODS}}, \phi_{\text{PAS}}\}$ with 100 iterations for each target class and initial step size of $2\varepsilon$. The number of target classes $K$ was set to 9. The following five models were used in the experiments. 1. ResNet-18 (Sehwag et al., 2022), 2. WideResNet-28-10 (Gowal et al., 2020), 3. PreActResNet-18 (Rice et al., 2020), 4. WideResNet-34-10 (Sitawarin et al., 2021), 5. ResNet-50 (Engstrom et al., 2019). These numbers correspond to the "No." column in table 11. The experimental results in table 11 show that the attack with PAS can achieve higher attack success rates for many target classes than other initial point selections. The experimental results support our hypothesis that PAS brings the starting point closer to the decision boundary, resulting in a more successful attack. As described in appendix G.2, the ablation results for the initial point of the EDA also indicate that the PAS contributes to the attack performance of the EDA and GS+LS.

# F  TARGETED ATTACK IN EDA

**Motivation**  The motivation for using a targeted attack is to efficiently diversify the most likely prediction class of the adversarial example away from the correct class (diversification in the output space). CW loss and DLR loss are objective functions that generate adversarial examples misclassified into the class with the highest prediction probability among classes other than the correct class. In other words, they attempt to generate an adversarial example misclassified into the class whose decision boundary is closest to the current point. However, it is difficult to approach the decision boundary when the gradient is zero, even if the distance to the decision boundary is close, because the gradient-based attack moves in the direction of the gradient. In addition, Yamamura et al. (2022) reported that in the steepest gradient-based attacks, the class with the highest prediction probability among classes other than the correct class hardly changes during the search. Considering these factors, diversification in the output space could be effective, especially for attacks with untargeted losses. Some existing research also supports the effectiveness of diversification in the output space (Tashiro et al., 2020; Gowal et al., 2019).

Table 11: Validation of PAS. The lowest robust accuracy is in bold.

| Dataset | No. | target | input | ODS | PAS |
|---|---|---|---|---|---|
| CIFAR-10 | 1 | 1 | 57.41 | 57.40 | **57.29** |
| | | 2 | 65.33 | 64.82 | **64.28** |
| | | 3 | 67.85 | 66.98 | **66.27** |
| | | 4 | 69.06 | 68.21 | **67.52** |
| | | 5 | 69.88 | 68.77 | **67.81** |
| | | 6 | 69.66 | 68.59 | **67.88** |
| | | 7 | 69.11 | 68.37 | **67.59** |
| | | 8 | 68.08 | 67.47 | **67.10** |
| | | 9 | 67.32 | 66.84 | **66.46** |
| CIFAR-10 | 2 | 1 | **57.52** | 57.54 | 57.57 |
| | | 2 | 65.93 | 65.79 | **65.33** |
| | | 3 | 68.88 | 68.63 | **68.15** |
| | | 4 | 70.62 | 70.34 | **69.33** |
| | | 5 | 71.41 | 70.95 | **70.00** |
| | | 6 | 72.37 | 71.82 | **70.71** |
| | | 7 | 72.57 | 71.99 | **70.80** |
| | | 8 | 73.09 | 72.49 | **71.31** |
| | | 9 | 72.86 | 72.28 | **71.08** |
| CIFAR-100 | 3 | 1 | 20.37 | 20.39 | **20.34** |
| | | 2 | 24.71 | 24.67 | **24.46** |
| | | 3 | 26.65 | 26.64 | **26.14** |
| | | 4 | 27.70 | 27.49 | **26.99** |
| | | 5 | 28.37 | 28.36 | **27.70** |
| | | 6 | 29.02 | 28.98 | **28.33** |
| | | 7 | 29.55 | 29.32 | **28.62** |
| | | 8 | 29.83 | 29.68 | **28.95** |
| | | 9 | 29.72 | 29.55 | **28.94** |
| CIFAR-100 | 4 | 1 | 27.25 | 27.22 | **27.17** |
| | | 2 | 31.87 | 31.74 | **31.50** |
| | | 3 | 33.70 | 33.73 | **33.18** |
| | | 4 | 34.92 | 34.72 | **34.17** |
| | | 5 | 36.00 | 35.91 | **35.13** |
| | | 6 | 36.55 | 36.50 | **35.58** |
| | | 7 | 36.75 | 36.56 | **35.87** |
| | | 8 | 37.34 | 37.20 | **36.39** |
| | | 9 | 37.81 | 37.54 | **36.73** |
| ImageNet | 5 | 1 | 32.66 | 32.62 | **32.42** |
| | | 2 | 38.04 | 37.98 | **37.68** |
| | | 3 | 39.64 | 39.64 | **39.22** |
| | | 4 | 41.50 | 41.34 | **40.74** |
| | | 5 | 41.68 | 41.48 | **41.00** |
| | | 6 | 42.82 | 42.80 | **42.20** |
| | | 7 | 43.10 | 42.92 | **42.46** |
| | | 8 | 43.22 | 43.02 | **42.52** |
| | | 9 | 43.20 | 43.12 | **42.58** |

Table 12: Validation of targeted attack

| Dataset | No. | EDA | EDA+ |
|---------|-----|-----|------|
| CIFAR-10 | 1 | **55.49** | **55.49** |
| CIFAR-10 | 2 | **56.75** | 56.78 |
| CIFAR-100 | 3 | **18.86** | 18.87 |
| CIFAR-100 | 4 | **24.47** | 24.48 |
| ImageNet | 5 | **29.00** | 29.02 |

**Target selection**   Although a multi-target attack shows high performance by achieving diversification in the output space, existing methods (Gowal et al., 2019; Croce & Hein, 2020b) are computationally expensive because they assign an equal number of iterations to each target. To reduce the computational cost of the multi-target attack, we propose Target Selection (TS), which estimates the easiest target class to attack based on a small-scale search. TS estimates the easiest target class to attack based on a small-scale search to reduce the computational cost of the multi-target attack. The objective is to reduce the computational cost by focusing the number of iterations on the selected target. The procedure of TS is as follows: (1) Upper $K$ classes with large logit values of initial points are selected as target candidates. (2) Targeted attacks are performed for $N_s$ iterations for each target candidate. (3) The output is the target candidate $T$ with the highest objective function value. The detailed pseudocode of target selection is provided in algorithm 2.

**Hyperparameters**   The parameters of the targeted attack are the number of candidate targets in target selection $K = 9, 14, 20$, the number of iterations $N_s = 10$, and the number of iterations in targeted attack $N_4 = 100$. We chose $N_4 = 100$ because the number of iterations per targeted attack in AA is set to 100, which achieves a reasonable trade-off between computational cost and attack performance.

**Experiments**   To investigate the validity of this target selection, we compare the attack performance of EDA+, which executes a normal targeted attack after GS+LS, with that of EDA. In this experiment, we performed $(\phi_{\text{PAS}}, \psi_{\cos}, \boldsymbol{\delta}_{\text{GD}}, L_{\text{CW}}^T)$ with 100 iterations for each target class and initial step size of $2\varepsilon$. The following five models were used in the experiments. 1. ResNet-18 (Sehwag et al., 2022), 2. WideResNet-28-10 (Gowal et al., 2020), 3. PreActResNet-18 (Rice et al., 2020), 4. WideResNet-34-10 (Sitawarin et al., 2021), 5. ResNet-50 (Engstrom et al., 2019). These numbers correspond to the "No." column in table 12. While the targeted attack with a single target class selected by TS requires $N_s \times K + N_4 = 10K + 100$ queries per image, the normal targeted attack requires $K \times N_4 = 100K$ queries. Given the parameter of $K = 9, 14, 20 \geq 2$, the targeted attack with a single target class selected by TS requires fewer queries than the normal targeted attack. Considering the results described in table 12, target selection may reduce runtime without significantly degrading the attack performance of EDA.

# G   ABLATION STUDY

## G.1   HYPERPARAMETER SENSITIVITY OF EDA

We investigated the impact of hyperparameter values on EDA's performance. In this experiment, the following five models were used. 1. ResNet-18 (Sehwag et al., 2022), 2. WideResNet-28-10 (Gowal et al., 2020), 3. PreActResNet-18 (Rice et al., 2020), 4. WideResNet-34-10 (Sitawarin et al., 2021), 5. ResNet-50 (Engstrom et al., 2019). Tables 13 to 15 show the robust accuracy obtained by EDA with each parameter value. Although the attack performance of GS+LS is different among different hyperparameters as described in table 10, the EDA's performance is stable regardless of the hyperparameter setting. These experimental results imply that the attacks composed by different strategies could be robust to the hyperparameter settings. In addition, we tested the EDA's performance with several targeted attacks. As described in table 16, the search directions based on the steepest direction performed better than the conjugate gradient-based direction. Also, the margin-based losses showed higher performance than the CE loss.

Table 13: Ablation study of EDA. The default parameter is in bold.

| Dataset | No. | $N_s$ | | | # sampled images | | | |
|---------|-----|-------|------|-------|------|-------|-------|-------|
| | | 5 | **10** | 15 | **1%** | 3% | 5% | 7% |
| CIFAR-10 | 1 | 55.59 | 55.49 | 55.49 | 55.49 | 55.47 | 55.48 | 55.50 |
| CIFAR-10 | 2 | 56.73 | 56.76 | 56.76 | 56.76 | 56.81 | 56.81 | 56.79 |
| CIFAR-100 | 3 | 18.85 | 18.86 | 18.86 | 18.86 | 18.87 | 18.88 | 18.87 |
| CIFAR-100 | 4 | 24.46 | 24.47 | 24.48 | 24.47 | 24.49 | 24.49 | 24.47 |
| ImageNet | 5 | 29.04 | 29.04 | 29.04 | 29.04 | 29.04 | 29.08 | 29.04 |

Table 14: Ablation study of EDA. The default parameter is in bold.

| Dataset | No. | initial stepsize | | | | $N_{\text{ADS}}$ | | | |
|---------|-----|------------------|------------------|------------|------------|------|------|------|------|
| | | $\varepsilon/4$ | $\varepsilon/2$ | $\varepsilon$ | **$2\varepsilon$** | 3 | **4** | 5 | 10 |
| CIFAR-10 | 1 | 55.50 | 55.48 | 55.51 | 55.49 | 55.47 | 55.49 | 55.50 | 55.51 |
| CIFAR-10 | 2 | 56.78 | 56.78 | 56.78 | 56.75 | 56.78 | 56.76 | 56.77 | 56.76 |
| CIFAR-100 | 3 | 18.92 | 18.92 | 18.87 | 18.86 | 18.88 | 18.86 | 18.89 | 18.89 |
| CIFAR-100 | 4 | 24.49 | 24.50 | 24.50 | 24.47 | 24.47 | 24.47 | 24.49 | 24.48 |
| ImageNet | 5 | 29.10 | 29.08 | 29.08 | 29.00 | 29.00 | 29.04 | 29.02 | 29.04 |

Table 15: Ablation study of EDA. The default parameter is in bold.

| Dataset | No. | $n_a$ | | | | $N_1$ and $N_2$ | | | | |
|---------|-----|-------|------|------|------|-------|------|------|------|------|
| | | 3 | 4 | **5** | 6 | $N_1$:**22** $N_2$:**19** | 30 10 | 10 30 | 10 10 | 30 30 |
| CIFAR-10 | 1 | 55.50 | 55.48 | 55.48 | 55.49 | 55.49 | 55.51 | 55.52 | 55.49 | 55.48 |
| CIFAR-10 | 2 | 56.77 | 56.82 | 56.76 | 56.76 | 56.75 | 56.81 | 56.73 | 56.77 | 56.80 |
| CIFAR-100 | 3 | 18.87 | 18.90 | 18.89 | 18.86 | 18.86 | 18.89 | 18.88 | 18.91 | 18.87 |
| CIFAR-100 | 4 | 24.48 | 24.49 | 24.48 | 24.47 | 24.47 | 24.48 | 24.48 | 24.51 | 24.51 |
| ImageNet | 5 | 29.06 | 29.04 | 29.04 | 29.02 | 29.00 | 29.04 | 29.06 | 29.04 | 29.04 |

Table 16: Ablation for the targeted attack. The default setting is in bold.

| Dataset | No. | $\delta_{\text{GD}}$ | | | $\delta_{\text{CG}}$ | | $\delta_{\text{APGD}}$ |
|---------|-----|----------------|----------------|-----------------|----------------|-----------------|-----------------|
| | | **$L_{\text{CW}}^T$** | $L_{\text{CE}}^T$ | $L_{\text{DLR}}^T$ | $L_{\text{CW}}^T$ | $L_{\text{DLR}}^T$ | $L_{\text{DLR}}^T$ |
| CIFAR-10 | 1 | 55.49 | 55.57 | 55.48 | 55.50 | 55.50 | 55.48 |
| CIFAR-10 | 2 | 56.76 | 56.89 | 56.81 | 56.81 | 56.83 | 56.79 |
| CIFAR-100 | 3 | 18.86 | 18.96 | 18.90 | 18.93 | 18.94 | 18.91 |
| CIFAR-100 | 4 | 24.47 | 24.63 | 24.50 | 24.53 | 24.53 | 24.50 |
| ImageNet | 5 | 29.00 | 29.18 | 29.02 | 29.08 | 29.06 | 29.00 |

## G.2 THE IMPACT OF THE MDO AND MT STRATEGIES AND PAS ON EDA PERFORMANCE

To investigate the contribution of the MDO and MT strategies and PAS to the performance of EDA, we compare the robust accuracy obtained in the presence and absence of these components. Table 18 shows the robust accuracy obtained by each setting, and $\Delta$ represents the averaged difference over the 41 models between robust accuracy obtained by EDA and that by each method. The negative $\Delta$ indicates lower attack performance than EDA. ID 9 in table 18 is the EDA. The ID columns in Table 18 indicate the presence and absence of PAS, GS+LS, and targeted attack, corresponding to table 17.

Table 17: Mapping of IDs to the presence or absence of the three diversification strategies.

| ID | 1 | 2 | 3 | 4 | 5 | 6 | 7 | 8 | 9 |
|---|---|---|---|---|---|---|---|---|---|
| initial point | $\phi_{org}$ | $\phi_{ODS}$ | $\phi_{PAS}$ | $\phi_{org}$ | $\phi_{ODS}$ | $\phi_{PAS}$ | $\phi_{org}$ | $\phi_{ODS}$ | $\phi_{PAS}$ |
| G&L search | ✓ | ✓ | ✓ | - | - | - | ✓ | ✓ | ✓ |
| multi-target | - | - | - | ✓ | ✓ | ✓ | ✓ | ✓ | ✓ |

Table 18: Robust accuracy of the ablation study for EDA.

| CIFAR-10 ($\varepsilon = 8/255$) | model | ID 1 | 2 | 3 | 4 | 5 | 6 | 7 | 8 | 9 |
|---|---|---|---|---|---|---|---|---|---|---|
| (Andriushchenko & Flammarion, 2020) | PARN-18 | 43.96±0.03 | 44.05±0.07 | 43.95±0.02 | 43.96±0.00 | 43.96±0.00 | 43.95±0.00 | 43.85±0.02 | 43.86±0.01 | 43.85±0.02 |
| (Addepalli et al., 2022) | RN-18 | 52.50±0.03 | 52.66±0.12 | 52.49±0.03 | 52.49±0.00 | 52.47±0.00 | 52.46±0.00 | 52.41±0.01 | 52.45±0.02 | 52.43±0.03 |
| (Sehwag et al., 2022) | RN-18 | 55.62±0.03 | 55.60±0.02 | 55.56±0.02 | 55.55±0.00 | 55.56±0.00 | 55.56±0.00 | 55.57±0.00 | 55.50±0.01 | 55.49±0.01 |
| (Engstrom et al., 2019) | RN-50 | 49.40±0.03 | 49.34±0.06 | 49.43±0.09 | 49.23±0.00 | 49.25±0.00 | 49.28±0.00 | 49.11±0.01 | 49.10±0.01 | 49.10±0.03 |
| (Carmon et al., 2019) | WRN-28-10 | 59.49±0.03 | 59.62±0.03 | 59.49±0.03 | 59.53±0.00 | 59.51±0.00 | 59.55±0.00 | 59.40±0.01 | 59.45±0.02 | 59.40±0.01 |
| (Gowal et al., 2020) | WRN-28-10 | 62.84±0.02 | 62.99±0.05 | 62.82±0.03 | 62.84±0.00 | 62.84±0.00 | 62.85±0.00 | 62.75±0.01 | 62.78±0.02 | 62.75±0.02 |
| (Hendrycks et al., 2019) | WRN-28-10 | 54.87±0.03 | 54.84±0.01 | 54.85±0.04 | 54.83±0.00 | 54.84±0.00 | 54.87±0.00 | 54.77±0.01 | 54.77±0.01 | 54.77±0.02 |
| (Rebuffi et al., 2021) | WRN-28-10 | 60.78±0.04 | 60.83±0.07 | 60.75±0.03 | 60.78±0.00 | 60.78±0.00 | 60.75±0.00 | 60.69±0.02 | 60.69±0.02 | 60.64±0.01 |
| (Sehwag et al., 2020) | WRN-28-10 | 57.11±0.03 | 57.22±0.05 | 57.11±0.02 | 57.17±0.00 | 57.17±0.00 | 57.16±0.00 | 57.01±0.02 | 57.07±0.03 | 57.03±0.01 |
| (Sridhar et al., 2022) | WRN-28-10 | 59.62±0.02 | 59.68±0.10 | 59.55±0.03 | 59.64±0.00 | 59.63±0.00 | 59.63±0.00 | 59.48±0.03 | 59.52±0.03 | 59.46±0.02 |
| (Wang et al., 2020) | WRN-28-10 | 56.48±0.06 | 56.52±0.24 | 56.29±0.02 | 56.32±0.00 | 56.34±0.00 | 56.36±0.00 | 56.18±0.01 | 56.20±0.04 | 56.15±0.02 |
| (Wu et al., 2020) | WRN-28-10 | 60.01±0.02 | 60.11±0.09 | 60.00±0.02 | 60.03±0.00 | 60.02±0.00 | 60.03±0.00 | 59.94±0.01 | 59.97±0.03 | 59.94±0.01 |
| (Ding et al., 2020) | WRN-28-4 | 43.33±0.16 | 43.69±0.13 | 43.32±0.12 | 42.27±0.00 | 42.63±0.00 | 42.58±0.00 | 41.70±0.04 | 41.84±0.03 | 41.74±0.06 |
| (Addepalli et al., 2022) | WRN-34-10 | 57.78±0.01 | 58.01±0.09 | 57.75±0.03 | 57.76±0.00 | 57.76±0.00 | 57.77±0.00 | 57.70±0.01 | 57.74±0.01 | 57.69±0.02 |
| (Sehwag et al., 2022) | WRN-34-10 | 60.27±0.02 | 60.30±0.02 | 60.27±0.05 | 60.25±0.00 | 60.31±0.00 | 60.31±0.00 | 60.17±0.01 | 60.18±0.02 | 60.18±0.02 |
| (Sitawarin et al., 2021) | WRN-34-10 | 50.72±0.04 | 50.69±0.03 | 50.72±0.03 | 50.73±0.00 | 50.73±0.00 | 50.73±0.00 | 50.59±0.02 | 50.60±0.02 | 50.59±0.01 |
| (Zhang et al., 2019a) | WRN-34-10 | 44.62±0.03 | 44.65±0.04 | 44.65±0.06 | 44.65±0.00 | 44.71±0.00 | 44.63±0.00 | 44.51±0.02 | 44.52±0.02 | 44.51±0.02 |
| (Zhang et al., 2020) | WRN-34-10 | 53.53±0.03 | 53.56±0.03 | 53.50±0.03 | 53.49±0.00 | 53.50±0.00 | 53.47±0.00 | 53.43±0.01 | 53.44±0.03 | 53.42±0.02 |
| (Sridhar et al., 2022) | WRN-34-15 | 60.43±0.02 | 60.44±0.11 | 60.38±0.03 | 60.40±0.00 | 60.37±0.00 | 60.40±0.00 | 60.31±0.02 | 60.32±0.03 | 60.32±0.01 |
| (Gowal et al., 2020) | WRN-34-20 | 56.89±0.02 | 57.08±0.05 | 56.86±0.02 | 56.83±0.00 | 56.86±0.00 | 56.88±0.00 | 56.79±0.01 | 56.81±0.01 | 56.79±0.03 |
| (Pang et al., 2020) | WRN-34-20 | 53.85±0.03 | 53.80±0.03 | 53.82±0.03 | 53.73±0.00 | 53.73±0.00 | 53.72±0.00 | 53.64±0.00 | 53.67±0.01 | 53.66±0.01 |
| (Rice et al., 2020) | WRN-34-20 | 53.47±0.02 | 53.56±0.09 | 53.46±0.04 | 53.35±0.00 | 53.39±0.00 | 53.41±0.00 | 53.33±0.01 | 53.34±0.01 | 53.34±0.01 |
| (Gowal et al., 2020) | WRN-70-16 | 57.21±0.02 | 57.28±0.05 | 57.18±0.02 | 57.15±0.00 | 57.17±0.00 | 57.17±0.00 | 57.12±0.00 | 57.14±0.01 | 57.12±0.01 |
| (Gowal et al., 2020) | WRN-70-16 | 65.92±0.04 | 65.91±0.05 | 65.87±0.01 | 65.90±0.00 | 65.89±0.00 | 65.91±0.00 | 65.81±0.03 | 65.81±0.02 | 65.83±0.01 |
| (Rebuffi et al., 2021) | WRN-70-16 | 64.31±0.04 | 64.36±0.06 | 64.30±0.05 | 64.26±0.00 | 64.24±0.00 | 64.25±0.00 | 64.20±0.01 | 64.20±0.02 | 64.20±0.03 |
| **CIFAR-100 ($\varepsilon = 8/255$)** | model | ID 1 | 2 | 3 | 4 | 5 | 6 | 7 | 8 | 9 |
| (Rice et al., 2020) | PARN-18 | 18.97±0.01 | 19.04±0.09 | 18.98±0.01 | 18.92±0.00 | 18.90±0.00 | 18.92±0.00 | 18.87±0.01 | 18.86±0.01 | 18.88±0.01 |
| (Hendrycks et al., 2019) | WRN-28-10 | 28.46±0.02 | 28.60±0.13 | 28.41±0.02 | 28.37±0.00 | 28.35±0.00 | 28.38±0.00 | 28.28±0.01 | 28.28±0.02 | 28.27±0.02 |
| (Rebuffi et al., 2021) | WRN-28-10 | 32.22±0.03 | 32.45±0.23 | 32.07±0.04 | 32.03±0.00 | 32.05±0.00 | 32.04±0.00 | 31.97±0.01 | 31.99±0.01 | 31.94±0.03 |
| (Addepalli et al., 2022) | WRN-34-10 | 31.90±0.03 | 32.01±0.16 | 31.87±0.02 | 31.79±0.00 | 31.81±0.00 | 31.83±0.00 | 31.78±0.01 | 31.78±0.01 | 31.78±0.01 |
| (Cui et al., 2021) | WRN-34-10 | 29.00±0.04 | 29.14±0.23 | 28.96±0.05 | 28.91±0.00 | 28.92±0.00 | 28.91±0.00 | 28.85±0.02 | 28.84±0.02 | 28.83±0.02 |
| (Sitawarin et al., 2021) | WRN-34-10 | 24.61±0.04 | 24.67±0.07 | 24.66±0.03 | 24.55±0.00 | 24.56±0.00 | 24.58±0.00 | 24.50±0.01 | 24.48±0.01 | 24.50±0.01 |
| (Wu et al., 2020) | WRN-34-10 | 28.95±0.04 | 29.17±0.19 | 28.89±0.04 | 28.82±0.00 | 28.83±0.00 | 28.84±0.00 | 28.76±0.01 | 28.77±0.01 | 28.76±0.01 |
| (Cui et al., 2021) | WRN-34-20 | 30.06±0.02 | 30.11±0.10 | 30.00±0.02 | 29.92±0.00 | 29.92±0.00 | 29.93±0.00 | 29.86±0.01 | 29.86±0.01 | 29.85±0.01 |
| (Gowal et al., 2020) | WRN-70-16 | 30.13±0.03 | 30.29±0.21 | 30.07±0.01 | 29.99±0.00 | 30.00±0.00 | 30.00±0.00 | 29.97±0.01 | 29.99±0.01 | 29.96±0.01 |
| (Gowal et al., 2020) | WRN-70-16 | 37.06±0.04 | 37.17±0.15 | 36.96±0.04 | 36.88±0.00 | 36.91±0.00 | 36.87±0.00 | 36.81±0.01 | 36.83±0.01 | 36.81±0.01 |
| (Rebuffi et al., 2021) | WRN-70-16 | 34.69±0.04 | 34.81±0.09 | 34.64±0.03 | 34.62±0.00 | 34.66±0.00 | 34.63±0.00 | 34.57±0.02 | 34.58±0.02 | 34.55±0.01 |
| **ImageNet ($\varepsilon = 4/255$)** | model | ID 1 | 2 | 3 | 4 | 5 | 6 | 7 | 8 | 9 |
| (Salman et al., 2020) | RN-18 | 25.31±0.05 | 25.48±0.21 | 25.25±0.04 | 25.18±0.00 | 25.18±0.00 | 25.22±0.00 | 25.12±0.02 | 25.10±0.01 | 25.11±0.02 |
| (Engstrom et al., 2019) | RN-50 | 29.29±0.06 | 29.77±0.48 | 29.27±0.07 | 29.04±0.00 | 29.06±0.00 | 29.06±0.00 | 29.01±0.01 | 29.00±0.02 | 29.01±0.01 |
| (Salman et al., 2020) | RN-50 | 34.67±0.09 | 35.13±0.30 | 34.73±0.07 | 34.60±0.00 | 34.58±0.00 | 34.60±0.00 | 34.54±0.03 | 34.54±0.03 | 34.52±0.02 |
| (Wong et al., 2020) | RN-50 | 26.41±0.12 | 26.71±0.34 | 26.32±0.13 | 26.26±0.00 | 26.30±0.00 | 26.32±0.00 | 26.13±0.12 | 26.14±0.08 | 26.12±0.10 |
| (Salman et al., 2020) | WRN-50-2 | 38.42±0.04 | 39.36±0.10 | 38.31±0.06 | 38.06±0.00 | 38.08±0.00 | 38.10±0.00 | 38.04±0.02 | 38.03±0.03 | 38.03±0.02 |
| $\Delta$: Average of ID 9 − ID $k$ | | -0.19 | -0.33 | -0.15 | -0.09 | -0.11 | -0.11 | $-2.7\times10^{-3}$ | -0.02 | - |

Table 19: Comparison in robust accuracy of EDA with GS+LS(ADS). This table shows the extracted data from Table 18 to directly compare the attack performance between EDA and GS+LS(ADS).

| CIFAR-10 ($\varepsilon = 8/255$) | model | GS+LS (ADS) | EDA |
|---|---|---|---|
| (Andriushchenko & Flammarion, 2020) | PARN-18 | 43.95±0.02 | 43.85±0.02 |
| (Addepalli et al., 2022) | RN-18 | 52.49±0.03 | 52.43±0.03 |
| (Sehwag et al., 2022) | RN-18 | 55.56±0.02 | 55.49±0.01 |
| (Engstrom et al., 2019) | RN-50 | 49.43±0.09 | 49.10±0.03 |
| (Carmon et al., 2019) | WRN-28-10 | 59.49±0.03 | 59.40±0.01 |
| (Gowal et al., 2020) | WRN-28-10 | 62.82±0.03 | 62.75±0.02 |
| (Hendrycks et al., 2019) | WRN-28-10 | 54.85±0.04 | 54.77±0.02 |
| (Rebuffi et al., 2021) | WRN-28-10 | 60.75±0.03 | 60.64±0.01 |
| (Sehwag et al., 2020) | WRN-28-10 | 57.11±0.02 | 57.03±0.01 |
| (Sridhar et al., 2022) | WRN-28-10 | 59.55±0.03 | 59.46±0.02 |
| (Wang et al., 2020) | WRN-28-10 | 56.29±0.02 | 56.15±0.02 |
| (Wu et al., 2020) | WRN-28-10 | 60.00±0.02 | 59.94±0.01 |
| (Ding et al., 2020) | WRN-28-4 | 43.32±0.12 | 41.74±0.06 |
| (Addepalli et al., 2022) | WRN-34-10 | 57.75±0.03 | 57.69±0.02 |
| (Sehwag et al., 2022) | WRN-34-10 | 60.27±0.05 | 60.18±0.02 |
| (Sitawarin et al., 2021) | WRN-34-10 | 50.72±0.03 | 50.59±0.01 |
| (Zhang et al., 2019a) | WRN-34-10 | 44.65±0.06 | 44.51±0.02 |
| (Zhang et al., 2020) | WRN-34-10 | 53.50±0.03 | 53.42±0.02 |
| (Sridhar et al., 2022) | WRN-34-15 | 60.38±0.03 | 60.32±0.01 |
| (Gowal et al., 2020) | WRN-34-20 | 56.86±0.02 | 56.79±0.03 |
| (Pang et al., 2020) | WRN-34-20 | 53.82±0.03 | 53.66±0.01 |
| (Rice et al., 2020) | WRN-34-20 | 53.46±0.04 | 53.34±0.01 |
| (Gowal et al., 2020) | WRN-70-16 | 57.18±0.02 | 57.12±0.01 |
| (Gowal et al., 2020) | WRN-70-16 | 65.87±0.01 | 65.83±0.01 |
| (Rebuffi et al., 2021) | WRN-70-16 | 64.30±0.05 | 64.20±0.03 |

| CIFAR-100 ($\varepsilon = 8/255$) | model | GS+LS (ADS) | EDA |
|---|---|---|---|
| (Rice et al., 2020) | PARN-18 | 18.98±0.01 | 18.88±0.01 |
| (Hendrycks et al., 2019) | WRN-28-10 | 28.41±0.02 | 28.27±0.02 |
| (Rebuffi et al., 2021) | WRN-28-10 | 32.07±0.04 | 31.94±0.03 |
| (Addepalli et al., 2022) | WRN-34-10 | 31.87±0.02 | 31.78±0.01 |
| (Cui et al., 2021) | WRN-34-10 | 28.96±0.05 | 28.83±0.02 |
| (Sitawarin et al., 2021) | WRN-34-10 | 24.66±0.03 | 24.50±0.01 |
| (Wu et al., 2020) | WRN-34-10 | 28.89±0.04 | 28.76±0.01 |
| (Cui et al., 2021) | WRN-34-20 | 30.00±0.02 | 29.85±0.01 |
| (Gowal et al., 2020) | WRN-70-16 | 30.07±0.01 | 29.96±0.01 |
| (Gowal et al., 2020) | WRN-70-16 | 36.96±0.04 | 36.81±0.01 |
| (Rebuffi et al., 2021) | WRN-70-16 | 34.64±0.03 | 34.55±0.01 |

| ImageNet ($\varepsilon = 4/255$) | model | GS+LS (ADS) | EDA |
|---|---|---|---|
| (Salman et al., 2020) | RN-18 | 25.25±0.02 | 25.11±0.02 |
| (Engstrom et al., 2019) | RN-50 | 29.27±0.07 | 29.01±0.01 |
| (Salman et al., 2020) | RN-50 | 34.73±0.07 | 34.52±0.02 |
| (Wong et al., 2020) | RN-50 | 26.32±0.13 | 26.12±0.10 |
| (Salman et al., 2020) | WRN-50-2 | 38.31±0.06 | 38.03±0.02 |

**The impact of PAS:** In table 18, IDs 1, 4, and 7 take the input point as the initial point, and the rest employ random sampling. Table 18 shows that attacks that use input point as the initial point tend to display higher attack performance than attacks with ODS and lower attack performance than attacks with PAS. These trends suggest that the PAS positively impacts the attack performance of EDA.

**Impacts of MDO strategy:** In table 18, IDs 4 to 6 are attacks that do not use the MDO strategy, whereas IDs 7 to 9 are attacks that use the MDO strategy. Table 18 shows that the attack with the MDO strategy has higher attack performance than those not using the MDO strategy. Therefore, the MDO strategy is considered to affect the attack performance of EDA positively.

**Impact of multi-target attack:** In table 18, IDs 1 through 3 are without multi-target attacks, and IDs 7 through 9 are with multi-target attacks. Comparing IDs 1 with 7, 2 with 8, and 3 with 9, IDs with the multi-target attack exhibit higher attack performance in all cases. That is, the multi-target attack positively affects EDA's attack performance.

Table 20: Robust accuracy of randomized defenses. The lowest robust accuracy is in bold.

| CIFAR-10 ($\varepsilon = 8/255$) | | | AA | | $A^3$ | | EDA | |
|---|---|---|---|---|---|---|---|---|
| Defense | Model | clean | acc | sec | acc | sec | acc | sec |
| RNA | RN-18 | 84.48 | 62.13 | 5,094 | 65.99 | 1,342 | **46.13** | 743 |
| RNA | WRN-32 | 86.49 | 64.53 | 30,560 | 69.70 | 16,087 | **46.57** | 4,764 |
| DWQ | PARN-18 | 82.11 | 53.81 | 6,469 | 56.63 | 3,161 | **48.00** | 897 |
| DWQ | WRN-34 | 81.56 | 58.05 | 46,058 | 60.21 | 7,043 | **50.32** | 5,856 |
| SVHN ($\varepsilon = 8/255$) | | | | | | | | |
| DWQ | PARN-18 | 81.97 | 31.90 | 4,117 | 38.80 | 1,802 | **28.53** | 676 |
| DWQ | WRN-34 | 88.65 | 37.20 | 29,969 | 43.87 | 5,438 | **35.23** | 4,027 |
| CIFAR100 ($\varepsilon = 8/255$) | | | | | | | | |
| RNA | RN-18 | 56.75 | 42.78 | 2,378 | 47.55 | 1,921 | **30.67** | 426 |
| RNA | WRN-32 | 60.19 | 41.94 | 15,296 | 45.98 | 17,575 | **30.71** | 1,851 |
| DWQ | PARN-18 | 55.96 | 32.61 | 3,559 | 33.72 | 1,491 | **26.51** | 607 |
| DWQ | WRN-34 | 55.94 | 36.77 | 25,539 | 37.82 | 4,658 | **29.64** | 3,311 |
| ImageNet ($\varepsilon = 4/255$) | | | | | | | | |
| DWQ | RN-50 (free) | 49.44 | 35.62 | 2,386 | 45.08 | 109 | **24.18** | 829 |
| DWQ | RN-50 (fgsmrs) | 62.18 | 39.08 | 21,994 | 40.72 | 4,854 | **33.72** | 3,773 |

These results suggest that each component greatly impacts EDA's attack performance in the following descending order with or without: a multi-target attack, MDO strategy, and PAS.

## H  EDA'S PERFORMANCE AGAINST RANDOMIZED DEFENSES

We investigated the efficacy of EDA against two randomized defenses, including Double-Win Quant (DWQ) (Fu et al., 2021) and Random Normalization Aggregation (RNA) (Dong et al., 2022). For the models trained on CIFAR-10/100 and SVHN (Netzer et al., 2011), we used $\varepsilon = 8/255$ and 10,000 test images. For the ImageNet, we used $\varepsilon = 4/255$ and 5,000 images, the same as the RobustBench. Due to the computation cost, we report the results of a single run with a fixed random seed. For consistency with AA, we calculated the robust accuracy using generated adversarial examples when the attack terminated and compared these values because the model with randomized defense may produce different outputs with each inference. Official implementations of AA and EDA calculate robust accuracy in this way, but the official implementation of $A^3$ calculates differently. Therefore, we slightly modified the implementation of $A^3$ to calculate robust accuracy in the same way. The parameters of the compared attacks were the same as in the main text. Table 20 shows the robust accuracy and runtime of the standard version of AA, $A^3$, and EDA. From table 20, EDA showed higher attack performance in less computation time. Although these are the results of a single run, EDA is expected to be effective for randomized defenses.

## I  EDA'S PERFORMANCE AGAINST TRANSFORMER-BASED ARCHITECTURES

We investigated the performance of EDA against robust models trained on ImageNet including transformer-based architectures. We used $\varepsilon = 4/255$ and 5,000 images, the same as RobustBench. Due to the computation cost, we report the results from a single run with a fixed random seed. The parameters of the compared models were the same as in the main text. Table 21 shows the robust accuracy and runtime. The robust accuracy of AA is the value reported in RobustBench. Although the improvement in robust accuracy by EDA was smaller than those for CNNs reported in the main text, EDA showed the SOTA attack performance in less computation time for all models.

Table 21: The comparison in the robust accuracy against robust models trained on ImageNet, including transformer-based architectures. The lowest robust accuracy is in bold.

| Defense | Model | clean | AA | $A^3$ | | EDA | |
|---|---|---|---|---|---|---|---|
| | | acc | acc | acc | sec | acc | sec |
| (Liu et al., 2023a) | ConvNeXt-B | 76.02 | 55.82 | 55.84 | 37,437 | **55.78** | 14,073 |
| (Liu et al., 2023a) | Swin-B | 76.16 | 56.16 | 56.06 | 20,620 | **56.04** | 13,988 |
| (Debenedetti et al., 2022) | XCiT-L12 | 73.76 | 47.60 | 47.54 | 20,044 | **47.44** | 12,705 |
| (Debenedetti et al., 2022) | XCiT-M12 | 74.04 | 45.24 | 45.22 | 11,067 | **45.14** | 7,884 |
| (Debenedetti et al., 2022) | XCiT-S12 | 72.34 | 41.78 | 41.74 | 6,560 | **41.60** | 5,548 |
| (Singh et al., 2023) | ConvNeXt-B+ConvStem | 75.90 | 56.14 | 56.14 | 35,422 | **56.02** | 14,399 |
| (Singh et al., 2023) | ConvNeXt-S+ConvStem | 74.10 | 52.42 | 52.38 | 24,512 | **52.30** | 10,601 |
| (Singh et al., 2023) | ConvNeXt-T+ConvStem | 72.72 | 49.46 | 49.44 | 19,391 | **49.38** | 7,311 |
| (Singh et al., 2023) | ViT-B+ConvStem | 76.30 | 54.66 | 55.02 | 17,141 | **54.62** | 11,962 |
| (Singh et al., 2023) | ViT-S+ConvStem | 72.56 | 48.08 | 48.08 | 5,928 | **48.02** | 5,355 |

## J  LIMITATIONS AND ASSUMPTIONS

This study has the following assumptions and limitations. EDA assumes to be able to access the outputs and gradients of the attacked models. Same as the $A^3$, EDA assumes to attack the set of images. Thus, EDA cannot apply to a single image. The number of combinations of search directions and objective functions grows exponentially. Thus, the number of search directions and objective functions to be selected should be manageable.

**Expected situations in EDA work well**   This study experimentally verifies the performance of EDA, focusing on image classifiers using deterministic defenses such as adversarial training. Although EDA was tested with images of different resolutions, domains, and models of different architectures, EDA worked in most settings. Therefore, it can be assumed that EDA has some generalization performance for image classifiers using deterministic defenses. Similar to AA, EDA is applicable to the models whose gradients are available. However, the performance of EDA against models out of image classifiers needs further investigation.

**Expected situations in EDA do not work well**   EDA contains gradient-based attacks as some components. Therefore, EDA's performance may be degraded for models which cause incorrect gradient calculations. As reported by Croce & Hein (2020b), existing techniques like expectation over transformation might help improve the attack performance on these models. Also, the performance of EDA may be degraded when the MDO and MT strategies do not work well.

## K  BROADER IMPACTS

Deep neural networks are known to be vulnerable to adversarial examples. A promising defense mechanism to address this vulnerability is adversarial training, where training is performed using adversarial examples. Many adversarial examples generated by strong attack methods are required to produce robust models through adversarial training. Therefore, developing strong and fast adversarial attacks helps improve the robustness of DNNs. The EDA attack method, which is the main proposal of this research, can generate a larger number of adversarial examples in a shorter time and can be used for both the robustness evaluation of defense methods and data generation in adversarial training. Therefore, this research significantly contributes to the security of DNNs. The positive impact of this research is twofold. First, we can make DNNs more robust through adversarial training using the data generated by the strong attack method EDA. Second, we can more accurately evaluate the robustness of the models. The potential negative impact of this research includes possible attacks by malicious users on systems containing DNNs. However, EDA is a white-box attack that assumes the accessibility of model gradients. In addition, it is difficult to access the model gradients involved in a real system. Therefore, it is unsuitable to use EDA to attack a real system. As described above, research benefits are more significant than the potential negative effects. This study helps to improve the robustness of DNNs, allowing them to be more safely applied to a broader range of domains, including safety-critical domains.

Table 22: Comparison in robust accuracy for transfer setting. "source→target" indicates that the transfer attack from source model to the target model. The robust accuracy with a superscript * represents the value reported by Liu et al. (2023b).

| source→target | clean acc (target model) | AA | CAA | AutoAE | $A^3$ | EDA |
|---|---|---|---|---|---|---|
| **CIFAR-10** | | | | | | |
| Carmon et al. (2019)→Ding et al. (2020) | 84.36 | 71.79* | 72.96* | 70.61* | 74.53 | **60.33** |
| Carmon et al. (2019)→Gowal et al. (2020) | 91.10 | - | - | - | 81.28 | **69.07** |
| Ding et al. (2020)→Carmon et al. (2019) | 89.69 | 83.58* | 84.99* | 82.52* | 85.34 | **80.13** |
| Ding et al. (2020)→Gowal et al. (2020) | 91.10 | - | - | - | 88.04 | **83.05** |
| Gowal et al. (2020)→Carmon et al. (2019) | 89.69 | - | - | - | 76.65 | **71.84** |
| Gowal et al. (2020)→Ding et al. (2020) | 84.36 | - | - | - | 76.57 | **69.22** |
| **CIFAR-100** | | | | | | |
| Rice et al. (2020)→Cui et al. (2021) | 62.55 | - | - | - | 58.54 | **56.31** |
| Rice et al. (2020)→Rebuffi et al. (2021) | 63.56 | - | - | - | 59.85 | **57.81** |
| Cui et al. (2021)→Rice et al. (2020) | 53.83 | - | - | - | 46.68 | **41.29** |
| Cui et al. (2021)→Rebuffi et al. (2021) | 63.56 | - | - | - | 55.98 | **52.10** |
| Rebuffi et al. (2021)→Rice et al. (2020) | 53.83 | - | - | - | 47.04 | **40.44** |
| Rebuffi et al. (2021)→Cui et al. (2021) | 62.55 | - | - | - | 53.27 | **47.90** |
| **ImageNet** | | | | | | |
| Salman et al. (2020)→Engstrom et al. (2019) | 62.52 | - | - | - | 56.24 | **52.52** |
| Salman et al. (2020)→Wong et al. (2020) | 55.64 | - | - | - | 48.36 | **44.84** |
| Engstrom et al. (2019)→Salman et al. (2020) | 68.64 | - | - | - | 64.84 | **61.98** |
| Engstrom et al. (2019)→Wong et al. (2020) | 55.64 | - | - | - | 46.74 | **43.08** |
| Wong et al. (2020)→Salman et al. (2020) | 68.64 | - | - | - | 65.66 | **64.00** |
| Wong et al. (2020)→Engstrom et al. (2019) | 62.52 | - | - | - | 57.90 | **55.40** |

## L  EVALUATION OF EDA'S TRANSFERABILITY

We investigated the transferability of EDA between models with different defenses and sizes. The experimental setup is identical to that described in the main text. We mainly use $A^3$ as a baseline for comparison. For further evaluation, we compare the other methods reported in Liu et al. (2023b) for some models. AA (AutoAttck), CAA (Composite Adversarial Attack), and AutoAE are all ensembles of attack methods. AutoAE runs 32 iterations of APGD with CE loss, 63 iterations of APGD with DLR loss, 160 iterations of FAB, and 378 iterations of MultiTargeted attack with nine target classes. According to the official implementation of AutoAE, the MultiTargeted attack runs 378 iterations for each target class. Therefore, AutoAE requires $32 + 63 + 160 + 378 \times 9 = 3,657$ queries for the adversarial attack. The experimental results in Table table 22 show that EDA has higher transferability than the baseline methods in all scenarios.

