# OpenReview forum: "Efficient Diversified Attack: Multiple Diversification Strategies Lead to the Efficient Adversarial Attacks"
_ICLR.cc/2024/Conference — Submitted to ICLR 2024_

### Official Review · Reviewer_pdmN · 2023-10-24

**Soundness:** 3 good
**Presentation:** 2 fair
**Contribution:** 2 fair
**Rating:** 6
**Confidence:** 4

**Summary:**

The paper presents an approach to improving adversarial attacks on deep learning models. The paper emphasizes the importance of diversification and intensification in constructing strong attacks. It introduces the multi-directions/objectives (MDO) strategy, employing multiple search directions and objective functions for diversification. The strategy results in more diverse and potent attacks. Furthermore, the Efficient Diversified Attack (EDA), combining the MDO and multi-target strategies, outperforms existing attacks in terms of diversification and efficiency.

**Strengths:**

- Research on adversarial examples is an important research topic.
- The paper is overall well-written and provides extensive supplementary material.
- While relatively intuitive, the discoveries in this work are novel to the best of my knowledge.
	- Diversification Strategy: Introduces the multi-directions/objectives (MDO) strategy, a unique approach using multiple search directions and objective functions to enhance diversification in adversarial attacks.
	- Efficient Diversified Attack (EDA), combines the MDO and multi-target strategies resulting in a fast and potent attack

**Weaknesses:**

- Overall the performance improvements are only marginal compared to previous results.
- Previous works are concerned with the transferability of adversarial attacks. A transferability evaluation is missing.
- The work mainly evaluates ResNet architectures. These days, the trend moves to the usage of transformer architectures. Hence an evaluation against transformer architectures would be beneficial and more timely.
- The paper uses many abbreviations, making it difficult to comprehend at times.

**Questions:**

Please address the points in my weaknesses section.

---

> ### Author Response · Authors · 2023-11-16
>
> We appreciate the reviewer's comment on our paper. We respond to each comment in turn.
>
> **Weakness 1**
>
> > Overall the performance improvements are only marginal compared to previous results.
>
> We have responded as a Global comment. Please refer to it.
>
> **Weakness 2**
>
> > Previous works are concerned with the transferability of adversarial attacks. A transferability evaluation is missing.
>
> We have tested the transferability of EDA for several models.
> The results show that EDA can generate more transferable adversarial examples than A$^3$ and other baselines, regardless of the data set. The performance improvement is about 5 to 10% for CIFAR10, 2 to 7% for CIFAR100, and 1.6 to 4% for ImageNet. Please take a look at Appendix L, p. 43 in the revised paper for details.
>
> **Weakness 3**
>
> > The work mainly evaluates ResNet architectures. These days, the trend moves to the usage of transformer architectures. Hence an evaluation against transformer architectures would be beneficial and more timely.
>
> We agree that the evaluations against transformer-based architectures are beneficial and more timely. In Appendix I (p.41-42 in the revised paper), we examined the performance of EDA for several models with transformer-based architectures. The experimental results in Table 21 show that EDA consistently outperforms AA and A$^3$ in terms of both computational efficiency and attack performance.
>
> **Weakness 4**
>
> > The paper uses many abbreviations, making it difficult to comprehend at times.
>
> A table summary of the abbreviations was added at the beginning of the appendix to reduce reading difficulty caused by the large number of abbreviations. (Table 3, p.15)
>
>
> We look forward to hearing from you regarding our submission. We would be glad to respond to any further questions and comments that you may have.

---

> > ### Comment · Reviewer_pdmN · 2023-11-22
> > **Thank you for the rebuttal**
> >
> > Hello authors, thank you for the detailed answers. My concerns were mainly addressed. From my side, I am increasing my score for now.

---

### Official Review · Reviewer_hF3e · 2023-10-31

**Soundness:** 3 good
**Presentation:** 2 fair
**Contribution:** 2 fair
**Rating:** 5
**Confidence:** 4

**Summary:**

The paper shows the ability to diversify the adversarial attck based on multi-restart is limited. The authors propose the multi-directions/objectives (MDO) strategy which shows higher diversification performance and attack performance. A combination of MDO and multi-target strategies is also provided. The experimental results show a relationship between attack and diversification performances.

**Strengths:**

- The idea is neat.
- The experimental results are plentiful.

**Weaknesses:**

- The proposed approach is a white-box attack.
- Comparison with ACG attack (Yamamura et al., 2022) is not provided in Table 1.
- From Table 1 and 6, EDA is not better than GS+LS (ADS). The authors should give more analysis on that.
- Too much mathematics.
- It is hard to follow the paper. What is PAS? What does the graph $G(X,\Theta)$ mean?

**Questions:**

- What does the total distortion of the proposed method under $l_\infty$ attack.
- What is the motivation of generalized-DLR (G-DLR) loss, $L_{G-DLR, q}$.
- Does the initial point selection method $\phi$ and the step size update rule $\psi$ matter? The authors do not provide details on these.
- What does "the highest objective values found by an attack $a$ from $R$ initial points" mean?

---

> ### Author Response · Authors · 2023-11-16
> **Author response (1/2)**
>
> We appreciate the reviewer's comment on our paper. We respond to each comment in turn.
>
> **Weakness 1**
>
> > The proposed approach is a white-box attack.
>
> We focus on white-box attacks because the security of deep learning models can be significantly improved by accurate robustness evaluations and the development of robust models based on stronger attacks.
>
> White-box and black-box attacks have different advantages and disadvantages from each other.
> It is known that the white-box setting can create a more potent attack than the black-box case. Therefore, developing white-box attacks helps develop robust models and evaluate robustness. However, from an attacker's point of view, a white-box setting is unrealistic.
>
> The black-box setting assumes the attacker can access the model's output alone. Therefore, the development of black-box attacks has the advantage of leading to more realistic threat assumptions. However, the black-box attacks tend to show lower attack performance than the white-box attacks and require many queries.
>
> **Weakness 2**
>
> > Comparison with ACG attack (Yamamura et al., 2022) is not provided in Table 1.
>
> We added the experimental results for ACG to Table 1. The proposed methods consistently showed higher performance than ACG.
>
> **Weakness 3**
>
> > From Tables 1 and 6, EDA is not better than GS+LS (ADS). The authors should give more analysis on that.
>
> We believe the reviewer is mistaken on this point. We carefully checked the experimental results, but the results showed that EDA consistently performed better than GS+LS (ADS). To make this comparison easier to understand, a table comparing the robust accuracy of EDA and GS+LS (ADS) was added to the Appendix as Table 19 (p.40 in the revised paper).
>
> **Weakness 4**
>
> > Too much mathematics.
>
> A table summary of mathematical symbols was added to the appendix to reduce reading difficulty caused by the large number of mathematical symbols. (Table 4, p.16)
>
> **Weakness 5**
>
> > It is hard to follow the paper. What is PAS? What does the graph $G(X,\theta)$ mean?
>
> PAS stands for Prediction Aware Sampling. We have changed the title of Appendix E (p.35) from Prediction Aware Sampling to Prediction Aware Sampling (PAS). In addition, the definition of $G(X, \theta)$ was revised (p.20, after the equation (10)).
> $G(X,\theta)$ is a graph with nodes $X$ and edges between nodes whose Euclidean distance is less than a given threshold $\theta$.

---

> > ### Author Response · Authors · 2023-11-16
> > **Author response (2/2)**
> >
> > **Question 1**
> >
> > > What does the total distortion of the proposed method under $\ell_\infty$ attack
> >
> > We calculated the average of the 2-norm of the noise as the metrics for quantifying total distortion. For the models trained on CIFAR10, the distortion of EDA averaged over models was 3.025 and 1.569 for A$^3$; for CIFAR100, 3.185 for EDA and 1.609 for A$^3$; for ImageNet, 14.109 for EDA and 5.992 for A$^3$.
> >
> > **Question 2**
> >
> > > What is the motivation of generalized-DLR (G-DLR) loss, $L_{G-DLR, q}$.
> >
> > The motivation for proposing generalized-DLR loss is to increase diversity in the output space. For the search points with larger values of G-DLR loss, the denominator/numerator will be smaller/larger value, respectively. The small value of denominator $g_{\pi_1}-g_{\pi_q}$ means that the values of $g_{\pi_1},g_{\pi_2},\ldots,g_{\pi_q}$ are close. Therefore, the large and small relationships between the predicted probability of classification classes are expected to be easier to variate than in the case of DLR loss.
> >
> > **Question 3**
> >
> > > Does the initial point selection method $\phi$ and the step size update rule $\psi$ matter? The authors do not provide details on these.
> >
> > Yes. However, it is not as important as the objective function or the update formulas, and there are not as many options. Previous studies [1,2,3] have shown that initial point sampling considering model output is more effective than simple random sampling. The ablation studies of EDA have shown that EDA is relatively robust to differences in initial point sampling.
> > For the step size update rule, gradually decreasing the step size from a large value is considered effective [2,3]. The existing step size update rules are heuristic, and high performance can be expected from any of the methods.
> >
> > [1] Yusuke Tashiro, Yang Song, and Stefano Ermon. Diversity can be transferred: Output diversification for white- and black-box attacks. NeurIPS 2020
> >
> > [2] Francesco Croce and Matthias Hein. Reliable evaluation of adversarial robustness with an ensemble of diverse parameter-free attacks. ICML2020
> >
> > [3] Ye Liu, Yaya Cheng, Lianli Gao, Xianglong Liu, Qilong Zhang, and Jingkuan Song. Practical
> > evaluation of adversarial robustness via adaptive auto attack. CVPR2022
> >
> > **Question 4**
> >
> > > What does "the highest objective values found by an attack $a$ from $R$ initial points" mean?
> >
> > It means the objective values of each best search points found after $R$ runs of the attack $a$ with different initial points. In other words, the points in $X^*(x_i, a, R)$ are the best search points found by the attack $a$ starting at each initial point. The best search point is the point whose objective value is higher than the other search points. The revised paper provides the precise mathematical definition in Table 4 (p.16).
> >
> > We look forward to hearing from you regarding our submission. We would be glad to respond to any further questions and comments that you may have.

---

### Official Review · Reviewer_GZwH · 2023-11-01

**Soundness:** 2 fair
**Presentation:** 3 good
**Contribution:** 2 fair
**Rating:** 5
**Confidence:** 3

**Summary:**

This paper contributes a white-box adversarial attack framework of Efficient Diversified Attack (EDA) with a new multi-directions/objectives (MDO) strategy. This method focuses on the diversification of adversarial examples (AE) by utilizing multiple attack methods and objective or loss function. The effectiveness of MDO depends on appropriate search directions (δ) and objective functions (L). Thus, the authors propose an Automated Diversified Selection (ADS) algorithm to select the combinations of δ and L. The MDO strategy consists of two phases: the diversification phase (global search, GS) and the intensification phase (local search, LS). The authors conduct experiments to demonstrate the effectiveness of ADS and the GS+LS, and the efficiency of MDO in comparison to baselines.

**Strengths:**

1.	The paper is demonstrated well. The procedure figure and most diagrams are easy to understand. The presentation is easy to follow.

2.	The authors test their methods on 41 models and 21 different defenses, covering typical datasets of CIFAR-10/100 and ImageNet.

3.	The method reduces the number of queries and time spent to 86.9% on average, according to Table 2.

**Weaknesses:**

1.	Even though this method saves time, the increase in attack success rate is little.
In Table 2, the delta of accuracy is very small. In Table 6 in the appendix, the delta of accuracy is less than 0.1% in most cases.

2.	The diversification of EDA is not good enough.
In Figure 5, the Diversity Index (DI) of MT_cos exceeds that of the authors' methods.

3.	The authors’ methods are not consistently effective.
In Table 6, there are some cases that the authors’ method even takes longer time than the baseline. Considering the weakness in lack of improvement in diversification and effectiveness, could the authors discuss the tradeoff between efficiency and diversification?

4.	This paper does not evaluate the effectiveness of combining these methods with their method.
There are many existing methods for improving the success rate of adversarial attacks, such as [1] and [2]. Can EDA and MDO be combined with them? If so, what will the diversification and effectiveness of the authors’ method and baselines be?

5.	There is a contradiction between the text and the figure.
In section 4.1, the paper claims that “Figure 6 indicates that GS+LS found AEs in fewer queries than MT_cos”. However, in Figure 6, it seems GS+LS and MT_cos have similar numbers of queries, and there are more cases in which MT_cos takes fewer queries. Could the authors explain the reason? What do the average lines represent?

[1] Xie, C., Zhang, Z., Zhou, Y., Bai, S., Wang, J., Ren, Z., & Yuille, A. L. (2019). Improving transferability of adversarial examples with input diversity. In Proceedings of the IEEE/CVF conference on computer vision and pattern recognition (pp. 2730-2739).

[2] Dong, Y., Pang, T., Su, H., & Zhu, J. (2019). Evading defenses to transferable adversarial examples by translation-invariant attacks. In Proceedings of the IEEE/CVF Conference on Computer Vision and Pattern Recognition (pp. 4312-4321).

**Questions:**

1.	Can EDA and MDO be combined with other augmentation methods mentioned above? If so, what will the diversification and effectiveness of the authors’ method and baselines be?

2.	Could the authors explain the data in Figure 6? What do the average lines represent?

3.	Considering the weakness in lack of improvement in diversification and effectiveness, could the authors discuss the tradeoff between efficiency and diversification?

---

> ### Author Response · Authors · 2023-11-16
> **Author response (1/3)**
>
> > Considering the weakness in lack of improvement in diversification and effectiveness, could the authors discuss the tradeoff between efficiency and diversification?
>
> The reviewer has raised an important question. Before answering the question directly, let me explain the relationship between attack performance, diversification, and intensification.
>
> ### For weakness 1 (The increase in attack success rate is little.)
>
> As discussed in detail in the global comment, the performance gains from the proposed method are large enough compared to the improvement in recent attacks.
>
> ### For weakness 2 (Not enough improvement in diversification)
>
> As Weakness 2, the reviewer pointed out that EDA is not diversified enough because the Diversity Index of EDA is smaller than that of MTcos. However, higher diversification is not always better. What is important in improving search performance is to strike a balance between diversification and intensification [1]. An example is step size control. Searching with a large step size promotes diversification, but it is difficult to further improve the objective value by continuing the search at the same step size. However, gradually decreasing the step size can adjust the balance between diversification and intensification. As a result, as existing studies have shown, the control of gradually decreasing step size contributes to performance improvement. Given that EDA showed higher attack performance than MTcos, Figure 5 just shows that EDA has a more appropriate balance of diversification and intensification than MTcos.
>
> In addition, there is no general definition of diversification. In the main text, we evaluated diversification based on the Diversity Index to check whether the search points form any cluster. However, different definitions of diversification may lead to different analysis results. In Appendix D.6 (p.31 in the revised paper), we quantified the degree of diversification using a measure based on the Euclidean distance. This analysis enables us to understand the diversification of EDA from a perspective that cannot be considered in the Diversity Index. Figure 14 (p.32 in the revised paper) shows the difference in the degree of diversification measured by the Euclidean distance-based measure. In this case, the difference in the degree of diversification between EDA and MT$_{cos}$ is smaller than those measured by the Diversity Index.
>
> The analysis results using the two diversification indices suggest that EDA and MT$_{cos}$ diversify their search differently.  Based on the above description, we discuss the tradeoff between diversification and efficiency.
>
> ### Trade off between efficiency and diversification.
>
> Diversification and computational efficiency are not necessarily trade-offs. If diversification and intensification are appropriately balanced, it is possible to enumerate different local solutions faster. In other words, appropriate diversification and intensification may improve both computational efficiency and search performance at the same time. The experimental results in Table 8 in our revised paper show that EDA shows higher attack performance in less computation time than Adaptive Auto Attack for many models. Therefore, Table 8 and Figure 5 in our revised paper support our contention that EDA balances diversification and intensification well.
>
> In addition, differences in attack performance should be considered when comparing computational efficiency. Table 8 in the revised paper shows that EDA showed higher attack performance than Adaptive Auto Attack for all models where EDA spent relatively more computation time.
> Adaptive Auto Attack terminates its search with a different number of queries per model depending on its search progress. Therefore, the potential reason for the difference in efficiency is that Adaptive Auto Attack terminates its search, leaving room for performance improvement, resulting in a shorter time than EDA.
>
> [1] CrepinšekMatej, LiuShih-Hsi, and MernikMarjan. Exploration and exploitation in evolutionary algorithms. ACM Computing Surveys (CSUR), 45(3):33, jul 2013. doi: 10.1145/2480741.2480752.

---

> > ### Author Response · Authors · 2023-11-16
> > **Author response (2/3)**
> >
> > > Could the authors explain the data in Figure 6? What do the average lines represent?
> >
> > This analysis using Figure 6 examines how many queries method A takes, on average, to find an adversarial example compared to method B when method A takes more queries than method B.
> > Figures 6 and 13 in the revised paper illustrate the difference in the number of queries and their averages for images where attack methods A and B successfully found adversarial examples, but took different numbers of queries. For example, "average (MTcos takes more queries)" in Figures 6 and 13 represents the average difference in the number of queries required by MTcos and those by GS+LS for images where MTcos spent more queries. Based on the comparison between "average (MTcos takes more queries)" and "average (MTcos takes fewer queries)", we argue in the main text that "Figure 6 shows that GS+LS found adversarial examples in fewer queries on average than MTcos".

---

> > > ### Author Response · Authors · 2023-11-16
> > > **Author response (3/3)**
> > >
> > > > Can EDA and MDO be combined with other augmentation methods mentioned above? If so, what will the diversification and effectiveness of the authors’ method and baselines be?
> > >
> > > Technically, it is possible to combine EDA with suggested methods. The suggested methods are expected to promote diversification because of their random influence on the search direction. However, as mentioned above, striking the appropriate balance between diversification and intensification is essential. Therefore, it is difficult to know in advance whether they will contribute to improved computational efficiency or attack performance. Although we experimented with a combination of the suggested methods and EDA in the default setting, the results were not better. However, there is room for improving attack performance by adjusting the suggested methods' hyperparameters.
> > >
> > > We look forward to hearing from you regarding our submission. We would be glad to respond to any further questions and comments that you may have.

---

> > ### Comment · Reviewer_GZwH · 2023-11-23
> >
> > Thanks for your response. If higher diversification is not always better, why do we need to compare different methods in terms of their diversification? Even when using the new Euclidean distance-based measure, the proposed method does not perform significant better.

---

### Official Review · Reviewer_C5Ls · 2023-11-01

**Soundness:** 2 fair
**Presentation:** 1 poor
**Contribution:** 2 fair
**Rating:** 5
**Confidence:** 4

**Summary:**

In this paper, the authors propose techniques to enhance the adversarial attack suite in terms of effectiveness and efficiency against robust models. Specifically, they start with a given search space containing possible gradient directions $\delta$ and losses $L$, and identify the top combinations of search direction $\delta$ and objective functions $L through Automated diversification search (ADS). They then proceed with a two-stage approach, 1 initial coarse-grained attack (GS) followed by a  2. localized attack within the region found by the best point in stage 1.  Furthermore, they combine the proposed approach with MultiTarget (MT) to obtain an Efficient diversified Attack.

**Strengths:**

- The authors present a unified formulation of an adversarial attack, encompassing aspects such as initialization, step size updates, search directions, and loss functions. This lets us to study each factor in a fine-grained manner and ultimately improve the attack performance

- The empirical evaluation is conducted on 41 models, which unquestionably provides a clear and comprehensive perspective and establishes robust baselines for future research endeavors.

**Weaknesses:**

**Presentation:**

- In my initial reading, I found the paper's notation to be confusing. Moreover, all the crucial algorithms (ADS, GS, LS, EDA) are deferred to the Appendix, making for a less-than-smooth reading experience. Within the Appendix itself, these algorithms are distributed at different places in the 38-page paper, further complicating the reading process. I would request to consider moving a few of them (ADS, GS, LS, EDA, Target selection) to the main draft

- As also suggested by Reviewer pdmN, this paper had many abbreviations and particularly difficult to follow sometimes. I request the authors to simplify the paper presentation and notations.

**Technical:**

- In my view, the proposed ADS for selecting the most effective search direction ($\delta$) and loss function ($L$) based on the diversity measures at both input and output, appears quite straightforward. Additionally, the Diversity Index (DI) has been widely studied in Yamamura et al. 2021, thereby paper's technical contribution may be somewhat limited as it just picks the combinations based on this measure and $P_i^e$.

- The GS and LS techniques introduced within the search framework are, to some extent, heuristic in nature. Although ablation studies have been carried out with respect to N1, N2, and N3, the current framework's configuration, comprising GS with N1 and N2 iterations, followed by LS with N3 iterations, strikes me as somewhat heuristic.

- Upon closer examination, the differences in performance between ADS and Random appear negligible in Table 1, and the variation between GS+LS (ADS) and GS+LS (Random) seems relatively small. On average, across nine models, this difference stands at less than approximately 0.05% (please correct me if I'm wrong).

- Notably, there exists a substantial number of hyperparameters tied to ADS, GS, LS, and the target selection schemes. While the paper presents ablation studies, I anticipate that the widespread adoption of the proposed method as a robust alternative to the AutoAttack suite might pose a considerable challenge.

- ADS requires access to the dataset. The authors perform ADS on 1% of the data, while AutoAttack and other baseline attacks operate on a per-datapoint basis without depending on additional datapoints.

**Questions:**

- Why are the step sizes set to 2*$\epsilon$ for $N_1$ iterations, $N_2$ iterations with $\epsilon$, and $N_3$ iterations with $\epsilon/2$?

- What is the individual influence of the two terms $P_i^e$ and DI(.) in the overall ADS search performance?

- What combinations were discovered by ADS for the different models? Are there any insights into the top $n_a$ combinations found by ADS for different models?

- Why is ADS performed in the LS stage? What are the benefits of conducting it before LS? While I understand that the initial points are different at these two stages, could the authors provide an empirical analysis of employing ADS in the LS stage?

- Are top combinations {$\delta_{a_j^*}, L_{a_j^*}$} found at GS, LS stages correlated?

- Can you report the average performance over  9 models in Table 1 for all methods

- In the absence of a dataset to perform ADS, is the combinations found by ADS transferable in the cross-domain setting?

---

> ### Author Response · Authors · 2023-11-16
> **Author response (1/3)**
>
> We appreciate the reviewer's comments on our paper. The order of comments has been changed in order to provide a concise and clear response.
>
> ### Technical weakness 1
>
> > In my view, the proposed ADS for selecting the most effective search direction and loss function based on the diversity measures at both input and output, appears quite straightforward. Additionally, the Diversity Index (DI) has been widely studied in Yamamura et al. 2021, thereby paper's technical contribution may be somewhat limited as it just picks the combinations based on this measure and $P_i^e$.
>
> We acknowledge that ADS has a clear and simple approach. However, the major contribution of ADS is its ability to determine the combination of objective function and update direction at low computational cost.
>
> Although various objective functions and update formulas have been proposed, no method has been proposed to make good use of them. The related research is those for finding ensembles of attacks, such as CAA [1] and AutoAE [2]. These methods determine the ensemble based on the attack success rate or the best objective function value of the candidate attack methods.
> Therefore, they are computationally expensive because the candidate attack methods must be executed in advance. Due to the many combinations of objective functions and search directions, it is impractical to select a pair of objective functions and search directions using these methods from a computational cost perspective.
>
> On the contrary, by focusing on the degree of search diversification, ADS has dramatically reduced the computational cost of combination decisions. ADS allows for better utilization of existing objective functions and update formulas, which may encourage the development of adversarial robustness research. In addition, ADS could facilitate the automation of the robustness evaluation process. For example, we can construct an ensemble of attacks from automatically generated candidates of the ensemble using ADS, CAA, and AutoAE. Thus, ADS is considered to be a versatile and valuable algorithm.
>
> [1] Xiaofeng Mao, Yuefeng Chen, Shuhui Wang, Hang Su, Yuan He, and Hui Xue. Composite adversarial attacks. AAAI 2021.
>
> [2] Shengcai Liu, Fu Peng, and Ke Tang. Reliable Robustness Evaluation via Automatically Constructed Attack Ensembles. AAAI 2023.
>
> ### Technical weakness 2 and question1
>
> > The GS and LS techniques introduced within the search framework are, to some extent, heuristic in nature. Although ablation studies have been carried out with respect to N1, N2, and N3, the current framework's configuration, comprising GS with N1 and N2 iterations, followed by LS with N3 iterations, strikes me as somewhat heuristic.
>
> > Why are the step sizes set to 2$\epsilon$ for $N_1$ iterations, $N_2$ iterations with $\epsilon$, and  $N_3$ iterations with $\epsilon/2$?
>
> We determined the step size inspired by the APGD heuristic, which is widely used and works well. The APGD heuristic has eight checkpoints. The checkpoints are written down as $w_1=\lceil0.22N\rceil, w_2=\lceil0.41N\rceil, w3=\lceil0.57\rceil,...,w_8=\lceil0.99N\rceil$, where $N$ is the total iterations.
> Our framework's $N_1$, $N_2$, and $N_3$ iterations correspond to the search from checkpoint $w_1$, $w_1$ to $w_2$, and $w_2$ and beyond in the APGD heuristic, respectively.
>
> APGD has an initial step size of $2\varepsilon$ and halves the step size if the condition is satisfied at each checkpoint. If the condition is satisfied at all checkpoints, the step size is $2\varepsilon$ up to $w_1$, $\varepsilon$ between $w_1$ and $w_2$, and $\varepsilon/2$ after $w_2$. Based on this, we set the step size in the proposed framework to $2\varepsilon$, $\varepsilon$, and $\varepsilon/2$ for $N_1$, $N_2$, and $N_3$ iterations, respectively.
>
> ### Technical weakness 4
>
> > Notably, there exists a substantial number of hyperparameters tied to ADS, GS, LS, and the target selection schemes. While the paper presents ablation studies, I anticipate that the widespread adoption of the proposed method as a robust alternative to the AutoAttack suite might pose a considerable challenge.
>
> EDA can be used without hyperparameter tuning in practice, similar to AutoAttack.
>
> AutoAttack's components, APGD, FAB, and Square attack, have explicitly given hyperparameters and magic numbers heuristically determined by the authors. While AutoAttack is not inherently parameter-free, their default parameters are carefully determined.
>
> Our ablation study suggests that EDA is robust to hyperparameter settings. In addition, the particularly influential parameter, step size, is determined based on the APGD heuristic, which is widely used and effective. Therefore, EDA can be used without hyperparameter tuning in practice.

---

> ### Author Response · Authors · 2023-11-16
> **Author response (2/3)**
>
> ### Technical weakness 3
>
> > Upon closer examination, the differences in performance between ADS and Random appear negligible in Table 1, and the variation between GS+LS (ADS) and GS+LS (Random) seems relatively small. On average, across nine models, this difference stands at less than approximately 0.05% (please correct me if I'm wrong).
>
> As mentioned in the global comment, based on the existing research, even an improvement in attack performance from AutoAttack of less than 0.1% is considered sufficient. Considering that GS+LS(ADS) showed competitive attack performance as AutoAttack, the difference between GS+LS(ADS) and GS+LS(random) is large enough.
>
> In addition, the performance of GS+LS(random) is expected to be unstable because GS+LS(random) is completely dependent on random numbers for selection. On the contrary, GS+LS(ADS) are expected to show stable and high performance.
>
>
> ### Questions 2 and 5
>
> > Question 5: Are top combinations $\{(\delta_a, L_a)\}$ found at GS, LS stages correlated?
>
> The brief answer is No. The potential reasons are: 1. $P_i^e$ and DI play different roles from each other, 2. the ACG's search direction may be similar to the steepest for small step sizes, and 3. the difference between Nesterov's acceleration gradient direction and gradient direction. Appendix D.8 (p. 33 in revised paper) provides further information.
>
> We explain the role of $P_i^e$ and DI as the answer to the question 2.
>
> > Question 2: What is the individual influence of the two terms $P_i^e$ and DI(.) in the overall ADS search performance?
>
> The $P_i^e$ measures the degree of diversification in the output space during the search. Therefore, a pair with the largest $P_i^e$ is expected to search for a high diversity in the output space. In addition, from Yamamura et al. (2022), it can be assumed that the ACG direction increases $P_i^e$, while the steepest-like direction does not. From the above, it is considered that the pair with the maximum $P_i^e$ is likely to include the ACG direction.
>
> In our use case, DI measures the diversity of the best point set obtained by the search. In other words, DI represents the dissimilarity between the best points. That is, we expect that pairs with the largest DI are more likely to enumerate dissimilar solutions. Intuitively, updates in diverse directions contribute to the enumeration of dissimilar solutions. Given that the search direction is gradient-dependent, the pair with the largest DI is likely to include a variety of objective functions and update formulas.
>
> ### Question 4
>
> > Why is ADS performed in the LS stage? What are the benefits of conducting it before LS? While I understand that the initial points are different at these two stages, could the authors provide an empirical analysis of employing ADS in the LS stage?
>
> The advantage of performing ADS in LS is that it reduces the possibility of redundant searches. The search direction of ACG is likely to be selected in GS, and the direction of ACG is more likely to be similar to the steepest direction for small step sizes. Therefore, an efficient search may not be possible if the search is continued using the pair selected in GS. In addition, given the multimodal nature of the objective function, searching in Nesterov's accelerated gradient direction and the gradient direction may help to find different local solutions.
>
> ### Question 3
>
> > What combinations were discovered by ADS for the different models? Are there any insights into the top $n_a$ combinations found by ADS for different models?
>
> There are slight differences in the pairs selected for each model. However, the trend is consistent: the ACG's direction is more likely to be selected for GS, and the Nesterov's accelerated gradient direction is more likely to be selected for LS.
> In addition, experimental results on EDA's transferability suggest that EDA can generate highly transferable adversarial examples.
> These experimental results suggest that different models may have some common features.

---

> > ### Author Response · Authors · 2023-11-16
> > **Author response (3/3)**
> >
> > ### Weakness (presentation)
> >
> > > In my initial reading, I found the paper's notation to be confusing. Moreover, all the crucial algorithms (ADS, GS, LS, EDA) are deferred to the Appendix, making for a less-than-smooth reading experience. Within the Appendix itself, these algorithms are distributed at different places in the 38-page paper, further complicating the reading process. I would request to consider moving a few of them (ADS, GS, LS, EDA, Target selection) to the main draft.
> >
> > We have carefully considered the comment, but the pseudocodes listed in the appendix are interrelated, and including only some of them in the text may cause confusion.
> > Instead, we have grouped the pseudocodes closely and added a diagram explaining the dependencies between the pseudocodes in the appendix.
> >
> > ### Technical weakness 5 and question 7
> >
> > > ADS requires access to the dataset. The authors perform ADS on 1% of the data, while AutoAttack and other baseline attacks operate on a per-datapoint basis without depending on additional datapoints.
> >
> > > In the absence of a dataset to perform ADS, is the combinations found by ADS transferable in the cross-domain setting?
> >
> > We have tested the transferability of ADS in the cross-domain setting for several models in CIFAR10 and ImageNet. The experimental results, summarized in the table below, suggest that the proposed method may work well in the cross-domain setting. A possible reason for this is that the pairs selected by ADS have a consistent trend regardless of the model under attack.
> >
> > | source->target                                                       | clean accuracy (target) | EDA (cross-domain) | EDA   |
> > | -------------------------------------------------------------------- | ------------------- | ----------------------- | ----- |
> > | Salman et al.,2020->Addepalli et al.,2022 (ImageNet->CIFAR10)   | 85.71               | 52.42                   | 52.43 |
> > | Salman et al.,2020 ->Carmon et al.,2019 (ImageNet->CIFAR10)     | 89.69               | 59.44                   | 59.40 |
> > | Salman et al.,2020 ->Sitawarin et al.,2021 (ImageNet->CIFAR10)  | 86.84               | 50.64                   | 50.58 |
> > | Addepalli et al.,2022->Salman et al.,2020 (CIFAR10->ImageNet)   | 64.02               | 34.52                   | 34.52 |
> > | Addepalli et al.,2022->Wong et al.,2020 (CIFAR10->ImageNet)     | 55.62               | 26.24                   | 26.12 |
> > | Addepalli et al.,2022->Engstrom et al.,2019 (CIFAR10->ImageNet) | 62.56               | 28.96                   | 29.01 |
> >
> > ### Question 6
> >
> > > Can you report the average performance over 9 models in Table 1 for all methods
> >
> > The average robust accuracy values over the nine models listed in Table 1 are as follows.
> >
> > | AA    | Naive | GS(ADS) | LS(ADS) | GS+LS(ADS) | GS+LS(R-ADS) | GS+LS(RAND) | MT$_{cos}$ |
> > | ----- | ----- | ------------- | ------------- | ---------------- | ------------------ | ----------------- | ------------ |
> > | 37.83 | 38.09 | 38.50         | 38.66         | 37.84            | 37.99              | 37.90             | 37.87        |
> >
> > We look forward to hearing from you regarding our submission. We would be glad to respond to any further questions and comments that you may have.

---

### Author Response · Authors · 2023-11-16
**Global comment (1/2)**

We wish to express our appreciation to the reviewers for their insightful comments on our paper. The comments have helped us significantly improve the paper. As a global comment, we discuss the performance improvement of recent attacks compared to AutoAttack and the revisions to the paper.

## Performance improvement

Several reviewers noted that the improvement in attack performance with EDA is marginal. However, existing research has shown that improving attack performance against robust models listed in RobustBench within the number of queries close to our setting is challenging.
The following table summarizes representative experimental results reported in the literature [1-4] for the models described in RobustBench.
As described in the following table, the difference in robust accuracy between AutoAttack (AA) and the other existing methods is less than 0.1% in many cases.
The existing methods that have shown better performance than AutoAttack are as follows: Composite Adversarial Attack (CAA, AAAI-21) [1], Automated Discovery of Adaptive Attacks (A$^2$, NeurIPS-21) [2], Adaptive Auto Attack (A$^3$, CVPR-22) [3], and AutoAE (AAAI-23) [4]. Of note, the robust accuracy of Adaptive Auto Attack (A$^3$)  is the value described in Table 8 of our revised paper because the clean accuracy reported by [3] differs from the value in the RobustBench leaderboard.

| paper | model | AA | CAA$_{sub}$ | A$^2$ | A$^3$ | AutoAE | EDA (proposed) |
| --- | --- | --- | --- | --- | --- | --- | --- |
| Engstrom et al.  | RN50 | 49.25 | 49.18 | - | 49.25   | - | 49.10 |
| Hendrycks et al. | WRN28-10 | 54.92 | 54.82 | - | 54.85   | - | 54.77 |
| Carmon et al.    | WRN28-10 | 59.53 | 59.45 | 59.51   | 59.44   | 59.38  | 59.40 |
| Sehwag et al.    | WRN28-10 | 57.14 | - | 57.16   | 57.14   | - | 57.03 |
| Gowal et al.     | WRN28-10 | 62.80 | - | 62.79   | 62.77   | 62.79  | 62.75 |
| Wu et al.        | WRN28-10 | 60.04 | - | 60.01   | 60.02   | 60.00  | 59.94 |

When comparing attack performance, computation time must be taken into account.
Therefore, we compared the number of queries for methods that use the same number of queries for different models and the computation time for the methods that do not. The number of queries for AA, CAA$_{sub}$ and EDA is described in detail in Appendix D.3 of our revised paper. AutoAE runs 32 iterations of APGD with CE loss, 63 iterations of APGD with DLR loss, 160 iterations of FAB, and 378 iterations of MultiTargeted (MT) attack with nine target classes. According to the official implementation of AutoAE, the MT attack runs 378 iterations for each target class. Therefore, AutoAE requires 32+63+160+378 $\times$ 9=3,657 queries for the adversarial attack.

| | AA | CAA$_{sub}$ | A$^2$ | A$^3$ | AutoAE | EDA |
| --- | --- | --- | --- | --- | --- | --- |
| \#query | 6,100 | 800  | - | - | 3,657  | 692.24 (CIFAR10), 742.24 (CIFAR100), 802.24 (ImageNet) |

The following table compares execution times of A$^2$, A$^3$, and EDA. Of note, A$^2$ is executed on RTX 2080Ti, whereas A$^3$ and EDA are executed on RTX3090.

| paper | model | A$^2$ (time in seconds) | A$^3$ (time in seconds) | EDA (time in seconds) |
| --- | --- | ---- | --- | --- |
| Carmon et al. | WRN28-10 | 16,920 | 4,223 | 3,316 |
| Sehwag et al. | WRN28-10 | 25,740 | 2,662 | 3,225 |
| Gowal et al.  | WRN28-10 | 13,560  | 3,841| 4,557 |
| Wu et al.     | WRN28-10 | 15,300 | 3,273 | 3,520 |

Based on the above comparison, the performance improvement of EDA against robust models listed in RobustBench is sufficiently large in computation time and attack performance.

In addition, EDA reduced robust accuracy by up to 17.96% against randomized defenses (Appendix H, Table 20 in the revised paper). Given that Adaptive Auto Attack showed lower attack performance than AutoAttack for randomized defenses, the contribution of EDA in improving attack performance is significant enough.

[1] Xiaofeng Mao, Yuefeng Chen, Shuhui Wang, Hang Su, Yuan He, and Hui Xue. Composite adversarial attacks. AAAI 2021.

[2] Chengyuan Yao, Pavol Bielik, Petar Tsankov, and Martin Vechev. Automated Discovery of Adaptive Attacks on Adversarial Defenses. NeurIPS 2021.

[3] Ye Liu, Yaya Cheng, Lianli Gao, Xianglong Liu, Qilong Zhang, and Jingkuan Song. Practical evaluation of adversarial robustness via adaptive auto attack. CVPR 2022.

[4] Shengcai Liu, Fu Peng, and Ke Tang. Reliable Robustness Evaluation via Automatically Constructed Attack Ensembles. AAAI 2023.

---

> ### Author Response · Authors · 2023-11-16
> **Global comment (2/2)**
>
> ## Revision of the paper
>
> ### Main text
>
> - A comparison with ACG was added to Table 1. (p.8)
> - The robust accuracy of AA --> AA's robust accuracy (Caption of Table 2, p.9)
> - ViT-based --> transformer-based (Additional results in Section 4.2, p.9)
> - The reference to experiments on Transferability was added. (Additional results in Section 4.2, p.9)
>
> ### Appendix
>
> - A table summary of the abbreviations was added at the beginning of the appendix to reduce reading difficulty caused by the large number of abbreviations. (Table 3, p.15)
> - A table summary of mathematical symbols was added to the appendix to reduce reading difficulty caused by the large number of mathematical symbols. (Table 4, p.16)
> - The pseudocodes were relocated closer to each other at the beginning of the appendix, and a figure was added to illustrate the relationship between pseudocodes. (p.17-19, Figure 7)
> - The definition of $G(X, \theta)$ was revised. (p.20, after the equation (10))
> - A detailed explanation of Figure 6 was added. (Appendix D.5, p.31)
> - The further explanation of the reason for ADS's search trends were added. (Appendix D.8, p.33)
> - Prediction Aware Sampling --> Prediction Aware Sampling (PAS) (Appendix E, p.35)
> - A table was added for direct comparison with EDA and GS+LS(ADS) in robust accuracy. (p.40, Table 19)
> - The evaluation of EDA's transferability was added. (Appendix L, p.43)
>
> We look forward to hearing from you regarding our submission. We would be glad to respond to any further questions and comments that you may have.

---

### Meta-Review · Area_Chair_9X2i · 2023-12-06

**Metareview:**

The paper introduces a white-box adversarial attack framework that incorporates a multi-directional/objective (MDO) strategy to enhance attack effectiveness and efficiency. MDO employs a two-phase search approach with multiple attack methods and objective functions to diversify the adversarial examples. While reviewers acknowledge the paper's contributions, several concerns have been raised, encompassing the marginal performance improvements, evaluations on limited architectures, and the rationale behind certain heuristic-based design choices. In the rebuttal phase, the authors made a good effort to address these concerns. Nevertheless, the primary concern regarding the marginal performance increase remains unresolved. Besides, almost all reviewers challenge the paper's presentation, including excessive use of abbreviations/mathematics, and an organization that scatters algorithmic explanations in appendices. Overall, reviewers acknowledge the merit of this paper, but the concerns of minor performance gain and potentially misleading organization blur the contribution and significance of the proposed framework.

**Justification For Why Not Higher Score:**

The paper was not assigned a higher score mainly due to the unresolved issue of marginal performance improvements of the proposed white-box adversarial attack framework and concerns about the presentation, including excessive abbreviations and mathematics, and dispersed algorithmic explanations. These factors obscure the framework's contribution and significance.

**Justification For Why Not Lower Score:**

N/A

---

### Decision · Program_Chairs · 2024-01-16

Reject